# A Model Intercomparison of CCN-Limited Tenuous Clouds in the High Arctic

Robin G. Stevens[1,*], Katharina Loewe[2], Christopher Dearden[3,**], Antonios Dimitrelos[4], Anna Possner[5,6], Gesa K. Eirund[5], Tomi Raatikainen[7], Adrian A. Hill[8], Benjamin J. Shipway[8], Jonathan Wilkinson[8], Sami Romakkaniemi[9], Juha Tonttila[9], Ari Laaksonen[7], Hannele Korhonen[7], Paul Connolly[3], Ulrike Lohmann[5], Corinna Hoose[2], Annica M. L. Ekman[4], Ken S. Carslaw[1], and Paul R. Field[1,8]

[1]Institute of Climate and Atmospheric Science, School of Earth and Environment, University of Leeds, Leeds, UK
[2]Institute of Meteorology and Climate Research, Karlsruhe Institute of Technology, Karlsruhe, Germany
[3]Centre for Atmospheric Science, School of Earth and Environmental Sciences, University of Manchester, Manchester, UK
[4]Department of Meteorology, Stockholm University, Stockholm, Sweden
[5]Institute for Atmospheric and Climate Science, Eidgenössische Technische Hochschule, Zürich, Switzerland
[6]Department of Global Ecology, Carnegie Institution for Science, Stanford, CA, USA
[7]Finnish Meteorological Institute, Helsinki, Finland
[8]Met Office, Exeter, UK
[9]Finnish Meteorological Institute, Kuopio, Finland
[*]Now at Air Quality Research Division, Environment and Climate Change Canada, Dorval, Canada
[**]Now at the Centre of Excellence for Modelling the Atmosphere and Climate, School of Earth and Environment, University of Leeds, Leeds, UK

*Correspondence to:* R. G. Stevens (robin.stevens@canada.ca)

**Abstract.** We perform a model intercomparison of summertime high Arctic (>80N) clouds observed during the 2008 Arctic Summer Cloud Ocean Study (ASCOS) campaign, when observed cloud condensation nuclei (CCN) concentrations fell below $1 \, \mathrm{cm}^{-3}$. Previous analyses have suggested that at these low CCN concentrations the liquid water content (LWC) and radiative properties of the clouds are determined primarily by the CCN concentrations, conditions that have previously been referred to as the tenuous cloud regime. The intercomparison includes results from three large eddy simulation models (UCLALES-SALSA, COSMO-LES, and MIMICA) and three numerical weather prediction models (COSMO-NWP, WRF, and UM-CASIM). We test the sensitivities of the model results to different treatments of cloud droplet activation, including prescribed cloud droplet number concentrations (CDNC) and diagnostic CCN activation based on either fixed aerosol concentrations or prognostic aerosol with in-cloud processing.

There remains considerable diversity even in experiments with prescribed CDNCs and prescribed ice crystal number concentrations (ICNC). The sensitivity of mixed-phase Arctic cloud properties to changes in CDNC depends on the representation of the cloud droplet size distribution within each model, which impacts on autoconversion rates. Our results therefore suggest that properly estimating aerosol-cloud interactions requires an appropriate treatment of the cloud droplet size distribution within models, as well as in-situ observations of hydrometeor size distributions to constrain them.

The results strongly support the hypothesis that the liquid water content of these clouds is CCN-limited. For the observed meteorological conditions, the cloud generally did not collapse when the CCN concentration was held constant at the relatively

high CCN concentrations measured during the cloudy period, but the cloud thins or collapses as the CCN concentration is reduced. The CCN concentration at which collapse occurs varies substantially between models. Only one model predicts complete dissipation of the cloud due to glaciation, and this occurs only for the largest prescribed ICNC tested in this study. Global and regional models with either prescribed CDNCs or prescribed aerosol concentrations would not reproduce these dissipation events. Additionally, future increases in Arctic aerosol concentrations would be expected to decrease the frequency of occurrence of such cloud dissipation events, with implications for the radiative balance at the surface. Our results also show that cooling of the sea-ice surface following cloud dissipation increases atmospheric stability near the surface, further suppressing cloud formation. Therefore, this suggests that linkages between aerosol and clouds, as well as linkages between clouds, surface temperatures and atmospheric stability need to be considered for weather and climate predictions in this region.

## 1 Introduction

A decrease in Arctic sea ice extent and thickness has been observed within recent decades (Vaughan et al., 2013). Further decreases in Arctic sea ice extent are expected to increase the fluxes of aerosol and aerosol precursor gases (Struthers et al., 2011; Corbett et al., 2010) as well as latent heat and sensible heat from the open ocean surface within the Arctic (Boisvert and Stroeve, 2015). Long-range transport of anthropogenic aerosol is currently a significant source to the Arctic region (Sand et al., 2017; Shindell and Faluvegi, 2009). Therefore, future changes in non-local sources of aerosol and long-range transport could have significant impacts on aerosol concentrations in the Arctic. Furthermore, an increase in shipping traffic is expected once the Arctic becomes seasonally ice-free, further increasing aerosol concentrations (Peters et al., 2011). This increase in shipping traffic would also be expected to yield an increased demand for accurate weather forecasts over the Arctic region. However, it remains unclear whether the net effect of these changes in aerosol concentrations and surface fluxes would result in an increase or a decrease in cloud cover or drizzle precipitation. The changes in cloud properties could strongly influence the radiation budget in the Arctic, resulting in feedbacks on the rate of sea-ice loss. Arctic clouds remain poorly understood, and the current representation of these processes in global climate models is most likely insufficient to realistically simulate long-term changes.

Few observations have been made of Arctic clouds relative to clouds at lower latitudes. Field campaigns that have investigated Arctic clouds include the International Arctic Ocean Expeditions in 1991 (AOE-91; Leck et al., 1996) and 1996 (AOE-96; Leck et al., 2001), the Arctic Ocean Experiment in 2001 (AOE-01; Leck et al., 2004; Tjernström et al., 2004), the First IS-CCP (International Satellite Cloud Climatology Project) Regional Experiment Arctic Clouds Experiment in 1998 (FIRE-ACE; Curry et al., 2000), the Surface Heat Budget of the Arctic Ocean project in 1997-1998 (SHEBA; Uttal et al., 2002), the Mixed-Phase Arctic Cloud Experiment in 2004 (M-PACE; Verlinde et al., 2007), the Indirect and Semi-Direct Aerosol Campaign in 2008 (ISDAC; McFarquhar et al., 2011), the Arctic Summer Cloud Ocean Study in 2008 (ASCOS; Tjernström et al., 2014), the VERtical Distribution of Ice in Arctic cloud campaign in 2012 (VERDI; Klingebiel et al., 2015), the Aerosol-Cloud Coupling and Climate Interactions in the Arctic campaign in 2013 (ACCACIA; Lloyd et al., 2015; Young et al., 2016), The Arctic Clouds in Summer Experiment in 2014 (ACSE Tjernström et al., 2015), and the Canadian Network on Climate and Aerosols:

Addressing Key Uncertainties in Remote Canadian Environment campaign in 2014 (NETCARE; Leaitch et al., 2016). Of these campaigns, only a few (AOE-91, AOE-96, AOE-01, ASCOS, and ACSE) have sampled the high Arctic North of 80N. These campaigns and subsequent analyses have provided insights into the structures and radiative impacts of Arctic clouds, including the following:

1. At supersaturations as high as 0.8%, observed cloud condensation nuclei (CCN) concentrations are usually less than $100 \ \mathrm{cm}^{-3}$ in the high Arctic summer, and have been observed to be as low as $1 \ \mathrm{cm}^{-3}$ (Bigg et al., 1996; Bigg and Leck, 2001; Lannerfors et al., 1983; Leck et al., 2002; Leck and Svensson, 2015; Mauritsen et al., 2011). During the AOE-91, AOE-96, AOE-01, and ASCOS campaigns more than 25% of observed CCN concentrations were $<10 \ \mathrm{cm}^{-3}$ at supersaturations $\leq 0.3\%$. Additionally, more than 60% of the low-altitude clouds observed via aircraft during the NETCARE campaign were

found to have CCN concentrations less than $16 \ \mathrm{cm}^{-3}$ at a supersaturation of 0.6% (Leaitch et al., 2016).

2. Arctic clouds often have a net warming effect on the surface, even in summer (Intrieri et al., 2002). The shortwave (SW) radiative effect of Arctic clouds is small relative to the longwave (LW) radiative effect due to the high albedo of sea-ice and the low angle of incoming solar radiation.

3. The LW surface warming effect of Arctic clouds strongly affects the surface temperature, and therefore would be expected

to impact on the thickness and extent of Arctic sea-ice (Curry et al., 1993; Kapsch et al., 2016).

In order to better understand the processes controlling Arctic clouds and their uncertainties in current models, we perform a model intercomparison of summertime high Arctic (>80N) clouds. We have chosen as our case study the final two days of the ice drift period of the 2008 ASCOS campaign (Paatero et al., 2009; Tjernström et al., 2014). During this period, a decrease

in cloud water content was observed coincident with a decrease in observed CCN concentrations to less than $1 \ \mathrm{cm}^{-3}$. The concentrations of CCN were measured continuously using a CCN counter operating at a fixed supersaturation of ~0.2%. Details on the quality and data processing of ship-based CCN measurements are available in Martin et al. (2011) and in Leck and Svensson (2015). Previous analysis (Birch et al., 2012; Mauritsen et al., 2011) has identified these clouds as existing within the tenuous cloud regime: cloud liquid water content (LWC) and surface radiative effects are limited by the availability of

aerosol to act as CCN. This cloud regime has been observed during the ASCOS campaign (Mauritsen et al., 2011) and the NETCARE campaign (Leaitch et al., 2016). Due to the low CCN concentrations observed in the high Arctic, this cloud regime is expected to be a frequent occurrence in the Arctic summer. Sedlar et al. (2011) has linked the dissipation of these clouds and the associated increase in surface LW cooling to the onset of the autumn sea-ice freeze-up in 2008. The tenuous cloud regime would be very sensitive to changes in aerosol concentrations due to increased emissions from either increased human activity

in the Arctic, or increased emissions due to decreasing sea ice. Changes in these clouds would be expected to affect the surface radiative energy balance, and thereby potentially affect Arctic sea-ice extent and thickness. The tenuous cloud regime therefore presents an important, but challenging case to represent within models.

The ASCOS ice drift period, in whole or in part, has been previously examined using models by Birch et al. (2012), Wesslén

et al. (2014), Sotiropoulou et al. (2015), Hines and Bromwich (2017), Loewe et al. (2017), and Igel et al. (2017). The models

used by these studies were a single-column model configuration of the Met Office Unified Model (UM); two versions of the Arctic System Reanalysis (ASR) and the ERA-Interim reanalysis; the Integrated Forecast System (IFS) model of the European Centre for Medium-Range Weather Forecasts (ECMWF); the polar-optimized version of the Weather Research and Forecasting (WRF) regional numerical weather prediction (NWP) model; the Consortium for Small-scale Modeling (COSMO) model con-
figured as a Large-Eddy Simulation (LES) model; and the MISU MIT Cloud and Aerosol (MIMICA) LES model, respectively. Birch et al. (2012) found that observations of surface radiative fluxes and surface temperatures were better reproduced by the single-column UM during the tenuous cloud regime period on Sep. 1$^{st}$, 2008, if prescribed CCN concentrations were reduced to 1 cm$^{-3}$. For higher CCN concentrations, the model produced cloud with much larger LWCs than observed. Wesslén et al. (2014) highlighted that the two configurations of ASR failed to reproduce the observed clouds from Aug. 27$^{th}$ to Sep. 1$^{st}$. They
noted that this period was better represented by ERA-Interim, and they hypothesized that this was due to differences in the treatment of cloud microphysics. Sotiropoulou et al. (2015) found that, while using a constant assumed CCN concentration, increased model vertical resolution and a newer cloud microphysics scheme including prognostic cloud ice, rain and snow were insufficient to reproduce cloud dissipation during the tenuous cloud periods. Similarly to Birch et al. (2012), Hines and Bromwich (2017) found that biases of the Polar WRF regional NWP model against surface radiative flux observations for the
entire ASCOS drift period were reduced as the prescribed cloud droplet number concentration (CDNC) was reduced from values representative of low latitudes (250 cm$^{-3}$) to values representative of pristine Arctic conditions (10 cm$^{-3}$). Biases during the periods labelled as in the tenuous cloud regime were further reduced if the prescribed CDNC was reduced to 1 cm$^{-3}$. Loewe et al. (2017) found that in the LES configuration of COSMO, a prescribed CDNC of 2 cm$^{-3}$ was insufficient to prevent cloud dissipation, but that a cloud could be maintained with a prescribed CDNC of 10 cm$^{-3}$. They additionally performed
sensitivity studies to moisture availability and to ice crystal number concentrations (ICNC). The cloud LWC was found to be sensitive to both moisture availability and ICNC, but none of the tested water vapour profiles resulted in cloud dissipation, and an unrealistically high ICNC was required for cloud glaciation. Using the MIMICA LES model, Igel et al. (2017) found that enhanced levels of accumulation mode particles, if located at the cloud top, may under certain conditions be an important source of accumulation mode particles in the Arctic boundary layer.

Previous model studies of other Arctic mixed-phase clouds have established the sensitivity of cloud LWC, ice water contents (IWC), and other cloud properties in models to the interaction of ice and liquid (Klein et al., 2009), representation of ice enhancement mechanisms (Fan et al., 2009), prescribed cloud ICNC (Morrison et al., 2003, 2011; Ovchinnikov et al., 2011, 2014; Prenni et al., 2007; Solomon et al., 2009), ice-nucleating particle (INP) concentrations (Avramov and Harrington, 2010;
Harrington et al., 1999; Jiang et al., 2000; Morrison et al., 2005b; Pinto, 1998; Possner et al., 2017; Prenni et al., 2007; Young et al., 2017), INP depletion and supply (Fridlind et al., 2012; Morrison et al., 2005b; Paukert and Hoose, 2014; Possner et al., 2017; Prenni et al., 2007; Solomon et al., 2015), the size distribution of cloud ice (Ovchinnikov et al., 2014), the habit of cloud ice (Avramov and Harrington, 2010; Fridlind et al., 2012), and enhancement of CCN concentrations by ship emissions (Possner et al., 2017). Additionally, Furtado and Field (2017) have investigated the importance of riming in mixed-phase clouds.
However, the clouds investigated in these studies had greater CDNCs, and would not be expected to show the same sensitivity

to changes in CCN concentrations as the tenuous cloud regime observed during ASCOS.

In this paper, we extend these previous studies by comparing the results of both LES and cloud-resolving NWP models of the tenuous cloud regime observed during ASCOS using increasingly complex representations of aerosol-cloud interactions. We begin with simulations of liquid-phase cloud only, and we later show results where ice nucleation is included through prescribed ICNCs. We show first the results of simulations where cloud droplet activation is represented using prescribed CDNCs, similar to the studies of Birch et al. (2012), Loewe et al. (2017), and Hines and Bromwich (2017). We then show the results of simulations with cloud droplet activation calculated based on a temporally and spatially constant aerosol size distribution. Finally, we include in our simulations prognostic aerosol concentrations, including aerosol uptake and removal by activation into cloud droplets, which reduces the available CCN for activation in subsequent model time steps. In this way, we attempt to determine the key processes contributing to dissipation of these clouds, and we isolate and attempt to attribute differences in model results to differences in model processes. We then discuss the implications for realistic representation of Arctic aerosol-cloud interactions.

Section 2 shows an overview of observed meteorological conditions during the case study period. Section 3 describes the models participating in this study. Section 4 describes the simulations performed for this study. Section 5 presents and discusses the results of our liquid-phase only simulations, and Sect. 6 presents and discusses the results of the simulations including cloud ice. Finally, Sect. 7 offers a summary and our conclusions.

## 2 Overview of the ASCOS campaign

A full description of the conditions during the ASCOS campaign is available in Tjernström et al. (2012). Observations during the ASCOS campaign were obtained on-board the icebreaker Oden, from two measurement sites set up on the ice floe, and by helicopter. However, helicopter observations were restricted to outside of clouds due to safety concerns regarding icing of the aircraft. In order to examine the tenuous cloud regime, we focus our study on the period from Aug. 30th to Sep. 1st, 2008. These were the last two days of the ice drift period, which ended at about 87°09 N 11°01 W. Observed winds were westerly at the site, with observed wind speeds varying between 2 and 6 $\mathrm{m\,s^{-1}}$ during the two-day period. Conditions were dominated by a high-pressure system over the North Pole, yielding anti-cyclonic winds on the synoptic scale. Observed surface pressures rose from ~1025 to ~1030 $\mathrm{hPa}$ during the two-day period. Mixed-phase stratocumulus clouds were observed during this period until approximately 2000 UTC on Aug. 31st, when a break in low-level cloud cover was observed, despite observed water vapour mixing ratios at or above saturation, coincident with a decrease in observed CCN concentrations from about 70 $\mathrm{cm^{-3}}$ to <1 $\mathrm{cm^{-3}}$ (Mauritsen et al., 2011). The CCN concentrations were measured continuously using a CCN counter operating at a fixed supersaturation of ~0.2%. A second identical CCN counter was cycled between supersaturations of 0.11 and 0.73%. Martin et al. (2011) give further details on the quality of the data. Near-surface air temperatures were observed to be near -4 °C,

falling to -13 °C after the break in cloud.

Fig. 1 shows cloud properties, surface radiation, and aerosol concentrations derived from observations. Net surface LW radiation is defined to be positive downwards (absorption by the surface) throughout this paper. The LWC and IWC were derived from measurements using microwave radiometer, 35-GHz millimetre cloud radar, vertical temperature profiles from radiosondes, and ceilometers, as detailed in Shupe et al. (2013). The methodology is described further in Shupe et al. (2015). The observed liquid water path (LWP) has a reported root-mean-square error of 25 $\mathrm{g\,m^{-2}}$ (Westwater et al., 2001) and the uncertainty in the observed ice water path (IWP) could be up to a factor of two (Birch et al., 2012).

For ease of comparison with the model results, we designate the period from 2100 UTC on Aug. 30[th] to 1200 UTC on Aug. 31[st] as the "cloudy" period, and the period from 0000 UTC to 0600 UTC on Sep. 1[st] as the "nearly-cloud-free" period. There is a clear transition in every variable shown in Fig. 1 between these two periods: The liquid and frozen parts of the cloud both descend towards the surface, and the liquid and ice water contents both decrease, causing an increase in the LW emission from the surface. These changes are coincident with a decrease in the observed surface concentrations of aerosol particles larger than 50 nm (N50) from >10 $\mathrm{cm^{-3}}$ to <1 $\mathrm{cm^{-3}}$. Total aerosol concentrations as measured by a twin differential mobility particle sizer with a lower detection limit of 3 nm fell generally below 10 $\mathrm{cm^{-3}}$, with a median value of 2 $\mathrm{cm^{-3}}$ during the "nearly-cloud-free" period. Further details on the quality and data processing of ship-based aerosol measurements are available in Heintzenberg and Leck (2012). CCN concentrations measured at supersaturations as high as 0.73% during this period were also below 1 $\mathrm{cm^{-3}}$. Additionally, helicopter profiles of aerosol number concentrations were performed from 19:53 UTC to 20:13 UTC on Aug. 31[st] and from 07:32 UTC to 07:55 UTC on Sep. 1[st] using a condensation particle counter (Kupiszewski et al., 2013). These indicate that the number concentrations of aerosol larger than 14 nm were generally below 10 $\mathrm{cm^{-3}}$ up to 850 m altitude during this Aug. 31[st] profile and up to 500 m altitude during this Sep. 1[st] profile. With reference to Fig. 1, we note that these heights are similar to the locations of the observed cloud top heights at these time periods, and these altitudes were also similar to temperature inversion base heights observed via a scanning microwave radiometer (Kupiszewski et al., 2013).

In-cloud measurements were not performed due to aircraft icing concerns (Tjernström et al., 2014). Additionally, Cloud-Sat+Cloud–Aerosol Lidar with Orthogonal Polarization (CALIOP) cloud retrievals are not available north of 82N, and are therefore unavailable for this case (Kay and Gettelman, 2009). Moderate Resolution Imaging Spectroradiometer (MODIS) retrievals have been shown to underestimate cloud cover in the Arctic, particularly over sea ice and for cloud top heights less than 2 km (Chan and Comiso, 2013). We therefore consider MODIS-derived cloud information unreliable for this case. Therefore, no reliable observations of cloud droplet number concentrations are available for this case.

As mentioned above, previous analysis (Birch et al., 2012; Mauritsen et al., 2011) has identified these clouds as existing within the tenuous cloud regime: cloud LWC is limited by the availability of aerosol to act as CCN. The hypothesis is that

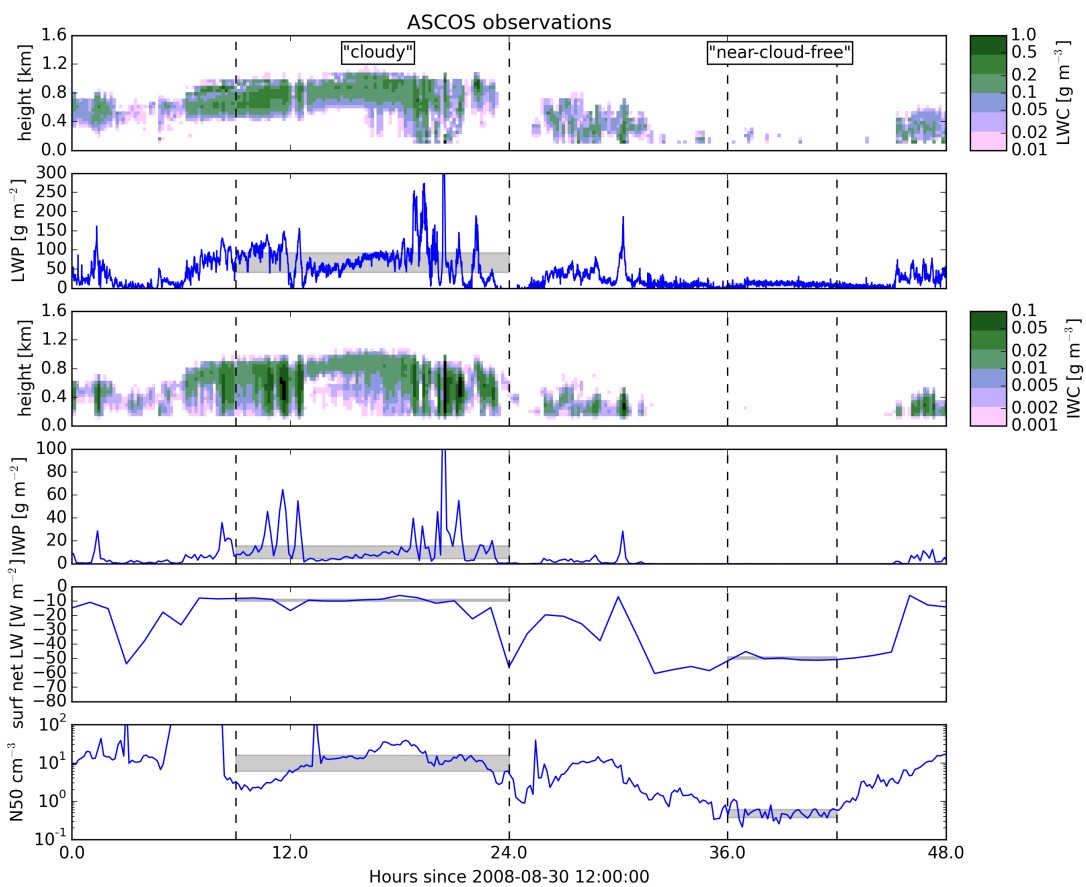

**Figure 1.** Observed cloud properties, surface radiation, and aerosol concentrations. Top row: liquid water content, second row: liquid water path, third row: ice water content, fourth row: ice water path, fifth row: surface net longwave flux, bottom row: concentrations of N50. Shaded rectangles indicate the interquartile ranges of LWP, IWP, surface net LW radiation and N50 during the "cloudy" and "nearly-cloud-free" periods, defined in Sect. 2. Dashed vertical lines indicate the beginnings and endings of these periods.

at extremely low CCN concentrations, each available CCN is activated, grows through condensation to drizzle droplet sizes, and is removed by sedimentation. It is implicit in this hypothesis that in-cloud precipitation occurs predominantly through liquid-phase processes, although frozen-phase processes could contribute to precipitation formation, and glaciation would be an alternate cause of cloud dissipation. In the following sections the aerosol and meteorological environment will be decoupled

5   via sensitivity tests to assess the validity of this hypothesis.

## 3 Description of participating models

Simulations were performed using three Large Eddy Simulation (LES) models and three Numerical Weather Prediction (NWP) models. LES models are fine resolution models (horizontally several metres to hundreds of metres) with domains typically from hundreds of metres to hundreds of kilometres capable of resolving turbulent eddies and useful for detailed studies of clouds.

NWP models are generally coarser-resolution (horizontally hundreds of metres to tens of kilometres) models with larger domains (tens of kilometres to global) capable of simulating mesoscale weather systems and performing operational forecasting. The NWP models used in this study all prognose surface temperatures and surface sensible and latent heat fluxes, but these values are prescribed for the LES models in this study. The NWP models can describe the full meteorological variability, and can therefore help to separate meteorological versus aerosol effects.

The LES models participating in this study are the University of California, Los Angeles LES with Sectional Aerosol module for Large Scale Applications (UCLALES-SALSA; Tonttila et al., 2017), the MISU MIT Cloud and Aerosol LES model (MIM-ICA; Savre et al., 2014) and the Consortium for Small-scale Modeling (COSMO) model configured as an LES model (Loewe et al., 2017) (hereafter referred to as COSMO-LES). The NWP models are v3.6.1 of the Polar Weather Research and Fore-

casting model (Polar WRF; Hines et al., 2015), the Met Office Unified Model with Cloud AeroSol Interacting Microphysics (UM-CASIM; Grosvenor et al., 2017), and COSMO configured as an NWP model (Steppeler et al., 2003) (hereafter referred to as COSMO-NWP). Each of the models is described in detail in previous publications, and so we will restrict ourselves to a brief overview here. The participating models are described and compared in Table 1.

UCLALES-SALSA is combination of an LES model (UCLALES; Stevens et al., 1999, 2005) and a sectional aerosol and cloud microphysics module (SALSA; Kokkola et al., 2008). A detailed description of UCLALES-SALSA can be found in Tonttila et al. (2017). A comparison of UCLALES-SALSA results against those of a previous model intercomparison based on the second Dynamics and Chemistry of Marine Stratocumulus Field Study (DYCOMSII) can also be found in Tonttila et al. (2017). The properties and microphysical processes of aerosol, cloud droplets, and rain are defined for certain size sections

(bins). In the current set-up, aerosol has 10 size bins based on dry particle size and cloud droplets have 7 bins that are parallel with the 7 largest aerosol bins. Rain drops have 7 size bins which are based on droplet size. Microphysics includes water vapour condensation and evaporation, cloud activation, rain formation, coagulation and deposition. With the exception of rain formation, these processes are modelled based on physical equations. Rain formation is based on an autoconversion scheme where a log-normal size distribution ($\sigma$=1.1) is expected for each cloud bin and droplets larger than 50 μm are moved to the

first precipitation bin. Subgrid-scale turbulence is based on the Smagorinsky-Lilly model as described in Seifert et al. (2010). Radiation transfer is calculated following the four-stream radiative transfer solver of Fu and Liou (1993).

MIMICA is an LES model which uses a two-moment bulk microphysics scheme with five hydrometeor categories (cloud droplets, rain drops, ice crystals, graupel and snow). MIMICA also includes a two-moment aerosol module providing the pos-

**Table 1.** Description of models participating in this study.

| | UCLALES-SALSA | MIMICA | COSMO-LES | COSMO-NWP | WRF | UM-CASIM |
|---|---|---|---|---|---|---|
| described in | Tonttila et al. (2017) | Savre et al. (2014) | Loewe et al. (2017), Seifert and Beheng (2006), Vogel et al. (2009) | Steppeler et al. (2003), Seifert and Beheng (2006), Vogel et al. (2009) | Hines et al. (2015) | Grosvenor et al. (2017) |
| condition for ice nucleation | N/A | $S_i > 0.05$ and $q_c > 0.001\,\mathrm{g\,kg^{-1}}$ | $S_i > 0.05$ and $q_c > 0.001\,\mathrm{g\,kg^{-1}}$ | $S_i > 0.05$ and $q_c > 0.001\,\mathrm{g\,kg^{-1}}$ | $S_l > -0.001$ and $T < -8^\circ\mathrm{C}$ | $S_l > -0.001$ and $T < -8^\circ\mathrm{C}$ |
| number of vertical levels below 2 km | 112 | 128 | 124 | 17 | 25 | 24 |
| finest vertical resolution [m] | 15.0 | 7.5 | 7.5 | 24.2 | 30.2 | 10.8 |
| coarsest vertical resolution below 2 km [m] | 47.2 | 47.7 | 228.3 | 237.1 | 141.9 | 156.7 |
| coarsest vertical resolution below 1.5 km [m] | 23.8 | 35.6 | 35.6 | 202.3 | 108.8 | 136.7 |
| horizontal resolution | 50 m | 62.5 m | 100 m | 1 km | 1 km | 1 km |
| horizontal domain size | 3.15 km | 6 km | 6.4 km | 600 km | 600 km | 600 km |
| prognostic aerosol[*] | Sectional aerosol (10 size bins; dry diameter from 3 nm to 1 μm) | Two-moment bulk (Igel et al., 2017) | None | None | None | Two-moment bulk |

[*]Only used in CCN30prog and CCN80prog simulations, described in Sect. 4.

sibility to represent different aerosol populations covering a range of size intervals and compositions (Ekman et al., 2006). The autoconversion parameterisation and the interactions between liquid particles follow the scheme of Seifert and Beheng (2006). Liquid-ice interactions are parameterised according to the microphysical scheme of Wang and Chang (1993). The subgrid-scale model is based on a Smagorinsky–Lilly eddy diffusivity closure (Lilly, 1992). At the surface, the model uses Monin-Obukhov similarity theory and the momentum fluxes are computed as described in Garratt (1994). The CCN activation is described by

the 'kappa-Köhler' theory (Petters and Kreidenweis, 2007). A four-stream radiative transfer solver (Fu and Liou, 1993) is used in the model. A thorough description of MIMICA is given in Savre et al. (2014). The MIMICA model has participated in the ISDAC model intercomparison study (Ovchinnikov et al., 2014), and has also been used to simulate the DYCOMSII case (Savre et al., 2014), and in both cases it compared well with other models.

Both COSMO-LES and COSMO-NWP use the two-moment cloud microphysics scheme described in Seifert and Beheng (2006). A fixed lognormal aerosol mode was implemented into COSMO-LES and prognostic aerosol transport, activation, and re-suspension following hydrometeor evaporation was implemented in COSMO-NWP following Possner et al. (2017). Aerosol activation to cloud droplets is performed following the scheme described in Nenes and Seinfeld (2003) and Fountoukis and
Nenes (2005). The two-stream radiation scheme after Ritter and Geleyn (1992) calculates the radiation transfer in COSMO. The boundary layer turbulence is parameterised using a 3-dimensional scheme in COSMO-LES (Herzog et al., 2002a, b) and a 1-dimensional vertical turbulent diffusion scheme based on Mellor and Yamada (1974) in COSMO-NWP. The minimum threshold for the eddy diffusivity in COSMO-NWP was adjusted to $0.01\ \mathrm{m^2\,s^{-1}}$ (Possner et al., 2014). The COSMO model participated in the ISDAC LES model intercomparison study (Ovchinnikov et al., 2014), and the predicted IWP and LWP were
within the range of the other models.

The physics options used in the Polar WRF simulations are based on the recommendations described in Hines et al. (2015). Cloud microphysical processes are parameterised according to the double-moment scheme of Morrison et al. (2005a). Auto-conversion of cloud droplets to rain is treated according to the scheme of Seifert and Beheng (2006). For droplet activation
in CCN30fixed and CCN80fixed cases (see Sect. 4), the scheme of Abdul-Razzak and Ghan (2000) is used assuming a fixed background concentration of CCN. There is no prognostic treatment of aerosols in the WRF simulations. The atmospheric boundary layer is represented by the Mellor-Yamada-Nakanishi-Niino (MYNN) scheme (Nakanishi and Niino, 2006), and the Rapid Radiative Transfer Model (RRTMG; Clough et al., 2005) is used for both longwave and shortwave radiation.

The UM-CASIM model has been described previously in Grosvenor et al. (2017) and Miltenberger et al. (2018). However, the sub-grid cloud scheme described in Grosvenor et al. (2017) was not used for this study. Boundary layer processes, including surface fluxes of moisture and heat, are parameterised with the blended boundary layer scheme (Lock et al., 2000, 2015) and sub-grid scale turbulent processes are represented with a 3D Smagorinsky-type turbulence scheme (Halliwell, 2014; Stratton et al., 2015). A two-stream radiation scheme is used, as described in Manners et al. (2016). It is possible to run the UM-CASIM
model as a fully-coupled atmosphere ocean model, but for this study a fixed sea ice fraction of 100% and a fixed sea ice thickness of 2 m were used. Activation of cloud droplets in simulations without prescribed CDNCs is performed following the scheme described in Abdul-Razzak et al. (1998) and Abdul-Razzak and Ghan (2000).

Excepting UCLALES-SALSA and WRF, all models in this study contained five hydrometeor classes: cloud droplets, rain,
cloud ice crystals, snow, and graupel. These hydrometeor classes are represented as gamma distributions with prescribed shape

parameters and prognosed bulk mass and number concentrations. WRF contains the hydrometeor classes described above except graupel. UCLALES-SALSA represents cloud droplets and rain drops using seven sectional size bins for each species tracking number and mass independently. Frozen water species are not currently simulated by UCLALES-SALSA. Sedimentation of cloud droplets is simulated only by UCLALES-SALSA, WRF and UM-CASIM.

Nucleation of cloud ice was conditionally permitted in each model within a defined range of temperatures ($T$), cloud droplet mass mixing ratios ($q_c$), liquid supersaturations ($S_l$), and ice supersaturations ($S_i$). In MIMICA and the two COSMO models, ice forms in the presence of supercooled liquid water ($S_i > 0.05$ and $q_c > 0.002$ or $0.001\,\mathrm{g\,kg^{-1}}$, respectively) and for WRF and UM-CASIM ice forms at $T < $ -8 °C in the presence of supercooled liquid water. These differences will have minimal impact

on the simulation, as cloud-top temperatures are generally below -8 °C.

For all models and all simulations, the rate of ice nucleation was parameterised following Fridlind et al. (2012) and Morrison et al. (2011). The change in ICNC due to nucleation of cloud ice in each timestep was therefore:

$$\Delta\mathrm{ICNC} = \max(0, \mathrm{ICNC}_{fixed} - \mathrm{ICNC}) \tag{1}$$

where ICNC is the cloud ice crystal number concentration, $\Delta$ICNC is the change in ICNC due to ice nucleation during a single model timestep, and $\mathrm{ICNC}_{fixed}$ is a chosen fixed value dependent on the experiment: $1\,\mathrm{L^{-1}}$, $0.2\,\mathrm{L^{-1}}$, or $0.02\,\mathrm{L^{-1}}$ for experiments labelled ICNC1p00, ICNC0p20, or ICNC0p02, respectively (see Sect. 4). Thus, whenever the conditions for ice formation are met, any loss in $N_{ice}$ due to sedimentation, autoconversion to snow, or scavenging will be exactly compensated by further activation to maintain the ICNC as $\mathrm{ICNC}_{fixed}$. For simulations labelled NOICE, the models were run without any

formation of frozen cloud water permitted.

## 4 Description of Simulations

For the UM-CASIM simulations, a global simulation initialized using the European Centre for Medium-Range Weather Forecasts (ECMWF) global analysis was performed to produce a set of time-varying boundary conditions. The WRF and COSMO-

NWP models used boundary conditions directly from the ECMWF global analysis. The three NWP models were then run with a $0.009°$ x $0.009°$ horizontal resolution rotated grid (approximately 1x1 km throughout the domain) spanning a 600 km x 600 km domain, centred at 87.3°N, 6.0°W. The period of interest for this study is the transition period of the observed cloud from the "cloudy" state to the "nearly-cloud-free" state, starting approximately at 1200 UTC on Aug. 31st (see Sect. 2 above). The NWP models were therefore started at 1200 UTC, Aug. 30th, 2008, to allow for 24 hours to reach a representative state,

and the total simulation duration was 48 hours.

Initial profiles of potential temperature, humidity, and wind speed for the LES models were taken from the Aug. 31st 0535 UTC radiosonde observations from the ASCOS campaign (Fig. 2). No flux of heat and moisture from or to the sur-

face was permitted due to the sea-ice cover. Sensible and latent heat fluxes at the surface were <1 W m$^{-2}$ in the UM-CASIM modelling results, and observed surface fluxes were generally <5 W m$^{-2}$ during the ASCOS campaign (Tjernström et al., 2012; Sedlar et al., 2011). Surface temperatures were prescribed to be -1.8 °C. Furthermore, the setup of all LES models follows the large scale subsidence description of Ovchinnikov et al. (2014), with divergence assumed to be constant below a height of

2 km. The value of the divergence was chosen to be $1.5 \times 10^{-6}$ s$^{-1}$. Preliminary simulations with UCLALES-SALSA showed that a divergence of $1.5 \times 10^{-6}$ s$^{-1}$ was too low in this model to balance radiative cooling and the associated mixing, and the cloud layer would continuously rise at a rate similar to the clouds in the COSMO-LES CDNC30 simulations (e.g. Fig. 3). The increased length of the UCLALES-SALSA simulations, compared to the COSMO-LES simulations (discussed next paragraph), allows the cloud layer to rise to unrealistic altitudes. A larger value of $5 \times 10^{-6}$ s$^{-1}$ was therefore used instead for

the subsidence in the UCLALES-SALSA simulations. While we do not investigate sensitivities to prescribed subsidence in this study, other studies have shown that differences in prescribed subsidence affect Arctic mixed-phase cloud LWP and IWP (Young et al., 2018). Within UCLALES-SALSA, subsidence only affects the tendencies of temperature and water vapour, and does not directly alter advection of air parcels, aerosols, cloud droplets or rain.

Due to numerical instabilities, the COSMO-LES simulations are restricted to a duration of 16 hours, including 2 hours of spin-up during which ice formation is not permitted. These instabilities are visible in the full model results as waves in the upper atmosphere. These waves do not reach the boundary layer during the simulations, and thus they don't influence the cloud in the boundary layer. In order to focus on the transition period starting approximately at 1200 UTC on Aug. 31[st], the COSMO-LES simulations were therefore started at 0600 UTC, Aug. 31[st]. UCLALES-SALSA simulations were run from 0000 UTC,

Aug. 31[st] for 36 hours, including three hours of spin-up, during which coagulation, sedimentation and autoconversion are disabled. MIMICA simulations were run from 1200 UTC, Aug. 30[th] for 72 hours, including two hours of spin-up, but we only show results from the first 48 hours in this study. As we have not prescribed any time-varying surface fluxes or large-scale forcings for the LES models, and the diurnal cycles in this case are weak, the LES model results are largely independent of the start time for this case.

Several sensitivity experiments with different treatments and concentrations of CCN and ICNC were carried out (Table 2). The values chosen for the sensitivity studies were based on observations of aerosol concentrations during the ASCOS campaign. First, to make the models as similar as possible, we performed simulations with prescribed CDNCs. We first prescribed a CDNC of 30 cm$^{-3}$ (CDNC30), as mean CCN concentrations at a supersaturation of 0.2% were observed to be 26.55 cm$^{-3}$

over the ice drift period (Martin et al., 2011). Then, in order to test the sensitivity to reduced aerosol concentrations, we perform simulations with the CDNC reduced to 3 cm$^{-3}$ (CDNC03).

We then performed simulations where cloud droplet activation was calculated based on an aerosol size distribution. We represented the aerosol size distribution using the lognormal fit of Igel et al. (2017). A single lognormal mode was fit to ob-

servations of accumulation-mode particles made on-board the icebreaker Oden using a twin differential mobility particle sizer

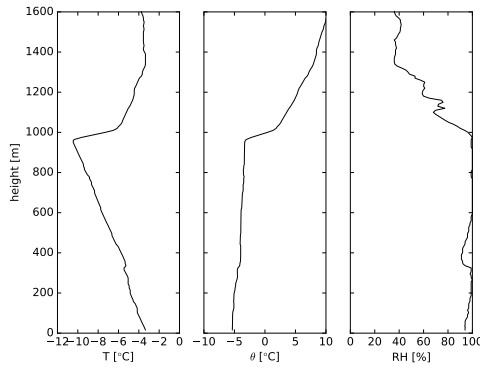

**Figure 2.** Aug. 31st 0535 UTC radiosonde observations of (left) absolute temperature, (centre) potential temperature and (right) relative humidity from the ASCOS campaign.

with an inlet height around 20-25 m above the surface (Leck et al., 2001). Further details on the quality and data processing of ship-based aerosol measurements are available in Heintzenberg and Leck (2012). This yielded a median diameter of 94 nm and a geometric standard deviation of 1.5. For simplicity, we assume that the aerosol particles are composed entirely of ammonium sulphate, but in reality 43% of the non-refractory aerosol mass was observed to be organic (Chang et al., 2011) with low hygro-

scopicity (Leck and Svensson, 2015). We initially chose an aerosol number concentration of 30 $cm^{-3}$ (CCN30) to represent the "cloudy" period based on the observed CCN concentrations. However, preliminary simulations with UCLALES-SALSA indicated that an initial CCN concentration of 30 $cm^{-3}$ would result in dissipation of the cloud (as will be shown in Sect. 5.4), so a larger value of 80 $cm^{-3}$ (CCN80) was chosen as a sensitivity study. Additionally, we chose a value of 3 $cm^{-3}$ to test the sensitivity of our results to further reductions in the CCN concentration. In order to assess the sensitivity to the removal of

aerosol by cloud processes within the models, we perform simulations with either constant aerosol or with prognostic aerosol processing. In the CCN30fixed and CCN80fixed cases, the aerosol concentration remains constant in space and time, and is not affected by cloud processes. Cloud droplet activation occurs only if the number of newly-activated cloud droplets exceeds the current number of cloud droplets in a given grid cell, in which case the CDNC is updated to the number calculated by the activation parameterisation. In the prognostic aerosol simulations (CCN03prog, CCN30prog and CCN80prog), aerosol is

removed through activation into cloud droplets, re-suspended upon evaporation, and transported by advection.

In addition to the sensitivity to CCN, we also investigated the sensitivity of the clouds within the models to ICNC. Observations of ice-nucleating particles (INP) are not available for this period, as the concentrations at the surface were below the detection limit of the instrument (Loewe et al., 2017). Following Loewe et al. (2017), we chose a prescribed ICNC of 0.2 $L^{-1}$

as our control simulation (ICNC0p20), based on previous observations of INP in the Arctic from AOE-91 and AOE-96 (Bigg, 1996; Bigg and Leck, 2001). Additionally, we performed a liquid-phase only sensitivity study with no ice nucleation (NOICE),

**Table 2.** Description of simulations performed. The last six columns indicate which models performed simulations of each case. UCL=UCLALES-SALSA, MIM=MIMICA, COL=COSMO-LES, CON=COSMO-NWP, UMC=UM-CASIM

| Name | initial CCN [cm$^{-3}$] | prognostic aerosol | CDNC [cm$^{-3}$] | ICNC [L$^{-1}$] | UCL | MIM | COL | CON | WRF | UMC |
|---|---|---|---|---|---|---|---|---|---|---|
| CDNC30_NOICE | None | N/A | 30 | 0.00 | | ✓ | ✓ | ✓ | ✓ | ✓ |
| CDNC03_NOICE | None | N/A | 3 | 0.00 | | ✓ | ✓ | ✓ | ✓ | ✓ |
| CDNC30_ICNC0p02 | None | N/A | 30 | 0.02 | | ✓ | ✓ | | ✓ | ✓ |
| CDNC03_ICNC0p02 | None | N/A | 3 | 0.02 | | ✓ | ✓ | | ✓ | ✓ |
| CDNC30_ICNC0p20 | None | N/A | 30 | 0.20 | | ✓ | ✓ | ✓ | ✓ | ✓ |
| CDNC03_ICNC0p20 | None | N/A | 3 | 0.20 | | ✓ | ✓ | ✓ | ✓ | ✓ |
| CDNC30_ICNC1p00 | None | N/A | 30 | 1.00 | | ✓ | ✓ | ✓ | ✓ | ✓ |
| CDNC03_ICNC1p00 | None | N/A | 3 | 1.00 | | ✓ | ✓ | ✓ | ✓ | ✓ |
| CCN30fixed_NOICE | 30 | no | prognostic | 0.00 | | ✓ | ✓ | ✓ | ✓ | ✓ |
| CCN80fixed_NOICE | 80 | no | prognostic | 0.00 | | ✓ | ✓ | ✓ | ✓ | ✓ |
| CCN30fixed_ICNC0p02 | 30 | no | prognostic | 0.02 | | ✓ | ✓ | | ✓ | ✓ |
| CCN80fixed_ICNC0p02 | 80 | no | prognostic | 0.02 | | ✓ | ✓ | | ✓ | ✓ |
| CCN30fixed_ICNC0p20 | 30 | no | prognostic | 0.20 | | ✓ | ✓ | ✓ | ✓ | ✓ |
| CCN80fixed_ICNC0p20 | 80 | no | prognostic | 0.20 | | ✓ | ✓ | ✓ | ✓ | ✓ |
| CCN03prog_NOICE | 3 | yes | prognostic | 0.00 | ✓ | ✓ | | ✓ | | ✓ |
| CCN30prog_NOICE | 30 | yes | prognostic | 0.00 | ✓ | ✓ | | ✓ | | ✓ |
| CCN80prog_NOICE | 80 | yes | prognostic | 0.00 | ✓ | ✓ | | ✓ | | ✓ |
| CCN03prog_ICNC0p02 | 3 | yes | prognostic | 0.02 | | ✓ | | | | ✓ |
| CCN30prog_ICNC0p02 | 30 | yes | prognostic | 0.02 | | ✓ | | | | ✓ |
| CCN80prog_ICNC0p02 | 80 | yes | prognostic | 0.02 | | ✓ | | | | ✓ |
| CCN03prog_ICNC0p20 | 3 | yes | prognostic | 0.20 | | ✓ | | ✓ | | ✓ |
| CCN30prog_ICNC0p20 | 30 | yes | prognostic | 0.20 | | ✓ | | ✓ | | ✓ |
| CCN80prog_ICNC0p20 | 80 | yes | prognostic | 0.20 | | ✓ | | ✓ | | ✓ |

and additional sensitivity studies with prescribed ICNCs of 0.02 L$^{-1}$ (ICNC0p02) and 1 L$^{-1}$ (ICNC1p00).

## 5   Liquid-phase only simulations

### 5.1   Base case: CDNC 30 cm$^{-3}$

5   We begin by discussing the CDNC30_NOICE case. Figure 3 shows the LWCs and the mass mixing ratios of cloud droplets and rain predicted by the MIMICA, COSMO-LES, COSMO-NWP, WRF, and UM-CASIM models. Results in this figure and

throughout the paper are shown at the centre of the domain for all models. We note that the direct comparison of results between LES and NWP models is not trivial: the LES models in this study used wrapped boundary conditions and time-invariant surface fluxes, and therefore would always be expected to tend towards some equilibrium cloud state. The NWP models, however, simulate the advection of different air masses with different histories through the domain, and changes due to differences in air masses can be conflated with the temporal evolution of a single cloud system. With these challenges in mind, we note that the surface is homogeneously covered in sea-ice in all models, and we expect that the centre of the domain will be representative for our case study. In order to assess this, we show statistics from the NWP models over a 100 $km^2$ area in the centre of the domain in the supplement material (Fig. S1-S3). Fig. S1 shows characteristics of the distribution of LWP and IWP within the specified 100 $km^2$ area as simulated by the three NWP models for the CDNC30_ICNC0p20 case. Figure S2 and Fig. S3 show statistics of LWP, IWP, and net surface longwave radiation for the NWP models for all of the sensitivity studies. We note that the centre-of-domain values are nearly always within the interquartile range of the 100 $km^2$ area values. Furthermore, centre-of-domain values are sufficiently close to the domain medians, and have similar enough responses to changes in CDNC, CCN, and ICNC, as not to change the conclusions of our study. We expect less spatial variability in the LES models than the NWP models, which were run with periodic boundary conditions and fixed surface fluxes. Thus, the centre-of-domain points are representative for the domain in both NWP and LES models.

All models produce clouds near 1 km altitude. Despite no inclusion of ice processes, the predicted LWC values are generally within a factor of two of those observed during the "cloudy" period. In all models, the cloud droplet mass mixing ratios generally increase with altitude within the cloud. The MIMICA model predicts the thickest cloud (cloud depth ~600 m) with the largest cloud droplet mass mixing ratios, reaching values greater than 0.5 $g\,kg^{-1}$ at cloud top. The cloud depths simulated by WRF and UM-CASIM are slightly thinner (~500 m), and the cloud droplet mass mixing ratios are smaller (~0.3 $g\,kg^{-1}$). The cloud depths produced by COSMO-NWP are similar to those produced by WRF and UM-CASIM, but the cloud droplet mass mixing ratios are much smaller (~0.05 $g\,kg^{-1}$). The cloud-top height predicted by COSMO-NWP is greater than for any other model. This is consistent for all cases in this study simulated by COSMO-NWP. We note that COSMO-NWP has the coarsest vertical resolution of all the models participating in this study. The COSMO-LES model produces the thinnest clouds (cloud depth ~400 m) with the lowest cloud droplet mass mixing ratios (<0.2 $g\,kg^{-1}$). COSMO-LES produces a consistent layer of rain below cloud with mass mixing ratios ~0.04 $g\,kg^{-1}$. The other four models, however, produce less rain with more variability.

None of the models predict the observed dissolution of the cloud during the second half of the examined period, except perhaps UM-CASIM. We will show in Sect. 6.1 that this is generally true even if cloud ice is included in the models. UM-CASIM predicts thinning of the cloud during the last six hours of simulation, suggesting a possible meteorological contribution to dissipation, but the other two NWP models do not predict this thinning. Previous analysis of this case has identified these clouds as existing within the tenuous cloud regime, and has suggested that the dissipation of the cloud is related to extremely low (<1 $cm^{-3}$) observed CCN concentrations. The prescribed CDNC cases would not be expected to reproduce this effect, as

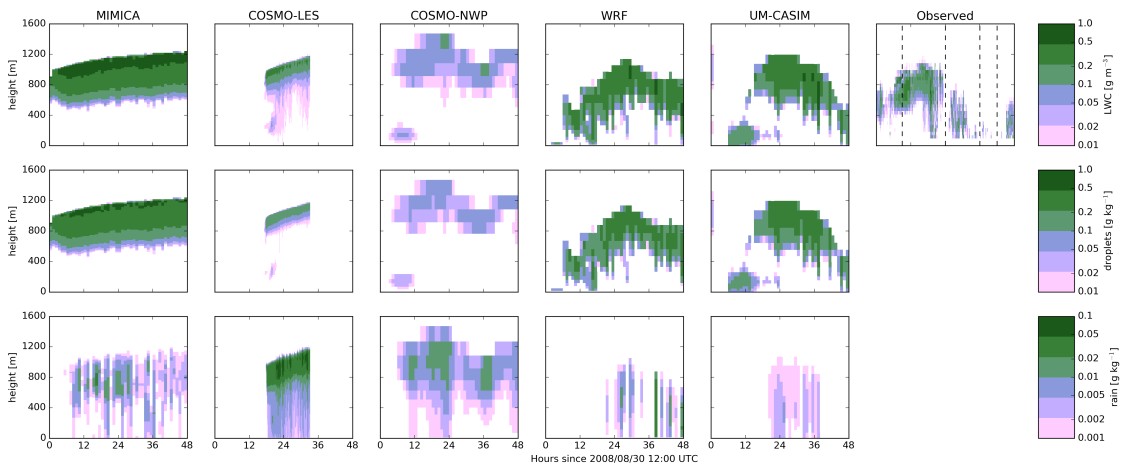

**Figure 3.** Liquid water content and cloud droplet and rain mass mixing ratios in the simulations with a prescribed CDNC of 30 $\mathrm{cm}^{-3}$ and no cloud ice permitted (CDNC30_NOICE), and liquid water content derived from observations. Top row: Liquid water content. Middle row: mass mixing ratios of cloud droplets. Bottom row: mass mixing ratios of rain. Results are shown from the (from left to right) MIMICA, COSMO-LES, COSMO-NWP, WRF, and UM-CASIM models. Observed liquid water contents are shown in the rightmost column. Vertical dashed lines indicate the beginnings and endings of the "cloudy" and "nearly-cloud-free" periods.

the parameterisation of the cloud droplet activation is not linked to CCN availability. However, other potential causes of the transition could be resolved by the models. In particular, the NWP models would be expected to yield more realistic changes in meteorological conditions due to advective transport, through changes with time in the boundary conditions applied to these models. However, the vertical atmospheric structure at the interiors of the domains will evolve to be different than at the bound-

5    aries. Nevertheless, the absence of this transition in these modelling results supports the interpretation that the LWC of these clouds is CCN-limited. We will discuss this further in Sect. 5.2, when we discuss the lower prescribed CDNC case.

In order to explain the differences between the results of the different models, Fig. 4 shows the liquid-phase process rates for this simulation (autoconversion of cloud droplets to rain, sedimentation of cloud droplets, and sedimentation of rain). The

10   larger mass mixing ratios of rain and the thinner cloud predicted by COSMO-LES is due to the larger autoconversion tendencies ($>1 \times 10^{-4}\,\mathrm{g\,m^{-3}\,s^{-1}}$ vs. $10^{-6}$ to $10^{-5}\,\mathrm{g\,m^{-3}\,s^{-1}}$ in other models). Autoconversion rates greater than $2 \times 10^{-6}\,\mathrm{g\,m^{-3}\,s^{-1}}$ exist even in regions where the cloud droplet mass concentration is less than $0.01\,\mathrm{g\,cm^{-3}}$, the lower limit of the colour scale shown in Fig. 3. Autoconversion rates and cloud droplet mass mixing ratios both decrease by about two orders of magnitude from their maximums near cloud top to the layer between 200 and 700 m.

By dividing the mass of cloud droplets by the autoconversion rates from each model, an autoconversion timescale can be estimated for each model. This autoconversion timescale is less than one hour for COSMO-LES, on the order of several

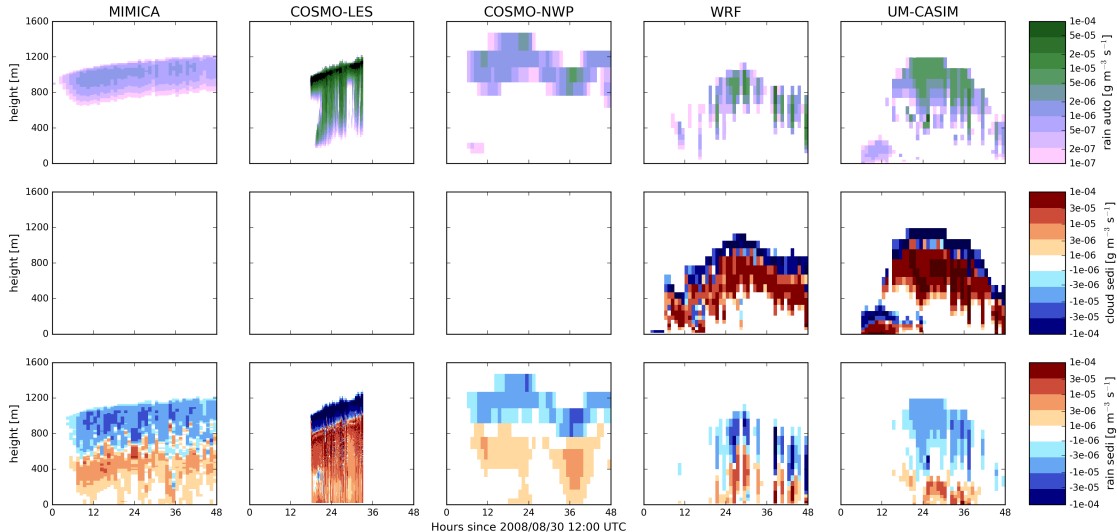

**Figure 4.** Tendencies of mixing ratios of cloud and rain water due to liquid-phase processes for simulations with a prescribed CDNC of $30\,\mathrm{cm^{-3}}$ and no cloud ice permitted (CDNC30_NOICE). Top row: autoconversion of cloud droplets to rain, middle row: sedimentation of cloud droplets, bottom row: sedimentation of rain drops. Results are shown from the (from left to right) MIMICA, COSMO-LES, COSMO-NWP, WRF, and UM-CASIM models. Note that only WRF and UM-CASIM simulate sedimentation of cloud droplets.

hours for COSMO-NWP, approximately one day for WRF and UM-CASIM, and several days for MIMICA for this case. The COSMO-LES model also has the greatest tendencies due to rain sedimentation ($>10^{-4}\,\mathrm{g\,m^{-3}\,s^{-1}}$). These large sedimentation tendencies are partially explained by the fact that COSMO-LES produces a greater mass of rain of all the models for this case. The higher cloud droplet mixing ratios seen in the MIMICA results are due to a combination of lower autoconversion

tendencies and a lack of cloud droplet sedimentation in this model.

Figure 5 shows scatter plots of the autoconversion tendencies plotted against the cloud droplet mass mixing ratios, in order to allow us to examine this process in more detail. The large differences in autoconversion tendencies are despite the fact that the same autoconversion scheme (Seifert and Beheng, 2006) is used in MIMICA, COSMO-LES, COSMO-NWP, and WRF.

COSMO-LES, COSMO-NWP, and WRF all prescribed the same maximum cloud droplet radius to be used for autoconversion (40 μm), and MIMICA used a similar value (50 μm). The initial difference in autoconversion tendencies between COSMO-LES and COSMO-NWP can be explained primarily by the difference in cloud droplet mass mixing ratios: as there is more mass of cloud droplets available to form rain in COSMO-LES, autoconversion tendencies are greater. The Seifert and Beheng (2006) autoconversion scheme also predicts greater autoconversion rates if rain constitutes a greater proportion of the liquid

water mass within a given grid cell. This results in a positive feedback on any other model differences that affect autoconversion rates, enhancing differences in autoconversion rates between COSMO-LES and COSMO-NWP. Autoconversion tendencies per

unit mass of cloud droplets are clearly greater in COSMO-LES and COSMO-NWP than in WRF, MIMICA, and UM-CASIM for the CDNC30_NOICE case. As cloud droplet activation is prescribed in this case, autoconversion is similarly treated in all models except for UM-CASIM, and no frozen processes are permitted in this case, we believe that the differences in autoconversion rates per unit cloud droplet mass are due primarily to the differences in the representation of the cloud droplet size distribution. MIMICA, COSMO-LES, and COSMO-NWP, and WRF represent the cloud droplet size distribution using a Gamma distribution defined by:

$$\frac{dN}{dx} = A \, x^{\nu_1} \, exp(-\lambda_1 \, x^{\mu}) \tag{2}$$

where $x$ is the cloud droplet mass, and $\mu$ and $\nu_1$ are shape parameters. The intercept and slope parameters $A$ and $\lambda_1$ are defined by:

$$A = \frac{\mu \, \text{CDNC}}{\Gamma(\frac{\nu_1+1}{\mu})} \lambda_1^{\frac{\nu_1+1}{\mu}} \tag{3}$$

$$\lambda_1 = [\frac{\Gamma(\frac{\nu_1+1}{\mu})}{\Gamma(\frac{\nu_1+2}{\mu})} \bar{x}]^{-\mu} \tag{4}$$

where $\bar{x}$ is the mean cloud droplet mass. However, the prescribed shape parameters are different between the different models: COSMO-LES and COSMO-NWP used shape parameters $\mu = 0.33$ and $\nu_1 = 0$, and MIMICA used $\mu = 0.33$ and $\nu_1 = 1$. In the WRF model, $\mu = 0.33$, and $\nu_1$ is diagnostically calculated based on Martin et al. (1994). UM-CASIM used a different form of the gamma distribution:

$$\frac{dN}{dD} = \text{CDNC}\frac{1}{\Gamma(1+\nu_2)}\lambda_2^{(1+\nu_2)} D^{\nu_2} exp(-\lambda_2 D) \tag{5}$$

where $D$ is the cloud droplet diameter, $\nu_2$ and $\lambda_2$ are shape and slope parameters, distinct in meaning from $\nu_1$ and $\lambda_1$. $\lambda_2$ is defined by:

$$\lambda_2 = [\frac{\Gamma(4+\nu_2)}{\Gamma(1+\nu_2)} \frac{\pi \, \rho}{6 \, \bar{x}}]^{1/3} \tag{6}$$

For the purposes of calculating autoconversion, UM-CASIM used a diagnostic $\nu_2$ based on Martin et al. (1994).

## 5.2 Sensitivity to prescribed CDNC

Next, we examine the CDNC03_NOICE case, in order to investigate the sensitivity of the model results to a reduction in prescribed CDNC from 30 to 3 $\text{cm}^{-3}$. Figure 6 shows the mass mixing ratios of cloud droplets and rain. All models produce thinner clouds with lower LWCs compared to the higher CDNC case. A stable cloud is produced by MIMICA, with cloud thickness reduced to ~300 m and cloud-top cloud droplet mass mixing ratios reduced to ~0.2 $\text{g\,kg}^{-1}$, but mixing ratios of rain are similar to those produced with the larger prescribed CDNC. In COSMO-LES, two clouds are produced initially, at

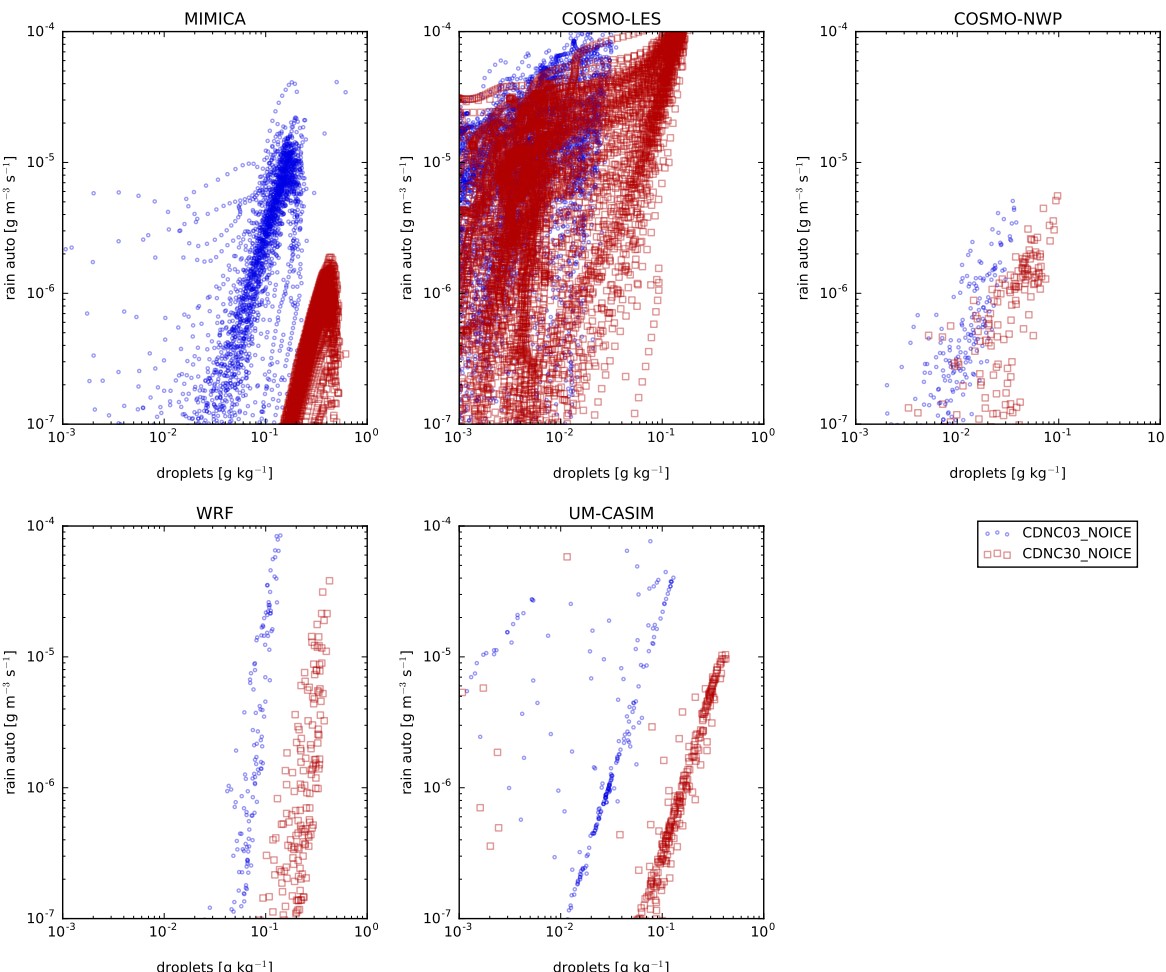

**Figure 5.** Tendencies of mixing ratios of cloud water due to autoconversion vs. cloud droplet mass mixing ratios for simulations with prescribed CDNCs and no cloud ice permitted (CDNC30_NOICE). Results are shown from the (top left) MIMICA, (top middle) COSMO-LES, (top right) COSMO-NWP, (bottom left) WRF, and (bottom right) UM-CASIM models. Results from simulations with a prescribed CDNC of 3 $cm^{-3}$ are shown as blue circle and those with a prescribed CDNC of 30 $cm^{-3}$ are shown as red squares.

~200 m and ~900 m. Available water is removed by precipitation, and the clouds begin to dissipate towards the end of the simulation (note that COSMO-LES simulations end at 34 hours since Aug. 30[th], 2008, 1200 UTC). COSMO-NWP produces a cloud with cloud droplet mass mixing ratios reduced to ~0.02 $g\,kg^{-1}$, that thins and temporarily dissipates towards the end of the simulation. UM-CASIM produces a stable cloud with cloud-top cloud droplet mass mixing ratios reduced to ~0.2 $g\,kg^{-1}$,

5    with rain mass mixing ratios larger than those predicted when using a prescribed CDNC of 30 $cm^{-3}$. WRF produces a fog layer between the surface and ~500 m. The reduced LWPs predicted by WRF early in the simulation, as compared to the CDNC30_NOICE case, allow greater longwave cooling of the surface, ultimately creating an inversion layer that tracks the

top of the fog layer. This effect would not be reproduced by COSMO-LES, despite the dissolution of the cloud, as the surface temperature in COSMO-LES was prescribed for this study.

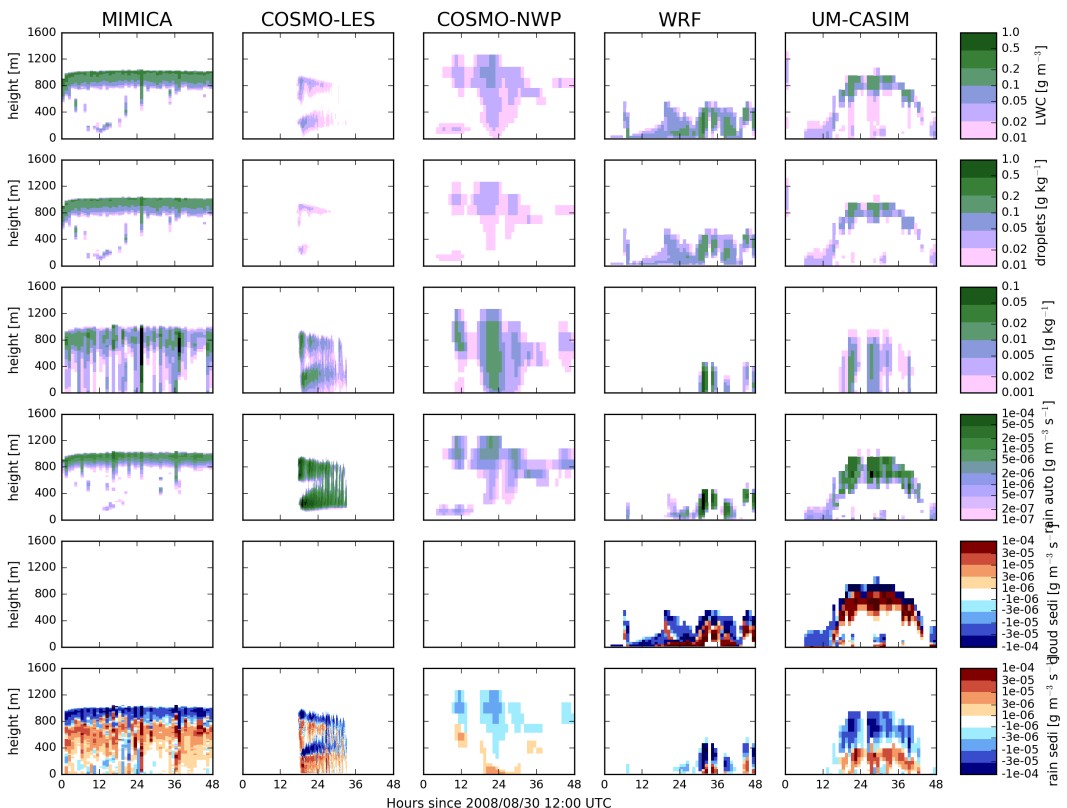

**Figure 6.** Liquid water contents, cloud and rain mass mixing ratios and tendencies due to liquid-phase processes for simulations with a prescribed CDNC of 3 cm$^{-3}$ and no cloud ice permitted (CDNC03_NOICE). Top row: liquid water contents. Second row: mass mixing ratios of cloud droplets. Third row: mass mixing ratios of rain. Lowest three rows: tendencies of mixing ratios of cloud droplets or rain due to (fourth row) autoconversion of cloud droplets to rain, (fifth row) sedimentation of cloud droplets, and (bottom row) sedimentation of rain. Results are shown from the (from left to right) MIMICA, COSMO-LES, COSMO-NWP, WRF, and UM-CASIM models. Note that only WRF and UM-CASIM simulate sedimentation of cloud droplets.

Figure 6 also shows the liquid-phase process rates for the CDNC03_NOICE case. The reduction in the prescribed CDNC values results in an increase in the autoconversion to rain tendencies in MIMICA and UM-CASIM. It can be seen in Fig. 5 that the autoconversion tendencies are increased even after accounting for changes in cloud droplet mass mixing ratios. Within WRF and UM-CASIM, cloud droplet sedimentation tendencies remain similar in magnitude to those in the CDNC30_NOICE

simulation. The mass of cloud droplets available to sediment is reduced by increased autoconversion to rain, but this is compensated by the increased fall speeds due to the increased size of the cloud droplets. Rain sedimentation tendencies in WRF and UM-CASIM are also similar in magnitude to the CDNC30_NOICE case. The rates of autoconversion to rain and sedimentation of rain predicted COSMO-NWP are similar to those in the CDNC30_NOICE case when the cloud thickness and LWC are greatest, but diminish to much smaller values as the cloud dissipates. Compared to the higher CDNC case, the MIMICA model predicts larger losses within the cloud through the sedimentation of rain, due to the larger mixing ratio of rain predicted in this case. The COSMO-LES model predicts lower mixing ratios of rain for this case relative to the CDNC30_NOICE case as the cloud dissipates. The changes in mass due to sedimentation are therefore lower than in the CDNC30_NOICE case.

## 5.3 Sensitivity to activation scheme

We will now discuss the CCN30fixed_NOICE and CCN80fixed_NOICE cases. These cases differ from the prescribed CDNC cases in that cloud droplet activation is predicted based on a constant background aerosol concentration of either $80 \text{ cm}^{-3}$ or $30 \text{ cm}^{-3}$ with median diameter 94 nm and geometric standard deviation of 1.5, instead of being prescribed to be $30 \text{ cm}^{-3}$ or $3 \text{ cm}^{-3}$.

Figure 7 shows time-averaged profiles of cloud properties for the CDNC30_NOICE, CDNC03_NOICE, CCN30fixed_NOICE, and CCN80fixed_NOICE cases. We average over the period from 1200 UTC to 2400 UTC on Aug. 31[st], (24-36 hours since 1200 UTC, Aug. 30[th]) in order to exclude the initial period before a stable cloud forms in the NWP models. We note that for the CDNC03 and CDNC30 cases in COSMO-LES, COSMO-NWP, and UM-CASIM, the CDNC is prescribed through activation, but is permitted to vary within cloud due to evaporation and transport. When the background CCN concentration is set to be $30 \text{ cm}^{-3}$, the CDNC within cloud (column a) is ~15 $\text{cm}^{-3}$ in MIMICA, ~25 $\text{cm}^{-3}$ in COSMO-LES, ~15 $\text{cm}^{-3}$ in COSMO-NWP, ~20 $\text{cm}^{-3}$ in WRF, and ~20 $\text{cm}^{-3}$ in UM-CASIM. The differences in activation fractions are more pronounced for the CCN80 cases: MIMICA, COSMO-LES, COSMO-NWP, WRF and UM-CASIM predict in-cloud CDNCs of ~25 $\text{cm}^{-3}$, ~60 $\text{cm}^{-3}$, ~20 $\text{cm}^{-3}$, ~40 $\text{cm}^{-3}$, and ~60 $\text{cm}^{-3}$, respectively. This diversity in CDNC of 15-20 $\text{cm}^{-3}$ or 20-60 $\text{cm}^{-3}$ for the same constant CCN concentrations underscores the variability that exists in model results and model sensitivities to perturbations in aerosol concentrations. Unless the models are constrained through common forcings and common scientific choices, there will remain diversity in model results and model sensitivity, for both LES and NWP models.

There are many model differences making it difficult to assign variations to particular processes, but one pair of models provides some insight as we shall see. These differences are due in part to differences in activation schemes used in the different models: the activation scheme described in Khvorostyanov and Curry (2006) is used in MIMICA, the scheme described in Nenes and Seinfeld (2003) and Fountoukis and Nenes (2005) is used in COSMO-LES and COSMO-NWP, and the scheme described in Abdul-Razzak et al. (1998) and Abdul-Razzak and Ghan (2000) is used in WRF and UM-CASIM. These differences may also be due to differences in the representation of small-scale turbulence within the models: COSMO-NWP, WRF and

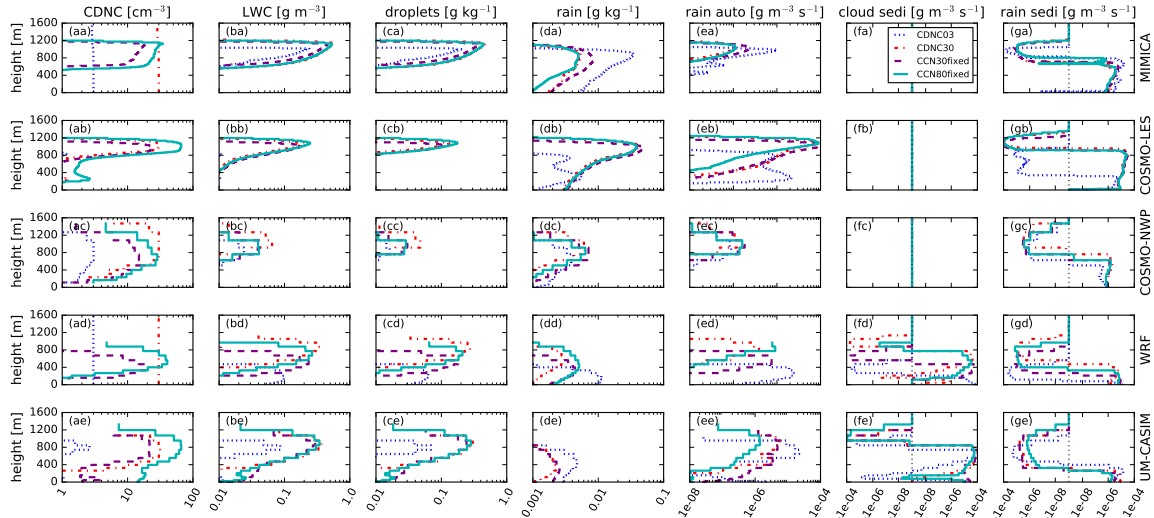

**Figure 7.** Time-averaged profiles of cloud properties and tendencies due to liquid-phase processes for simulations with no cloud ice permitted. First letter in subplot labels refers to column and second to row. Column a: cloud droplet number concentrations. Column b: liquid water contents. Column c: cloud droplet mass mixing ratios. Column d: rain mass mixing ratios. Rightmost three columns: tendencies of mixing ratios of cloud droplets or rain due to autoconversion of cloud droplets to rain (column e), sedimentation of cloud droplets (column f), and sedimentation of rain (column g). Results are shown from the (from top to bottom) MIMICA (row a), COSMO-LES (row b), COSMO-NWP (row c), WRF (row d), and UM-CASIM (row e) models. Blue dotted line indicates the CDNC03 case, red dash-dotted line indicates the CDNC30 case, the purple dashed line indicates the CCN30fixed case, and the solid turquoise line indicates the CCN80fixed case. Note that only WRF and UM-CASIM simulate sedimentation of cloud droplets.

UM-CASIM have horizontal resolutions too coarse to resolve individual updrafts. WRF and UM-CASIM therefore assume minimum updraft velocities for activation as $0.1 \mathrm{\ m\ s^{-1}}$. COSMO-NWP parameterises the updraft velocity used for activation by adding $0.8 \times \sqrt{TKE}$ to the grid-resolved updraft velocity, where TKE is the turbulent kinetic energy. The fine resolutions of MIMICA and COSMO-LES allow them to resolve these updrafts explicitly. Differences in sink terms across models, such

5 as collision-coalescence of cloud droplets and cloud droplet sedimentation, would also be expected to contribute to these differences. As WRF and UM-CASIM have the same activation scheme, and the same minimum updraft velocity, we infer that remaining differences in CDNC are due to differences in sink terms. For the CCN30fixed case, CDNCs are similar in both models, but CDNCs simulated by UM-CASIM are greater in the CCN80fixed case. Therefore, CDNC sinks must be similar in the CCN30fixed case, but faster for WRF in the CCN80fixed case.

As cloud properties within the tenuous cloud regime are expected to be dependent on CCN concentrations via changes in CDNC, it is informative to examine how cloud properties are related to the modelled CDNC for these four cases. With the exception of the CDNC03 case, the vertical cloud extent and cloud droplet mass mixing ratios (column c) are similar across

the different cases in MIMICA, COSMO-LES, and UM-CASIM (differences <100 m and <0.1 $\mathrm{g\,kg^{-1}}$, respectively). The COSMO-NWP model (subplot cc) shows higher cloud altitudes and cloud droplet mass mixing ratios for the CDNC30 case. The WRF model results (subplot cd) generally show an increase in both cloud vertical thickness and cloud height correlated with increasing CDNC. The mass mixing ratios of rain within MIMICA (subplot da) and UM-CASIM (subplot de) clearly increase with decreasing CDNCs due to increases in autoconversion from cloud droplets (subplots ea and ee), mitigated somewhat by increases in rain sedimentation rates (subplots ga and ge). For the CCN80fixed case, the CDNC is sufficiently high in UM-CASIM to reduce concentrations of rain below $10^{-3}$ $\mathrm{g\,kg^{-1}}$. This effect is present within WRF (subplot dd), but is more difficult to discern because of coincident changes in cloud height and thickness. Within COSMO-LES (subplot db), there is a weak increase in rain mass mixing ratios with decreasing CDNC, until CDNC is reduced to 3 $\mathrm{cm^{-3}}$, at which point rain mass mixing ratios are reduced due to cloud dissipation.

## 5.4 Sensitivity to prognostic aerosol

We consider now the CCN80prog_NOICE case. In these simulations, the aerosol is initialised as in the CCN80fixed_NOICE case, but is then allowed to evolve with time due to advection, removal by cloud droplet activation and re-suspension upon evaporation. Figure 8 shows profiles vs. time of the mass mixing ratios of cloud droplets and rain, CDNC, N50 concentrations, and potential temperature.

For this case, the MIMICA and COSMO-NWP models produce results very similar to those for the CCN80fixed_NOICE case. In COSMO-NWP, re-suspension of aerosol upon evaporation of cloud droplets and rain drops leads to a build-up of aerosol below cloud, leading to an enhancement of CDNCs at cloud base, particularly after 24 hours of simulation time. The UM-CASIM model produces a cloud that is reduced in vertical extent and liquid water content, with more rain compared to the case without aerosol processing. The reduction in available CCN by activation reduces CDNC, leading to larger cloud droplets and increased autoconversion to rain. The UCLALES-SALSA model also produces a stable cloud with cloud-top height near 1 km and cloud droplet mixing ratios of ~0.3 $\mathrm{g\,kg^{-1}}$, but with no autoconversion to rain. Unlike the other models included in this study, UCLALES-SALSA does not assume a gamma distribution for cloud droplets, and instead uses 7 sectional bins to represent the cloud droplet size distribution and explicitly calculates drop-drop collisions using the bin representation (see Sect. 3). Therefore, the UCLALES-SALSA model does not necessarily produce any large (>50 µm) cloud droplets upon activation, as would be implicitly assumed by a gamma distribution. The UCLALES-SALSA model resolves narrower cloud droplet size distributions than those represented by the other models in this study, with no cloud droplets large enough to trigger partitioning into the rain category. Differences in cloud thickness between MIMICA and UCLALES-SALSA (thickening in MIMICA and thinning with time in UCLALES-SALSA) for this case are primarily due to the different subsidence rates as described in Sect. 4. Simulations performed by UCLALES-SALSA using the same lower subsidence rate as the MIMICA simulations yielded a cloud layer with a similar LWP to the MIMICA simulation (~125 $\mathrm{g\,m^{-2}}$ and 140 $\mathrm{g\,m^{-2}}$, respectively),

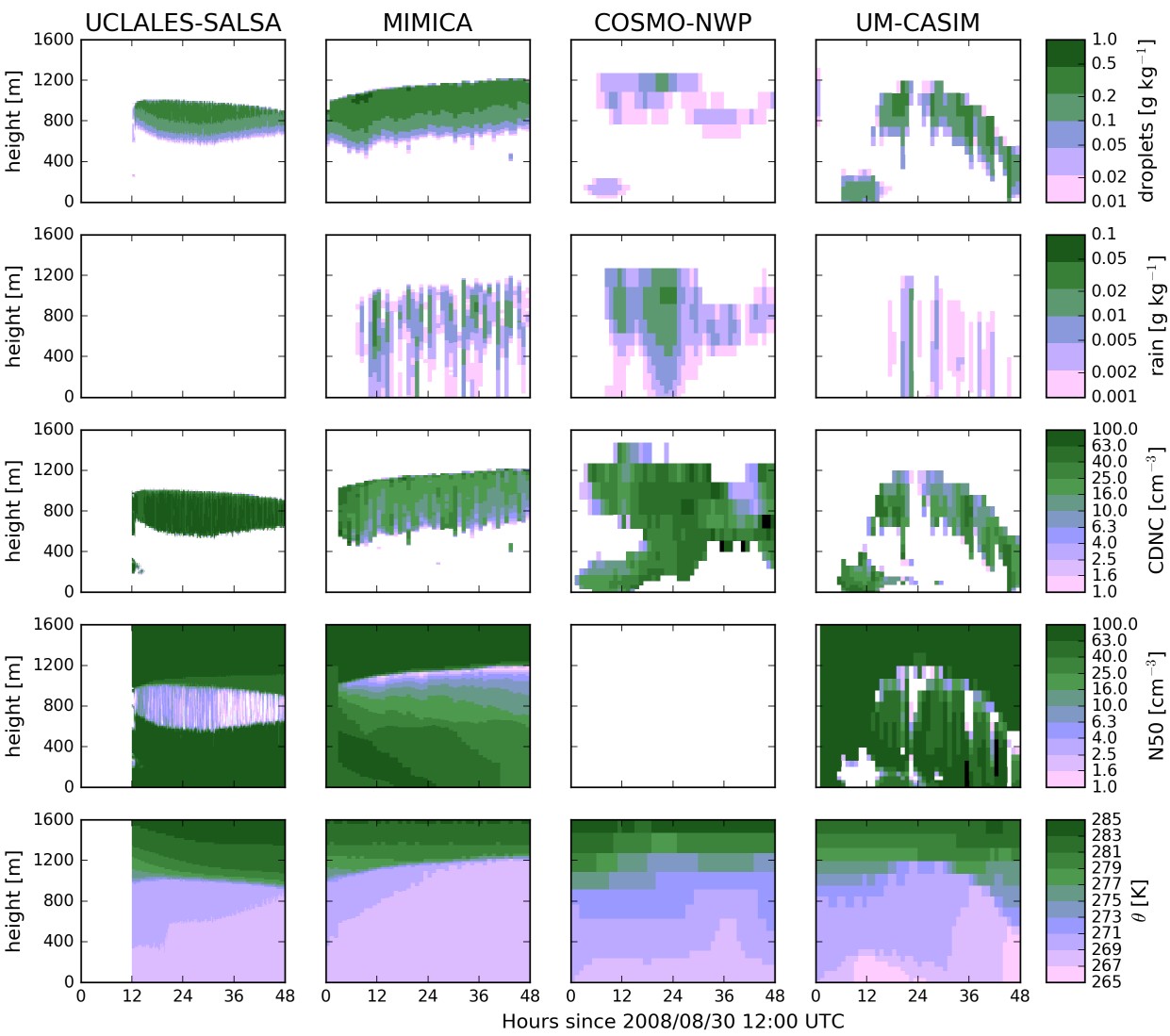

**Figure 8.** Cloud properties in the simulations with prognostic aerosol and an initial CCN concentration of 80 $cm^{-3}$ (CCN80prog_NOICE). Top row: cloud droplet mass mixing ratio, second row: rain mass mixing ratio, third row: cloud droplet number concentration, fourth row: N50 concentration, bottom row: potential temperature. Results are shown from the (from left to right) UCLALES-SALSA, MIMICA, COSMO-NWP and UM-CASIM models. Note that aerosol concentrations and CDNCs are fixed during the two-hour spin-up period in MIMICA, and N50 concentrations are not available for COSMO-NWP.

but the cloud layer rose at an unrealistic rate.

When the initial CCN concentration is reduced to 30 $cm^{-3}$ (CCN30prog_NOICE, Fig. 9), the UCLALES-SALSA model no longer maintains a stable cloud. Instead, the larger size of cloud droplets allows for partitioning into rain, which sub-
sequently removes the available aerosol by sedimentation. As the cloud thins, radiative cooling of the cloud top weakens, resulting in less generation of turbulence. The above-cloud temperature inversion subsequently descends due to subsidence. Within UCLALES-SALSA, subsidence only affects the tendencies of temperature and water vapour, and does not directly alter advection of aerosols. Therefore the temperature inversion descends into the aerosol-depleted layer, suppressing any further entrainment of aerosol from above the cloud. The reduction in aerosol concentrations further reduces CDNCs, leading to larger
cloud droplets and further enhances conversion to rain. The depletion of aerosol therefore results in a positive feedback loop that ends with total dissipation of the cloud. The MIMICA, COSMO-NWP and UM-CASIM models, conversely, do maintain clouds to the end of the simulation, although the water content of the clouds are reduced. The COSMO-NWP model shows the weakest sensitivity to the decrease in CCN concentrations, similarly to the weak sensitivity shown in Sect. 5.2 to changes in prescribed CDNC. The vertical extent of the cloud simulated by the MIMICA model decreases with time. This cloud has
similar cloud droplet mass mixing ratios to the case with fixed aerosol concentrations (CCN30fixed_NOICE), but is thinner (~300 m vs. ~500 m). The CDNC decreases during the simulation to ~2 $cm^{-3}$ after 48 hours, resulting in faster autoconversion rates and larger mixing ratios of rain. Results from the UM-CASIM model are qualitatively similar to those with the higher ini­tial aerosol concentration, but cloud droplet mass mixing ratios and CDNC are lower (0.1 vs. 0.15 $g\,kg^{-1}$ and 5 vs. 20 $cm^{-3}$). The concurrent reductions in both cloud droplet mass mixing ratios and CDNC yield only small changes in cloud droplet sizes,
and so there are no large changes in rain autoconversion rates, cloud droplet sedimentation rates, mass mixing ratios of rain, or rain sedimentation rates.

Further reducing the initial CCN concentration to 3 $cm^{-3}$ (CCN03prog_NOICE, Fig. 10) results in dissipation of the orig­inal cloud in UCLALES-SALSA, MIMICA, and UM-CASIM. The results of UCLALES-SALSA are qualitatively similar to
the results with an initial CCN concentration of 30 $cm^{-3}$, except that the cloud dissipates much more quickly. The cloud completely dissipates after less than six hours into the simulation, while in the simulation with an initial CCN concentration of 30 $cm^{-3}$, formation of rain started after six hours of simulation, and complete dissipation of the cloud did not occur until the end of the 36 hour simulation. The original cloud layer in the MIMICA model dissipates after about 36 hours of simulation time. A second cloud layer forms 12 hours from the beginning of the simulation at around 200 m from the surface and rises to
700 m by the end of the simulation. Rain falling from the upper cloud layer evaporates before reaching the lower cloud layer. This transports moisture and aerosol vertically closer to the lower cloud layer, where they are subsequently mixed into the lower cloud layer by turbulence. COSMO-NWP maintains a drizzling cloud throughout most of the simulation. Evaporation of cloud droplets and rain drops transports aerosol below cloud, resulting in larger aerosol concentrations and larger CDNCs at cloud base than those predicted by the other models. In UM-CASIM, reduction of the initial aerosol concentration to 3 $cm^{-3}$
results in dissipation of the cloud by drizzle. The formation and dissipation of the cloud is not visible in the centre-of-domain

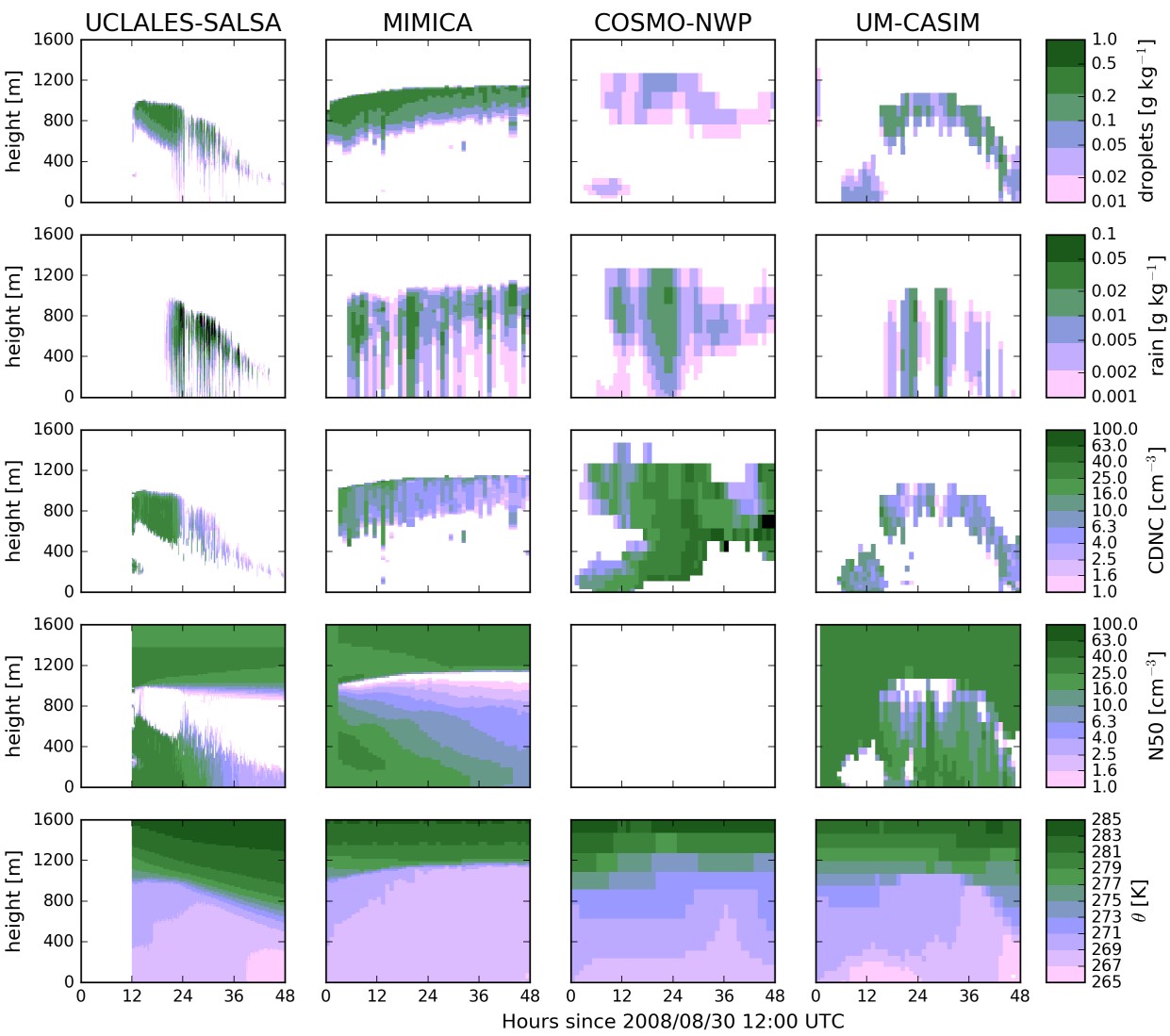

**Figure 9.** Cloud properties in the simulations with prognostic aerosol and an initial CCN concentration of 30 $cm^{-3}$ (CCN30prog_NOICE). Top row: cloud droplet mass mixing ratio, second row: rain mass mixing ratio, third row: cloud droplet number concentration, fourth row: N50 concentration, bottom row: potential temperature. Results are shown from the (from left to right) UCLALES-SALSA, MIMICA, COSMO-NWP and UM-CASIM models. Note that aerosol concentrations and CDNCs are fixed during the two-hour spin-up period in MIMICA, and N50 concentrations are not available for COSMO-NWP.

results shown here, but the aerosol number concentrations remain depleted in the air mass where the cloud formed, which passes through the centre of the domain from 24-36 hours from the start of the simulation. The thinning of the cloud layer allows cooling of the surface via longwave emission, creating a stable layer near 200 m. This restricts any cloud from forming above this layer. This feedback will not occur in the LES models due to prescribed surface conditions and fluxes used in our study.

The timescale of aerosol removal depends strongly on the model and the initial CCN concentration. UCLALES-SALSA predicts that below-cloud N50 concentrations would be unaffected for initial CCN concentrations of 3 or 80 $cm^{-3}$, due to a lack of mixing to the surface after cloud dissipation in the former case and a lack of precipitation in the latter case. If the initial CCN concentration is 30 $cm^{-3}$, UCLALES-SALSA predicts that N50 concentrations throughout the boundary layer fall below 1 $cm^{-3}$ after 36 hours. The MIMICA model predicts a steady decrease in surface N50 concentrations for all three prognostic cases simulated, ranging from ~0.4 $cm^{-3} h^{-1}$ for the 80 $cm^{-3}$ case to 0.05 $cm^{-3} h^{-1}$ for the 3 $cm^{-3}$ case. Aerosol removal rates are difficult to diagnose from COSMO-NWP and UM-CASIM due to advection of different air masses being simultaneous with aerosol processing.

The model results shown above demonstrate a robust relationship between decreases in CDNC, either through direct pre-scription or from the effects of activation and processing, and the thinning or even collapse of the cloud layer. However, the sensitivity of the cloud layer to decreases in CDNC differs between models due to differences in partitioning of cloud liquid between cloud droplets and rain, and differences in the representation of surface properties. In the next section we build on these liquid-only results by adding the complication of ice interactions.

## 6   Sensitivity to ice formation

### 6.1   Base case

Figure 11 shows the liquid and ice water contents from the models when the CDNC is prescribed as 30 $cm^{-3}$ and the ICNC is prescribed as 0.2 $L^{-1}$ (CDNC30_ICNC0p20). The IWCs predicted by the models vary by an order of magnitude between the models, with COSMO-LES and WRF predicting IWCs less than 0.002 $g m^{-3}$, but MIMICA producing highly-variable IWCs often as great as 0.02 $g m^{-3}$. We note that the IWCs derived from observations are often greater than 0.05 $g m^{-3}$, but the un-certainty could be as great as a factor of two, as stated in Sect. 2. Even accounting for this uncertainty, all models underestimate the IWC for this case. Any model bias in IWC does not seem to be related to biases in LWC.

Figure 12 shows the mass mixing ratios of cloud droplets, rain, cloud ice crystals, snow, and graupel from the models when the CDNC is prescribed as 30 $cm^{-3}$ and the ICNC is prescribed as 0.2 $L^{-1}$ (CDNC30_ICNC0p20). We note with comparison to Fig. 3 that the introduction of ice does not change cloud height or cloud depth by more than 100 m in any model, and cloud

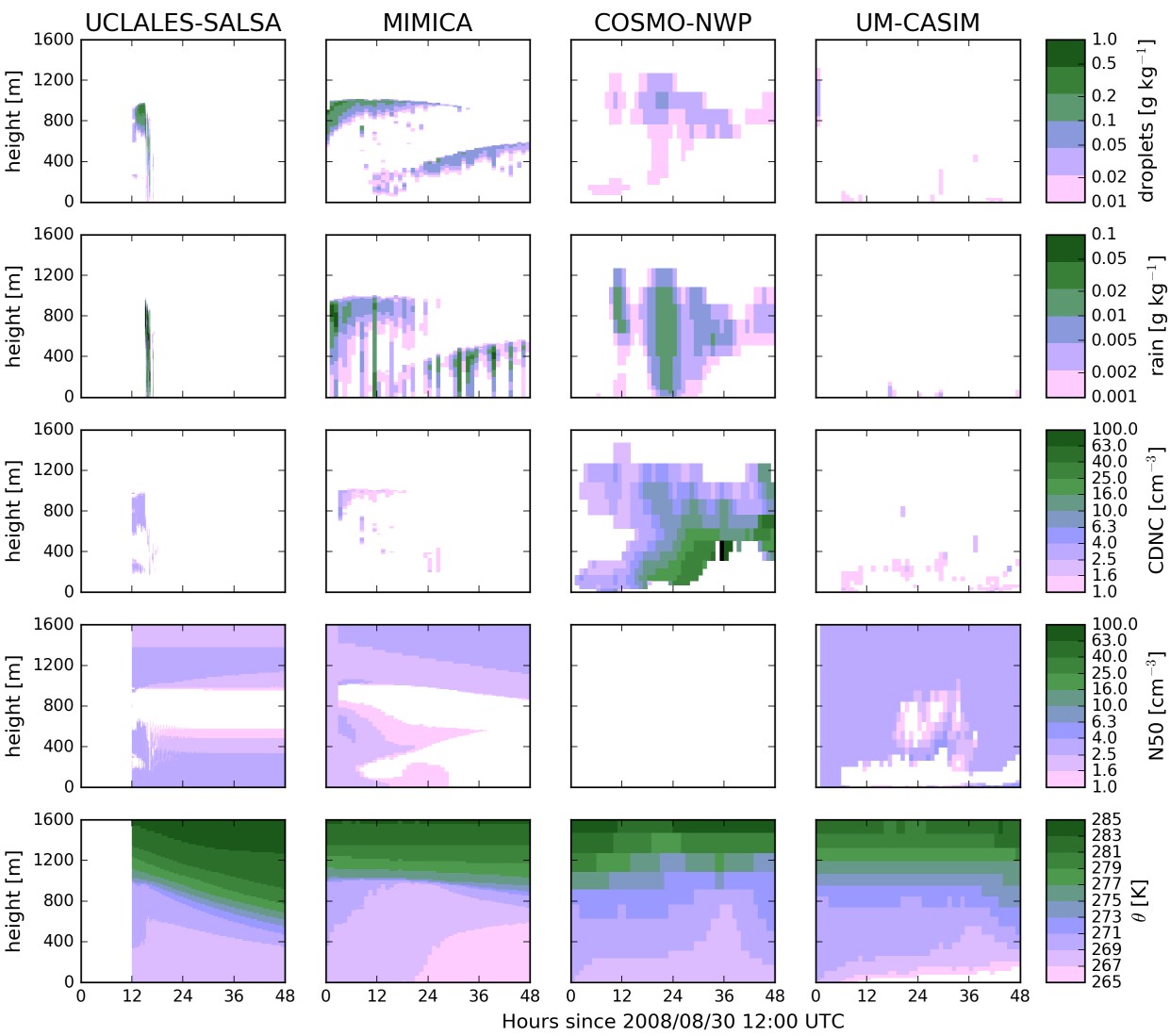

**Figure 10.** Cloud properties in the simulations with prognostic aerosol and an initial CCN concentration of 3 $\mathrm{cm^{-3}}$ (CCN03prog_NOICE). Top row: cloud droplet mass mixing ratio, second row: rain mass mixing ratio, third row: cloud droplet number concentration, fourth row: N50 concentration, bottom row: potential temperature. Results are shown from the (from left to right) UCLALES-SALSA, MIMICA, and UM-CASIM models. Note that aerosol concentrations and CDNCs are fixed during the two-hour spin-up period in MIMICA, and N50 concentrations are not available for COSMO-NWP.

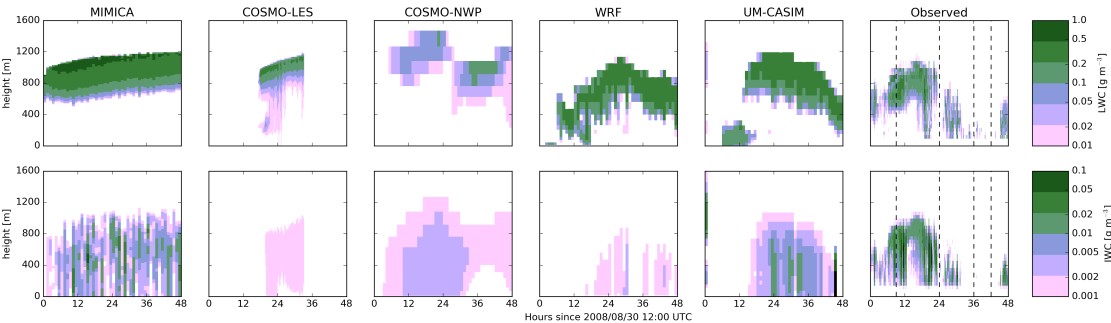

**Figure 11.** Liquid and ice water content in the simulations with a prescribed CDNC of 30 cm$^{-3}$ and a prescribed ICNC of 0.2 L$^{-1}$ (CDNC30_ICNC0p20) and derived from observations. Top row: liquid water contents, bottom row: ice water contents. Results are shown from the (left to right) MIMICA, COSMO-LES, COSMO-NWP, WRF, and UM-CASIM models. Values derived from observations are shown in the rightmost column. Vertical dashed lines indicate the beginnings and endings of the "cloudy" and "nearly-cloud-free" periods.

mass mixing ratios change by less than 20% in all models. However, mass mixing ratios of rain are reduced in the results of the MIMICA and WRF models.

The form of frozen mass depends on which model is used: only MIMICA produces a significant amount of graupel, and
only WRF predicts that most frozen water would be snow. COSMO-LES, COSMO-NWP, and UM-CASIM predict the frozen water to exist predominantly as cloud ice crystals, but UM-CASIM also predicts a small amount of mass in the snow category. Within MIMICA, any collision between a liquid hydrometeor and a frozen hydrometeor will move the resulting mass to the graupel category. Within all other models, collisions between cloud ice crystals smaller than 160 μm and cloud droplets do not form graupel. Collisions between ice crystals larger than 160 μm and cloud droplets can produce graupel in COSMO-LES
and COSMO-NWP, but the collision and sticking efficiencies are small. So even if large ice crystals are present, this remains a negligible source of graupel. Since cloud ice crystals are the dominant form of frozen hydrometeors in all other models aside from WRF, and cloud droplets are the dominant form of liquid hydrometeors in all models, no graupel is formed in COSMO-LES, COSMO-NWP, or UM-CASIM. As mentioned in Sect. 3, the setup of WRF used in this study does not possess a graupel category, so riming by snow will increase the mass of snow instead of forming graupel in WRF.

In order to examine the causes and implications of these differences in ice between the models, Fig. 13 shows time-averaged profiles of process rates affecting ice crystals and snow for each of the models for the prescribed CDNC cases. We average over Aug. 31$^{st}$, 2008, from 1200 UTC to 2400 UTC in order to exclude the initial period of the NWP models before a stable cloud forms. We note that mass mixing ratios of snow (column b) are often an order of magnitude less than cloud ice mass mixing
ratios (column a), but even these small amounts of snow can have significant effects on cloud species or water vapour mixing ratios (columns f and h). Within COSMO-LES, excepting the CDNC30_ICNC1p00 case, insignificant autoconversion to snow

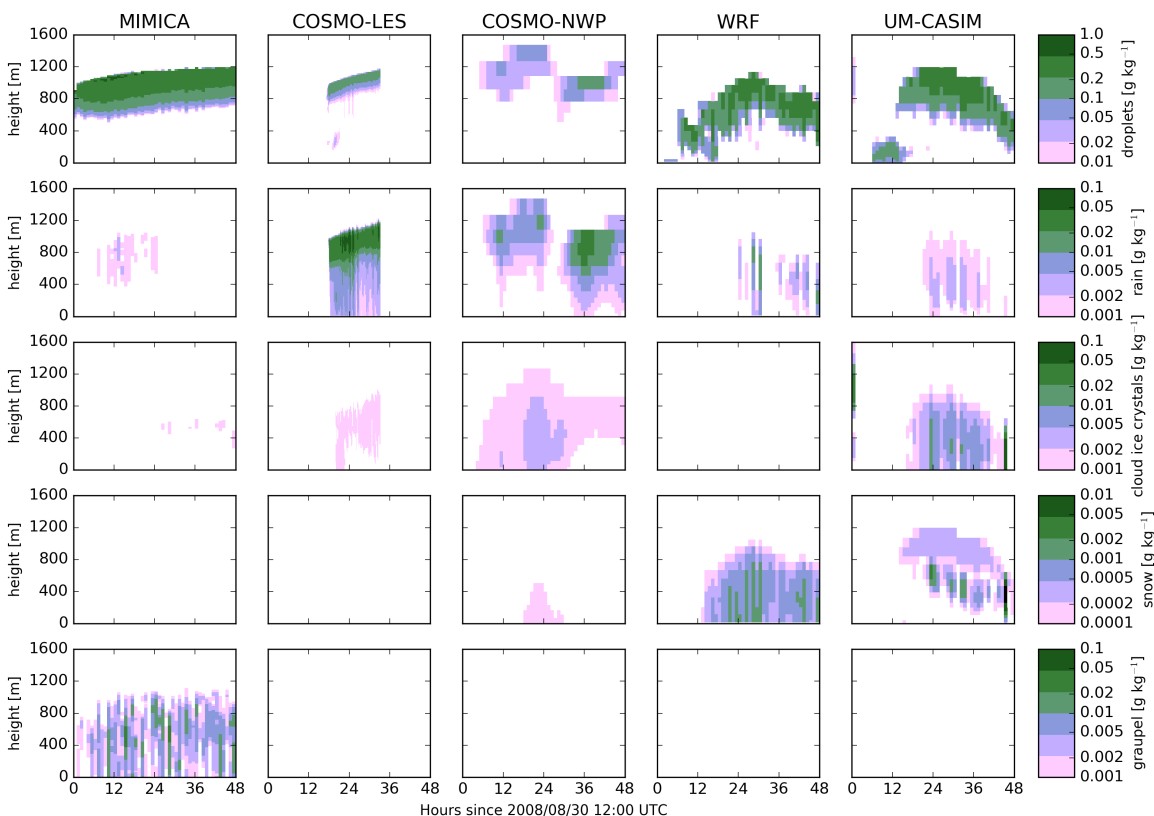

**Figure 12.** Cloud mass mixing ratios in the simulations with a prescribed CDNC of 30 $cm^{-3}$ and a prescribed ICNC of 0.2 $L^{-1}$ (CDNC30_ICNC0p20). Top row: mass mixing ratios of cloud droplets, second row: mass mixing ratios of rain, third row: mass mixing ratios of cloud ice crystals, fourth row: mass mixing ratios of snow, bottom row: mass mixing ratios of graupel. Results are shown from the (left to right) MIMICA, COSMO-LES, COSMO-NWP, WRF, and UM-CASIM models. Note that WRF does not possess a graupel category.

occurs (subplot cb) and nearly all frozen cloud mass remains as cloud ice crystals (subplot ab). The cloud ice grows by deposition within cloud and sublimates below cloud (subplot gb), frequently sublimating completely before reaching the surface. COSMO-NWP (row c) behaves similarly to COSMO-LES, but the cloud ice grows by deposition throughout the boundary layer (subplot gc). As stated previously, only WRF maintains significant mixing ratios of snow (subplot bd). Autoconversion to snow proceeds more quickly than in the other models for the same cloud ice crystal mixing ratios (compare subplots cd and ad). The snow that is produced through autoconversion subsequently grows efficiently by riming of cloud droplets (subplot ed) and deposition of water vapour (subplot hd). UM-CASIM simulates the greatest autoconversion rates of all the models (subplot ce). This is in part due to UM-CASIM producing the greatest cloud ice crystal mixing ratios of all the models (subplot ae), but

autoconversion proceeds more quickly even for similar cloud ice mixing ratios. The snow produced by UM-CASIM grows efficiently by deposition and collection of cloud water (subplots he and ee), but also sediments more quickly per unit mass than in any other model (subplot fe), and sublimates quickly below cloud (subplot he), and thus the mass of snow maintained in the atmosphere is small.

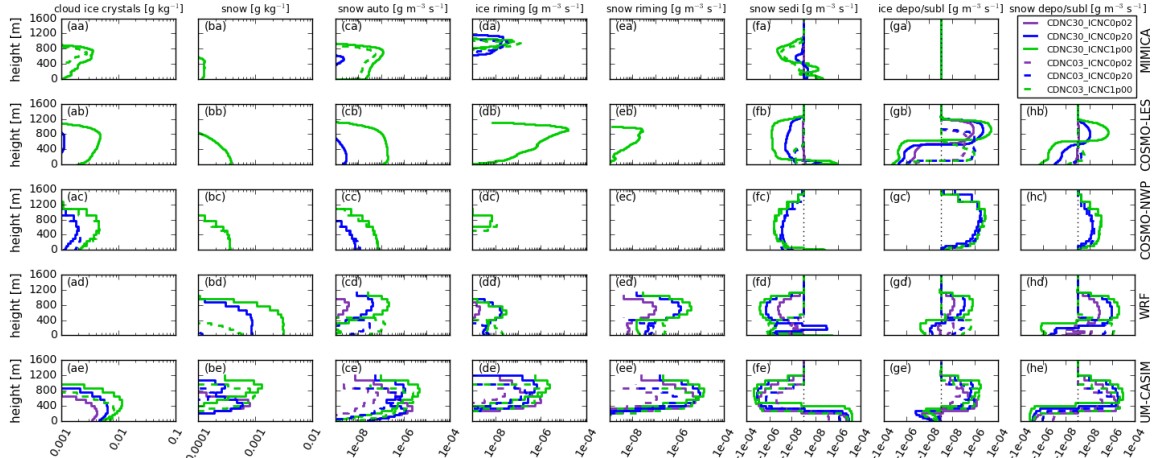

**Figure 13.** Tendencies of ice and snow mass due processes affecting frozen cloud mass for the prescribed CDNC simulations. First letter in subplot labels refers to column and second to row. Mass mixing ratios of cloud ice (column a) and snow (column b), tendencies of cloud ice and snow mass due to autoconversion to snow (column c), riming by cloud ice (column d), riming by snow (column e), sedimentation of snow (column f), deposition + sublimation of cloud ice (column g), and deposition + sublimation of snow (column h). Results shown for the (from top to bottom) MIMICA (row a), COSMO-LES (row b), COSMO-NWP (row c), WRF (row d), and UM-CASIM (row e).

The differences in process rates between models are due to both differences in the parameterisation of the physical processes as well as differences in the representation of the size distributions of the frozen cloud species in the different models. Additional contributions to these differences would come from differences in model meteorology and model resolution. In the next section we will examine the sensitivity to ICNC in the context of CCN and CDNC changes.

## 6.2 Sensitivity to CDNC, CCN, and ICNC

### 6.2.1 Prescribed CDNC and fixed aerosol simulations

In order to summarise our results with different prescribed ICNCs, Fig. 14 shows box plots of the LWP (including cloud droplets and rain), IWP (including cloud ice crystals, snow, and graupel), and surface net LW radiation from each model for
15 all of the CDNC30, CDNC03, CCN30fixed, and CCN80fixed cases during the period after Aug. 31st, 2008, 1200 UTC. For

the three NWP models, we show a similar figure with spatial statistics for a $100 \text{ km}^2$ area as Fig. S2. The NWP models show more variation across time because they include time-varying large scale features not considered by the LES models. We note that we do not expect the prescribed CDNC or prescribed CCN cases to capture the "cloudy" to "nearly-cloud-free" transition, so we do not attempt to sample the models during these observed time periods. However, if the tenuous cloud hypothesis is correct, the cloud states resulting in each model for the cases with greater prescribed CDNC and CCN concentrations would be expected to be more representative of the "cloudy" period, and the cloud states for the cases with lesser prescribed CDNC and CCN concentrations would be expected to be more representative of the "nearly-cloud-free" period. Our choice of time period allows 24 hours for the models to reach a representative state and consists of 24 hours of modelled time for the three NWP models and MIMICA. The COSMO-LES results include seven hours of model time before averaging, and the averaging period covers nine hours of modelled time. We note that the choice of averaging period is arbitrary, but our conclusions are not sensitive to changes in the averaging period, with a few exceptions: First, the initial period required for each NWP model to form a liquid cloud above the surface must be excluded (6-18 hours). Second, the MIMICA model predicts increased glaciation of the cloud with time in the two ICNC1p00 cases, with LWP, IWP, and surface net LW radiation steadily decreasing in magnitude with time. Third, the UM-CASIM model predicts that the cloud altitude decreases after ~36 hours of simulation for all cases where a cloud is simulated, as can be seen in e.g. Figs. 3, 6, 8, and 11. This leads to decreases in LWP, IWP, and the magnitude of the surface net LW radiation, if this time period is included. Fourth, the COSMO-NWP model predicts a stable frozen cloud in the CDNC30_ICNC1p00, CDNC03_ICNC0p20, and CDNC03_ICNC1p00 cases until 30 hours of simulation time (Aug. 31st, 1800 UTC). After 30 hours a drizzling mixed-phase cloud forms, similar to the results shown after 30 hours of simulation in Fig. 12. All of these effects will be discussed further later in this section.

Figure 14 also shows the observed interquartile range for the "cloudy" and "nearly-cloud-free" periods as hatched and shaded region, respectively. These periods are defined and discussed in Sect. 2. The interquartile range plotted accounts for time variance in the observations. We do not explicitly account for observational error, but random observational error will contribute to this time-variance.

The median LWP predicted by the models spans nearly two orders of magnitude, from $2.5 \text{ g m}^{-2}$ for the COSMO-LES CDNC03_ICNC1p00 simulation to $190 \text{ g m}^{-2}$ for the MIMICA CDNC30_ICNC0p02 case. The MIMICA model tends to produce the largest LWPs. COSMO-LES produces the smallest LWPs for the CDNC03 cases, and COSMO-NWP produces the smallest LWPs for all other cases, where simulated. Every model for every value of ICNC shows an increase in LWP as CDNC is increased from $3 \text{ cm}^{-3}$ to $30 \text{ cm}^{-3}$, and almost every model shows an increase in LWP as the fixed CCN concentration is increased from $30 \text{ cm}^{-3}$ to $80 \text{ cm}^{-3}$. However, the magnitude of this increase varies greatly from model to model. Notably, COSMO-NWP shows the smallest differences in LWP between different cases, with no significant change in LWP between the CCN30fixed and CCN80fixed cases. We noted earlier in Sect. 5.1 that a greater fraction of cloud droplet mass autoconverts to rain in COSMO-NWP compared to MIMICA, WRF, or UM-CASIM, regardless of the prescribed CDNC value chosen for activation. Therefore, a larger fraction of the liquid in the COSMO-NWP results consists of rain as opposed to cloud droplets,

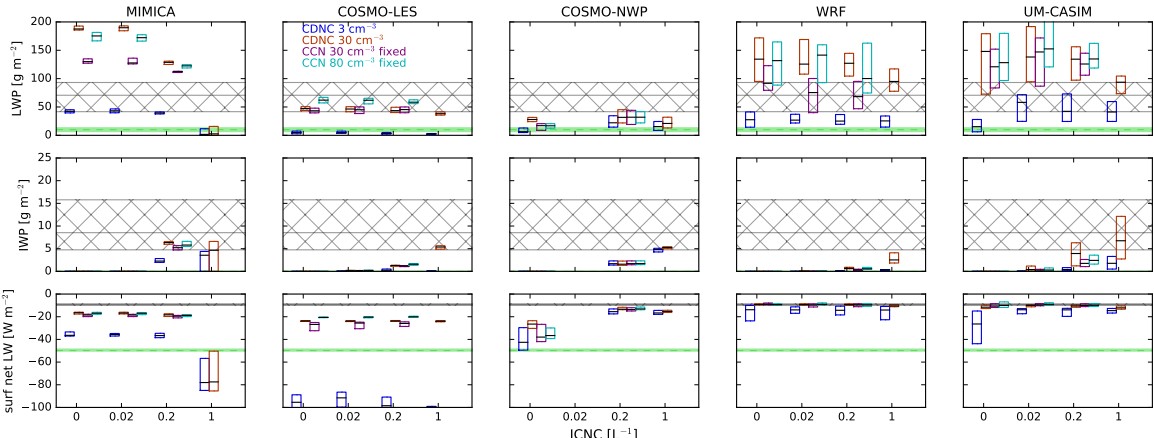

**Figure 14.** Water paths and net longwave radiation for all simulations without aerosol processing. Top row: liquid water path, middle row: ice water path, bottom row: surface net longwave radiation. Each subplot shows results from a single model. From left to right: MIMICA, COSMO-LES, COSMO-NWP, WRF, and UM-CASIM. Simulations with prescribed CDNCs of 3 $\mathrm{cm}^{-3}$ (CDNC03) and 30 $\mathrm{cm}^{-3}$ (CDNC30) are shown as blue and red boxes, respectively, and simulations with prescribed CCN concentrations of 30 $\mathrm{cm}^{-3}$ (CCN30fixed) and 80 $\mathrm{cm}^{-3}$ (CCN80fixed) are shown as purple and turquoise boxes, respectively. Within each subplot, the ICNC is increased from left to right as 0 $\mathrm{L}^{-1}$, 0.02 $\mathrm{L}^{-1}$, 0.2 $\mathrm{L}^{-1}$, and 1 $\mathrm{L}^{-1}$. Boxes show the interquartile range over model results after Aug. 31$^{\text{st}}$ 1200 UTC, and the black horizontal lines denote the medians. Hatched regions indicate observed interquartile range for the "cloudy" period, and the green shaded regions indicate the range for the "nearly-cloud-free" period.

compared to the other models. As the CDNC is decreased, either through changes in the prescribed CDNC or changes in the CCN concentration, further losses in cloud droplet mass mixing ratios are partially compensated by increases in rain mass, reducing differences in total LWP. We showed in Sect. 5 that MIMICA generally predicts less autoconversion than the other models, and as a result the proportion of the LWP composed of rain in MIMICA is less, and so it shows the greatest sensitivity
5  to changes in CDNC.

In general, the model results show decreases in LWP with increasing ICNC, but these changes are generally small relative to the sensitivity to our choice of representation of cloud droplet activation. Larger prescribed ICNCs increase removal of liquid water through riming, and through deposition via the Wegener–Bergeron–Findeisen process (see Fig. 13). The MIMICA
10  model predicts almost complete glaciation for ICNC = 1 $\mathrm{L}^{-1}$, and so produces a much reduced LWP for those cases. LWPs within COSMO-NWP are reduced to near-zero for the first 30 hours of the CDNC30_ICNC1p00, CDNC03_ICNC0p20, and CDNC03_ICNC1p00 COSMO-NWP simulations due to glaciation of the cloud, but after 30 hours a drizzling cloud forms with LWP not strongly dependent on the prescribed ICNC concentration.

Median IWPs predicted by the models for non-zero ICNC range from ice-free for the MIMICA CDNC03_ICNC0p02 case to 7.2 g m$^{-2}$ for the UM-CASIM CDNC30_ICNC1p00 case. The model results show increases in IWP with prescribed ICNC, excepting the MIMICA ICNC1p00 cases where the cloud glaciates and dissipates. If a shorter averaging period was used, the IWPs for these two cases would be larger than those for the ICNC0p20 cases. The IWPs predicted by WRF and UM-CASIM are roughly linear with respect to the prescribed ICNC concentration over the range used here: each ten-fold increase in ICNC increases the IWP by roughly a factor of ten. Within COSMO-LES, increases in IWP are sub-linear with respect to increases in ICNC: The IWP increases by a factor between 5.3 and 7.6 as the prescribed ICNC is increased by a factor of ten from 0.02 L$^{-1}$ to 0.2 L$^{-1}$. IWPs are also sub-linear with respect to ICNC in COSMO-NWP: The IWP increases by a factor of either 2.8 or 3.3 as the prescribed ICNC is increased by a factor of five from 0.2 L$^{-1}$ to 1 L$^{-1}$. Median IWPs also generally increase with increases in CDNC or increases in CCN concentrations, due to the increased cloud water available to freeze and form ice.

The net surface LW radiation within each model is generally well correlated with the LWP within each model. As has been discussed in Intrieri et al. (2002), Arctic clouds have a net warming effect over sea-ice due to the high albedo of the surface and the low angle of incoming solar radiation. Variability in the surface net LW is greater for cases with lower LWPs than for cases with high LWPs, as the LW emission by clouds saturates for large values of LWP. The LW dependence on LWP is stronger in the LES models than in the NWP models. This is primarily due to the experimental setup: within the NWP models the surface temperature is predicted in part based on radiative flux balance, whereas it is held fixed in the LES models. When there is less cloud, less LW radiation is re-emitted back towards the surface, and the surface would be expected to cool more quickly, which would then reduce the LW emission from the surface.

For the MIMICA, COSMO-LES, WRF, and UM-CASIM models, a CDNC between 3 cm$^{-3}$ and 30 cm$^{-3}$ could be prescribed that yields a LWP within the interquartile range of observed LWP during the "cloudy" period, but this prescribed CDNC value is not consistent across models. Unfortunately, in-cloud CDNC measurements were not available for the period studied here, so the models cannot be constrained based on this measurement. Also, as discussed above, the CDNC-LWP relationship for this case appears to be dominated by the partitioning of liquid water between cloud droplets and rain within each model, which is often tunable through the cloud droplet size distribution parameters or a parameter in the autoconversion scheme such as the maximum cloud droplet size. LWPs consistent with those observed during the "nearly-cloud-free" period were produced by simulations where the cloud dissipated, regardless of the mechanism of cloud dissipation. The cloud glaciates in MIMICA simulations with a prescribed ICNC of 1 L$^{-1}$, and the cloud temporarily glaciates in the CDNC30_ICNC1p00, CDNC03_ICNC0p20, and CDNC03_ICNC1p00 COSMO-NWP simulations. The cloud rains out in COSMO-LES simulations with a prescribed CDNC of 3 cm$^{-3}$, and in the COSMO-NWP simulation with a prescribed CDNC of 3 cm$^{-3}$ and no cloud ice.

The median IWP from each model for every case is less than the median observed IWP for the "cloudy" period. However, as discussed in Sect. 2, there is a large uncertainty in the observed IWP, which is partially responsible for the large time-variance in the observed IWP. For COSMO-LES, COSMO-NWP, and UM-CASIM, a prescribed ICNC of 1 L$^{-1}$ is required to produce

a median IWP within the inter-quartile range of the observed IWP. The MIMICA model produces an IWP within this range with an ICNC of $0.2\,\text{L}^{-1}$. As noted previously, the MIMICA model predicts glaciation and dissipation if an ICNC of $1\,\text{L}^{-1}$ is prescribed, and the averaging period used here includes the dissipation of the cloud. If a shorter averaging period was used, the IWP for these two cases would be larger than those for the ICNC0p20 cases.

Median surface net LW radiation from nearly all WRF and UM-CASIM simulations with LWP $> 75\text{g}\,\text{m}^{-2}$ is consistent with the observations for the "cloudy" period. However, despite larger LWPs, MIMICA predicts too much LW emission. This is due in part to the prescribed surface temperatures in our experimental setup being too warm, as described above. This also contributes to the discrepancy between the LW emission observed during the "nearly-cloud-free" period and the MIMICA and COSMO-LES results with LWPs consistent with the "nearly-cloud-free" period.

### 6.2.2 Prognostic aerosol simulations

Figure 15 shows a similar plot to Fig. 14 for the cases with prognostic aerosol processing. For the COSMO-NWP and UM-CASIM models, we show a similar figure with spatial statistics for a $100\,\text{km}^2$ area as Fig. S3. Here we also include N50 concentrations at $20\,\text{m}$ from the surface, for consistency with the measurement inlet height. Note that N50 concentrations are not available from COMSO-NWP. N50 concentrations from the other three models for the CCN30prog simulations overlap with those observed for the "cloudy" period, except for the UM-CASIM CCN30prog_NOICE case, which yields greater N50 concentrations. N50 concentrations from MIMICA and UM-CASIM for the CCN03prog cases overlap with those observed for the "nearly-cloud-free" period. The UCLALES-SALSA CCN03prog_NOICE simulation predicts very little depletion of N50 from the initial values, discussed in Sect. 5.4. The MIMICA and UM-CASIM models simulate clouds with reduced vertical extents and lower LWCs, and therefore lower LWPs, with prognostic aerosol than with time-invariant aerosol concentrations. Similarly, IWPs are also lower due to the lower amount of liquid water available to freeze. The LWPs simulated by MIMICA with an initial CCN concentration of $30\,\text{cm}^{-3}$ are consistent with observations during the "cloudy" period. UM-CASIM and UCLALES-SALSA produce LWPs consistent with the "cloudy" period with initial CCN concentrations of $80\,\text{cm}^{-3}$. All simulations where the cloud layer dissipated (initial CCN concentration of $3\,\text{cm}^{-3}$ in all models, and initial CCN concentration of $30\,\text{cm}^{-3}$ with UCLALES-SALSA) produce LWPs within measurement error of the "nearly-cloud-free" period.

When the initial CCN concentration is $80\,\text{cm}^{-3}$, UCLALES-SALSA, MIMICA, and UM-CASIM predict that below-cloud N50 concentrations remain above 50% of initial N50 concentrations (see Fig. 8). This reduction in aerosol number is due to in-cloud processing and drizzle deposition to the surface, offset by re-suspension of aerosol from evaporation and sublimation of hydrometeors. An initial CCN concentration of $30\,\text{cm}^{-3}$ yields N50 concentrations at $20\,\text{m}$ consistent with observations for all cases where this information is available, except for the UCLALES-SALSA case and the UM-CASIM case with no ice nucleation. In the former, the cloud dissipates and N50 is depleted throughout the boundary layer. The latter case produces the least rain of all the cases simulated with an initial CCN concentration of $30\,\text{cm}^{-3}$, and has the least removal of aerosol

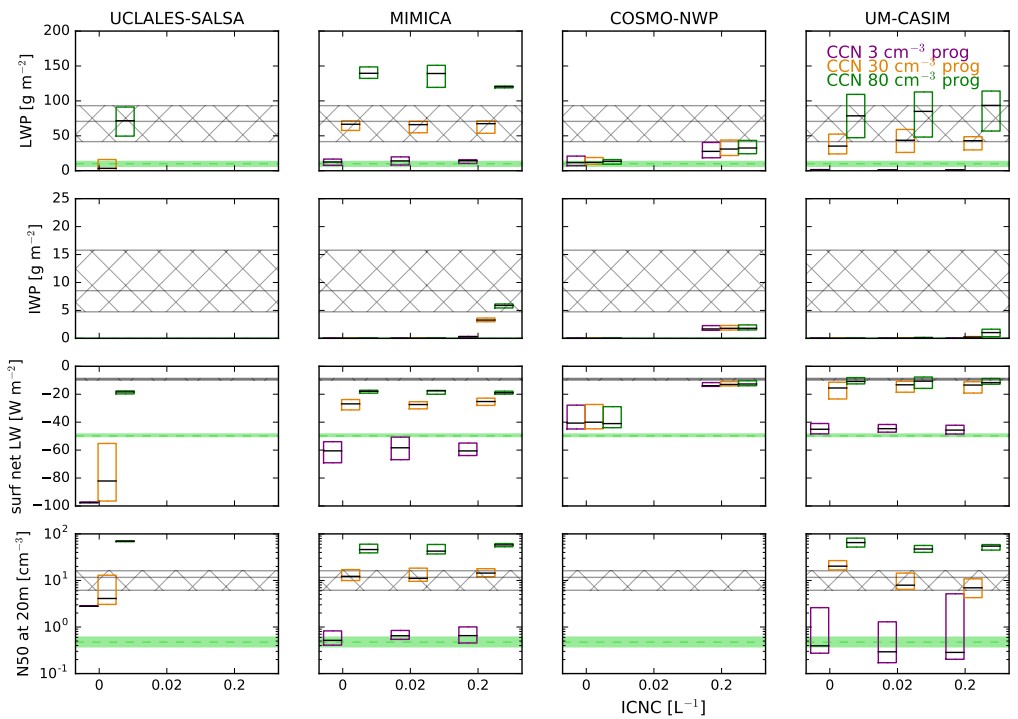

**Figure 15.** Cloud and surface properties for all simulations with prognostic aerosol. Top row: liquid water path, second row: ice water path, third row: surface net longwave radiation, bottom row: N50 concentrations at 20 m from the surface. Each subplot shows results from a single model. From left to right: UCLALES-SALSA, MIMICA, COSMO-NWP, and UM-CASIM. Simulations with an initial CCN concentration of 3 $cm^{-3}$ (CCN03prog), 30 $cm^{-3}$ (CCN30prog) and 80 $cm^{-3}$ (CCN80prog) are shown as purple, orange and green boxes, respectively. Within each subplot, the ICNC is increased from left to right as 0 $L^{-1}$, 0.02 $L^{-1}$, and 0.2 $L^{-1}$. Boxes show the interquartile range over model results after Aug. 31st 1200 UTC, and the black horizontal lines denote the medians. Hatched regions indicate observed interquartile range for the "cloudy" period, and the green shaded regions indicates the range for the "nearly-cloud-free" period. Note that N50 concentrations are not available from the COSMO-NWP model.

to the surface. Median N50 at 20 m for all cases with an initial CCN concentration of 3 $cm^{-3}$ is below 1 $cm^{-3}$, except for the UCLALES-SALSA results, where N50 is depleted in-cloud, but no mixing of the depleted layer with lower layers occurs following cloud dissipation (see Fig. 10). There is no clear effect across models of changes in prescribed ICNC on modelled N50 concentrations.

# 7 Conclusions

In this study, we have compared the results of three LES models and three NWP models for a tenuous-cloud-regime case study observed during the 2008 ASCOS field campaign. We began with simulations using prescribed CDNC and prescribed ICNC, progressed to simulations with prognostic CDNC based on a constant aerosol size distribution, and finally showed simulations using prognostic aerosol processing along with prognostic CDNC. Our key findings are the following:

Our modelling results strongly support the hypothesis that the LWC, and hence the radiative effects, of these clouds are highly sensitive to CCN concentrations; in order words, they are CCN-limited. For the observed meteorological conditions, all models predict that the cloud does not collapse as observed when the CCN concentration is held constant at the value observed during the cloudy period, but the clouds thin or collapse as the CCN concentration is reduced. Cloud dissipation due to glaciation is predicted only by the MIMICA model, and only for a prescribed ICNC of $1\ \mathrm{L}^{-1}$, the largest value tested in this study. As the IWP was generally underestimated compared to the observed IWP, it is possible that the contribution of glaciation to dissipation was also underestimated. Global and regional models with either prescribed CDNCs or prescribed aerosol concentrations would not reproduce cloud dissipation due to low CCN concentrations, and so would not capture this source of variability in cloud LWC and hence cloud radiative effects. Therefore, this suggests that linkages between aerosol and clouds need to be considered for weather and climate predictions in this region. In particular, we recommend that studies are carried out to determine if CCN-controlled cloudiness has a remote effect on important weather phenomena such as mid-latitude blocking. If it does, then we recommend that aerosol-cloud interactions be included to capture the impact on the more populated mid-latitude regions.

All models predict increasing LWP with increasing CDNC, either through prescribed CDNC values or changes in available CCN concentrations. The increases in LWP and subsequent decreases in surface net LW radiation with increasing CCN concentrations or prescribed CDNC suggest that increased aerosol concentrations in the high Arctic during the clean summer period would have a warming effect on the surface, potentially resulting in more thinning of sea-ice or a delay in autumn freeze-up events. Our results suggest this effect would be most dramatic where CCN concentrations increase beyond the threshold value required to prevent cloud dissipation.

Most models simulate increasing IWP with increasing prescribed ICNC, and decreasing LWP with increasing ICNC, due to increased efficiency of the WBF process with increased ICNC. This is consistent with the results of previous investigations of the sensitivity of Arctic mixed-phase cloud to the representation of ice nucleation (e.g. Avramov and Harrington, 2010; Fridlind et al., 2012; Harrington et al., 1999; Jiang et al., 2000; Klein et al., 2009; Morrison et al., 2003, 2005b, 2011; Ovchinnikov et al., 2014; Pinto, 1998; Prenni et al., 2007; Solomon et al., 2009; Young et al., 2017). However, the effects of changes in ICNC on LWP and surface net LW were generally weaker than the effects of changes in CDNC or CCN across the ranges tested in this study. This is consistent with results found by Possner et al. (2017) where the total water path and net surface LW were

to first order determined by CDNC or CCN concentrations, rather than INP concentrations, for CCN and INP perturbations of similar magnitude as considered in this study. However, for larger INP perturbations (exceeding $1 \, \mathrm{L}^{-1}$) in a low-INP regime, INP perturbations were seen to potentially offset, if not reverse, the cloud response to CCN perturbations. If INP concentrations in the Arctic were to increase beyond $1 \, \mathrm{L}^{-1}$ due to changes in transport from low latitudes or increases in local emissions,

these could induce large changes in cloud properties. However, this value is greater than those observed previously in the high Arctic (Bigg, 1996; Bigg and Leck, 2001).

Despite some common model behaviours, there is large inter-model diversity in the sensitivities of the models to changes in CDNC or CCN concentrations. The change in LWP due to an increase in prescribed CDNC from $3 \, \mathrm{cm}^{-3}$ to $30 \, \mathrm{cm}^{-3}$ varies

from ~$10 \, \mathrm{g \, m}^{-2}$ to ~$100 \, \mathrm{g \, m}^{-2}$ depending on the choice of model alone. Cloud dissipation was predicted by the COSMO-LES, COSMO-NWP and WRF models for a prescribed CDNC of $3 \, \mathrm{cm}^{-3}$, suggesting that the critical CDNC for these models was between 3 and $30 \, \mathrm{cm}^{-3}$. The critical CDNC for the other models must be less than $3 \, \mathrm{cm}^{-3}$. In the prognostic aerosol cases, the critical initial CCN concentration was between 30 and $80 \, \mathrm{cm}^{-3}$ for the UCLALES-SALSA model, and between 3 and $30 \, \mathrm{cm}^{-3}$ for the MIMICA and UM-CASIM models. The COSMO-NWP model did not predict dissipation of the cloud for any

of the prognostic aerosol cases. We did not test the sensitivity of these critical values to model processes, but it is likely that they are sensitive to the specific setup of each model used in this study, specifically regarding cloud droplet size distributions and the representation of autoconversion of cloud droplets to rain. Faster autoconversion rates per unit cloud droplet mass are associated with lower sensitivities in all cloud properties to changes in prescribed CDNC or CCN concentrations. Large differences in autoconversion rates per unit cloud droplet mass were simulated despite a similar treatment of autoconversion

in four of the models, even in cases with prescribed cloud droplet activation and no frozen cloud processes permitted. Our results therefore suggest that some caution is necessary in interpreting the results of any single model, including the sensitivities of model results to perturbations in aerosol concentrations. Properly estimating aerosol-cloud interactions requires careful consideration regarding the representation of cloud droplet size distributions, as well as the choice of autoconversion scheme, and the parameters set therein if an empirical formulation is chosen. Our results also suggest that observations should aim to

constrain the representation of rain formation and extend the validity of parameterisations to the Arctic domain. We therefore recommend that future observational campaigns aim to perform in-situ observations of cloud LWC, IWC, and hydrometeor size distributions, as well as aerosol size and concentration profiles above and below cloud.

The strength of aerosol sources will be critical for the stability of tenuous Arctic clouds. When aerosol removal by activa-

tion into cloud droplets was included in the simulations, this decreased simulated CDNCs and LWPs. The rate of depletion of potential CCN within the boundary layer varied strongly between different models and depending on the initial aerosol concentration. For greater initial aerosol concentrations, precipitation formation was suppressed, decreasing the removal of aerosol to the surface. This supports a positive feedback mechanism whereby increasing aerosol concentrations suppress drizzle formation, reducing the sink of aerosol to the surface. We note that we did not investigate here replenishment of CCN by surface

sources or by aerosol nucleation and growth, but that Igel et al. (2017) have shown that cloud-top entrainment is important for

CDNC (and hence cloud radiative properties) in this case. Entrainment would be included in the results presented here, but as we applied constant initial CCN concentrations throughout the simulated atmosphere, the above-cloud aerosol concentration available for entrainment was identical to the initial boundary-layer aerosol concentration.

A potentially important feedback is that cooling of the sea-ice surface following cloud dissipation increases atmospheric stability near the surface, further suppressing cloud formation. Surface fluxes were predicted to be small by the NWP models so long as a sufficiently thick cloud layer was simulated (surface fluxes were prescribed in the LES models). However, under thin-cloud or cloud-free conditions, the cooling of the surface due to LW emission increased the stability of the near-surface atmospheric layer. The WRF model with a prescribed CDNC of $3 \, \mathrm{cm}^{-3}$ predicts that any subsequent cloud will be constrained

to a shallow mixed layer at the surface, resulting in surface fog (Fig. 6). This effect can also be seen in the potential temperature profiles predicted by UM-CASIM for the CCN03prog_NOICE case (Fig. 10). Therefore, this suggests that linkages between clouds, surface temperatures and atmospheric stability may need to be considered for weather and climate predictions in this region.

We primarily focus on cloud microphysical processes in this work, but it is important to note also the contribution of large-scale atmospheric circulation patterns to cloud cover and thickness (e.g. Kay and Gettelman, 2009) as well as sea ice (e.g. Serreze and Stroeve, 2015). However, our results highlight the sensitivity of high-Arctic clouds to CCN concentrations, the importance of the model representation of rain formation in clouds for correctly capturing this sensitivity, and the interactions between clouds, surface temperatures, and atmospheric stability. Future studies of the interactions between Arctic clouds, sea-

ice, and climate must take account of all of these findings.

There are many aspects of high Arctic aerosol-cloud interactions that were beyond the scope of this study to address. Future studies should aim to address the possible role of aerosol replenishment by new-particle formation, surface sources, and transport using models that include coupled aerosols and chemistry with active sources and sinks. The formation of new clouds or

fog after dissipation events as aerosol concentrations are replenished also needs to be investigated. More case studies based on additional observational campaigns need to be performed. Uncertainty analyses are necessary to explore the simultaneous contributions of multiple compensating factors. More investigation of surface thermodynamics and feedbacks is also necessary.

*Acknowledgements.*  We thank the two anonymous reviewers for their comments on this manuscript. We gratefully acknowledge support from

the European Union's Seventh Framework Programme (FP7/2007-2013) Impact of Biogenic versus Anthropogenic emissions on Clouds and Climate: towards a Holistic UnderStanding (BACCHUS) project (grant no. 603445) and European Research Council projects ECLAIR (grant no. 646857) and C2Phase (grant no. 714062). We acknowledge use of the MONSooN system, a collaborative facility supplied under the Joint Weather and Climate Research Programme, a strategic partnership between the UK Met Office and the Natural Environment Research Council. We also acknowledge use of the JASMIN system operated by Centre for Environmental Data Archival (CEDA) as well as the Swiss

National Supercomputing Centre (CSCS). B. Wehner, D. Orsini, M. Martin and J. Sjögren are much appreciated for providing the size-resolved particle number and the CCN observations. Caroline Leck and Michael Tjernström are specifically thanked for their coordination of ASCOS. The Swedish Polar Research Secretariat provided access to the icebreaker Oden and logistical support. We would like to thank Joseph Sedlar, Thorsten Mauritsen, and Matthew Shupe for the observational data reprinted in this manuscript, and for their comments on an

5   early version of the manuscript.

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
