# Peer review of "Figure S1. Time-varying distributions of liquid water paths and ice water paths from COSMO-NWP, WRF, and UM-CASIM for the case with a prescribed CDNC of $30 \text{ cm}^{-3}$ and ICNC of $0.2 \text{ L}^{-1}$ (CDNC30\_ICNC0p20). All values are shown for a $100 \text{ km}^2$ area centred at t"

_Atmospheric Chemistry and Physics, 2017_

## Referee Comment (RC1) · Anonymous Referee #1 · 23 Mar 2018

The paper presents an intercomparison of simulated summertime mixed-phase Arctic clouds from large-eddy simulations and numerical weather prediction models. The simulated case is based on observations from the 2008 Arctic Summer Cloud Ocean Study when the CCN concentration was very low ($\sim$ 1 cm-3). The study reports the results of several sensitivity tests, which show the dependence of cloud properties on the treatment of cloud droplet activation and on the number concentration of CCN or droplets. Several models are run with fixed and prognostic droplet number concentration representations. Most models show dissipation of the liquid cloud when the CCN concentration falls below a certain value, but the threshold values vary significantly among models. Sensitivity of simulations to ice crystal number concentration (ICNC) has also been tested. In general, it is found that the changes in liquid water

path (LWP) resulting from changes in ICNC are smaller that changes in LWP due to varying CCN concentration, although the ranges of tested concentrations are somewhat arbitrary. Overall, the LWP increases with increasing CCN or droplet number and decreases with increasing ICNC, in agreement with previously published results. Although in many respects models behave qualitatively similarly, there are large quantitative differences. In several sections the manuscript documents performances of and the differences among the models in these sensitivity tests, which is certainly a useful exercise. Unfortunately, the paper provides little insight into the causes of the differences. More analysis and expanded interpretation of the results are definitely needed and more simulations are highly recommended to make the manuscript publishable in ACP. At present, most of the sections describe the inter-model differences shown on the plots but provide little to none of substantive analysis. The paper summarizes a significant multi-institutional research effort and so readers expect to learn more than just that the models produce different results. Many of the statements in the paper that go beyond simple description of the results are either obvious (state previously well-established facts, e.g., that modeled cloud-aerosol interaction depends on droplet size distribution and autoconversion scheme) or speculative. I strongly encourage the authors to expand the analysis to tease out specific reasons for the differences and hopefully provide practical rather than general recommendations to other modelers and observationalists. Several more specific comments and recommendations are given below.

General comments: 1) Model and simulation choices: Although six models participate in the intercomparison, for each considered case no more than 2 LES models or 3 NWP models can be directly compared, so the representativeness of the results is somewhat questionable. The setup of LES and NWP models is so different by design, that comparing or even putting them on same plots is not really meaningful. Although in the text the paper acknowledges these differences between LES, run with constant forcing and aimed primarily at steady state regimes, and NWP models, in which results at any location are affected by mesoscale variability evolving in time, the presentation

of the results in figures implies that the two groups are stacked against each other, which will likely confuse some readers. Furthermore, out of 3 LES models (UCLALES-SALSA, COSMO-LES, and MIMICA) only MIMICA ran all the cases, while the other two do not even have a single overlapping case. Simulations from three NWP models (COSMO-NWP, WRF, and UM-CASIM) are more evenly distributed with 10 cases run by all three. 2) LES results: It is hard to make sense of LES results because the models disagree a lot even for a seemingly simple case of no ice and "high" droplet concentration (80 per cc). LES of boundary layer clouds have a long history of intercomparisons, including several for super-cooled clouds (e.g., M-PACE, SHEBA, and ISDAC cases), so the modeling community has a relatively good understanding of what the simulations should behave like under similar conditions. The models used in this study are relatively new and it is not clear how they measure up against an ensemble of previously tested models. Cited "numerical instabilities" that prevented COSMO-LES to be run for the required period of time are very disconcerting and cast shadow on all simulations from that model. Notable differences in LWP between USLALES-SALSA and MIMICA in the CCN80prog_noice case, which are apparent in figure 7, deserve an explanation, especially since UCLALES-SALSA employs an original and relatively untested microphysics scheme, which is different from what was previously used in the UCLALES model. Repeating these simulations with collision-coalescence (or autoconversion) turned off may help to identify the sources for the difference. Explaining this difference may also shed light on much stronger sensitivity of SALSA microphysics to the droplet concentration reduction from 80 to 30 per cc. 3) Aerosol, CCN, and droplet concentrations: The study targets clouds with extremely low droplet concentrations (~1 to 10 per cc) and more justification is needed to convince readers that these concentrations are relevant for the considered case. This can be accomplished by answering, or at least discussing, the following questions. Are surface-based CCN measurements representative of cloud layer conditions, given that, according to the sounding in figure 2, the cloud layer between 600 and 1000 m appears to be decoupled from the surface? Supersaturation of 0.2 % for which the CCN concentration were measured may

represent conditions in boundary layer clouds in a typical aerosol environment. When the CCN concentration is very low, however, wouldn't higher supersaturation values be achievable even in clouds with moderate updrafts? Another way to look at it is this. Are there smaller aerosol particles, or less efficient CCN, that could be activated in the considered clouds? Finally, are there any measurements, direct or via remote sensing retrievals, of actual droplet concentrations in these clouds that would serve as a target for model simulations?

Technical comment: P 5, lns 19-25: It is worth to provide a brief basic description of the ASCOS campaign, specifying location, overall synoptic situation, whether all measurements were collected from the surface or whether aircraft was involved, etc.. Table 1: (i) typo in coarsest vertical resolution below 2 km for COSMO-LES; it should be smaller than 228.3 m; (ii) any reason why LES models don't use identical horizontal grid size? Seems like an extra and unnecessary source of uncertainty to deal with; (iii) please include domain size in the table, at least in horizontal directions. For NWP models it is given somewhere in the text, but I don't recall seeing numbers for LES. P 9, ln 31: remove "five" P 10, ln 8-9: delta_ICNC has units of concentration and, therefore, is not technically a "rate". P 13, ln 16-17: Showing profiles from a single grid column in the middle of the domain is an unorthodox way to compare LES models. Wouldn't the comparison be more robust if domain mean profiles were used? NWP output can then also be averaged over LES domain-size area to be more comparable in terms of represented horizontal scales. P 13, ln 26: "Despite no inclusion of ice . . ." The sentence does not make sense, because ice won't help models in their current setup to produce clouds. Please re-phrase. P 14, figure 3: Here and in number of subsequent figures, the deepest mixed layer or highest cloud top seems to be predicted by a model with coarsest vertical resolution (COSMO-NWP). May be worth pointing this out. P 14, ln 10: ". . . adequately resolve . . ." This statement seems too optimistic. Although time varying advective tendencies at the studied location in NWP models are almost certainly more realistic than constant tendencies imposed on LES, it is not clear how "adequate" they are. Boundaries are 100's km away from that location and so even if the

boundary conditions were perfect (they are not), shallow layers and sharp inversions can be significantly eroded during advection, e.g., due to excessive diffusion because of coarse vertical resolution. P 15, figure 4: High autoconversion rates in COSMO-LES between 200 and 700 m altitudes are puzzling since no cloud water is shown for these levels in figure 3. An explanation is needed here. P 16, ln 1-2: Since different papers often use different formulations and/or notation for gamma distributions, please provide the functional form of that distribution. This would also define your shape parameters, which are currently undefined. Also, in most common notations, nu=0 results in an exponential size distribution, which is often employed for precipitation species, but presents a questionable choice for cloud droplet size spectra. Please clarify what droplet size distributions are used in COSMO and UM-CASIM and, if they are indeed exponential, justify the choice. P 16, ln 31: "mass concentration": I think "mixing ratio" was used earlier in the paper. Better to use the same terminology/unit throughout. P 18: This is one of the sections, which lacks a clear message. What the reader is supposed to take out of this, except that the model results differ? The description in this section is mundane: turbulence may contribute, collision-coalescence could play a role, and an activation scheme obviously affects how many droplets are formed. Is there anything new that the intercomparison can teach us and that has some broader implications? P 19, figure 6: here and on other figures, lines that are called "red" and "purple" are hard to distinguish on this plot. P 20, ln 7-9: "It is therefore possible . . ." As written, it is not clear if narrower spectra in UCLALES-SALSA is the author's speculation or an actual finding of the study. This can be shown clearly and explicitly by plotting the cloud droplet size distributions from different models. P 20 ln 10-28: I find it odd to pull 1 out of 3 figures in this set of sensitivity experiments out of the main paper into the supplement leaving 1/2 page description in the text. Suggest to put the figure back into the paper. Figure S2, caption: "CCN80prog_noice" should be "CCN30prog_noice" P 21, figure 7: Non-zero CDNC throughout the vertical column in MIMICA during the model spin up looks odd and should probably be masked outside of clouds. P 22, ln 2-3: Not clear what is meant by the statement "Evaporation of falling rain . . . transports

moisture ... to the lower cloud layer", since the cloud layer is presumably saturated. Please re-phrase. P 22, ln 21-25: This section summary is not very insightful. Do we need a model intercomparison to state that the decrease in CDNC leads to thinning or collapse of the cloud layer and that this effect is sensitive to cloud-rain partitioning? P 31, ln 1-2: The recommendation to include linkages between aerosol and clouds in models is too general to be useful. Please elaborate. Many, if not most, climate models include prognostic aerosol and CDNC. Whether prediction models are a different story though. Do you recommend that NWP models move to prognostic aerosol too? Should they consider assimilating aerosol information? Please be more specific. P 32, ln 16-26: Specific "threshold" CCN or CDNC values for a given model and a single case are of little value to the broader modeling community. On the other hand, the authors can do more to disentangle some of the effects at play here. E.g., CDNC is certainly a factor in autoconversion rate, but the rate also depends on LWC. The models predict hugely different LWC and it is not clear whether different autoconversion rates are the reason or the consequence of differences in LWC. The most direct way to determine this is of course to swap the autoconversion parameterizations between the models. Another approach would be to examine autoconversion rate normalized by LWC, which may provide some hints into the interplay between the two but could still be affected by other factors. P 33, ln 9-13: Surface fluxes no doubt are important in the Arctic, but how is this relevant to the current study. Aren't the fluxes set to zero for LES models and very small in NWP models? And isn't the cloud layer, in fact, decoupled from the surface to begin with?
* * *

---

## Referee Comment (RC2) · Anonymous Referee #2 · 23 Mar 2018

General comments:

The authors test the sensitivities of simulated Arctic clouds to aerosol perturbations represented differently by multiple variables, such as prescribed droplet number concentrations, prescribed or variable cloud condensation nuclei (CCN) concentrations, and also prescribed ice crystal number concentrations, using three large-eddy simulation (LES) models and three numerical weather prediction (NWP) models. Microphysical processes in the simulated clouds are investigated in detail. The sets of simulations listed in Table 2 are well designed to test the sensitivity of clouds to different perturbations. Observational data from a field campaign is also presented, which helps the evaluation of the simulation results. They conclude that the clouds (or their water content) are CCN-limited, meaning that the properties of the clouds are heavily de-

pendent on the concentrations/existence of CCN. They also conclude that changes in these Arctic clouds may have impacts on the surface radiative balance since these clouds tend to have a warming effect. Figure 11 and 12 are especially interesting and highlight the findings from this paper. I would like to suggest some minor revisions/questions/comments below;

Specific comments:

Figure 1: Is there any observational data of surface precipitation available?

Page 7 line 1-2: Does this mean that microphysical processes in these clouds are purely liquid-based, not involving ice, due to possibly the lack of ice-nucleating particles (INPs)? Or ice often exists in these clouds (e.g. Figure 1), but just precipitation processes are dominantly through warm-rain?

Page 11 line 6-8: Does the fact that MIMICA is initialized at much earlier than the other two LES models have any impacts on the results?

Figure 2: It may be nice to also show actual temperature so that the temperatures for cloud-base and cloud-top can be roughly estimated.

Page 13 line 16-17: Does this mean that the results are taken from a single column at the center of the simulation domain? According to Figure S1 there seems to be a wide spatial variation in simulated results, but are analyzed results quite similar/different if domain-averaging or other methods are used, instead of extracting data from the center of the domain?

Figure 3 and 5: Since the observed quantity is LWC, how does simulated LWC look like (maybe add it in the third row)? Also, it would be helpful if you add dashed lines to indicate the defined "cloudy" and "nearly-cloud-free" periods in the "Observed LWC" figure.

Figure 4 and 5 captions: Although it says "Rain sedimentation tendencies for COSMO-NWP are not available.", the quantities are still plotted (bottom row, middle column), if

I'm understanding it correctly?

Figure 6: Can you maybe add a column for LWC so that the comparison of observation and simulations is possible?

Figure S2 caption: "CCN80prog_noice" should be modified to "CCN30prog_NOICE"

Page 22 line 30-32: While simulated results may be relatively insensitive to the inclusion of ice, IWC seems not to be negligible in Figure 1 as compared to LWC, though scales are different for LWC and IWC there. Does this mean that simulations are underrepresenting ice mass?

Figure 9: Although I'm aware of the consistent color scale for rain, ice, snow, and graupel, can you change the color scale for snow so that more information can be seen in colors? Or if snow and graupel masses do not play a major role in the whole microphysical processes in those clouds, the bottom two rows could be omitted. Also, I suggest adding observed and simulated IWC to the bottom row if possible.

Page 25 line 30-31: I wonder if some of the differences are due to other reasons, such as ambient conditions (simulated meteorology, especially when LES and NWP simulations are compared) and/or model resolutions, for example?

Figure 10: Maybe some of the columns that are not discussed much in the text can be omitted, so that each plot becomes a little larger?

Figure 11 and 12: If I understand the figure and the timeline in Figure 1 correctly, the boxes in these figures represent the simulated results for the transition period from cloudy to nearly-cloud-free period (1200UTC August 31st – 1200UTC September 1st). However, it does not include the cloudy period itself. That means, if one were to evaluate model performance or compare simulations with observations in this figure, should the boxes lie somewhere between hatched and shaded regions? Maybe the shading and hatching are for reference, but can you also provide the observed average over the same period (1200UTC August 31st – 1200UTC September 1st)? Also, hatching

seems to be a little weak and hard to see, so I suggest thickening it. Additionally, either shading for cloud-free period or hatching for cloudy period can be modified to a different color (currently both of them look grey), so that their differences become clearer (e.g., the row for net LW).

Figure 11 and 12 caption: Modify "indicates" to "indicate" in "... the shaded regions indicates ..."

Page 26 line 4: Remove "be" in "This would be result in..."

P.26 line 8-9: Does COSMO-LES have 7 hours for spin-up + 9 hours for analysis? At page 11 line 3, it was stated to be 2 hours of spin-up.

Page 28 line 33: Add "is" in "... This primarily due to ..."

It is better to consistently capitalize the word "_NOICE" throughout the paper, since there are currently places with "_noice" instead.

Figures: Minor suggestion, but it may be a good idea for all figures to have consistent labels. For example, it may be in the form of "1e-01", "0.1", or "10ˆ1" (e.g., Figure 6).
* * *

---

## Author Comment (AC1) · 5 May 2018

**We thank both referees for their comments. We copy their comments in italics below, and respond to them point-by-point.**

*Referee #1*

*The paper presents an intercomparison of simulated summertime mixed-phase Arctic clouds from large-eddy simulations and numerical weather prediction models. The simulated case is based on observations from the 2008 Arctic Summer Cloud Ocean Study when the CCN concentration was very low (~1 cm-3). The study reports the results of several sensitivity tests, which show the dependence of cloud properties on the treatment of cloud droplet activation and on the number concentration of CCN or droplets. Several models are run with fixed and prognostic droplet number concentration representations. Most models show dissipation of the liquid cloud when the CCN concentration falls below a certain value, but the threshold values vary significantly among models. Sensitivity of simulations to ice crystal number concentration (ICNC) has also been tested. In general, it is found that the changes in liquid water path (LWP) resulting from changes in ICNC are smaller that changes in LWP due to varying CCN concentration, although the ranges of tested concentrations are somewhat arbitrary. Overall, the LWP increases with increasing CCN or droplet number and decreases with increasing ICNC, in agreement with previously published results.*

*Although in many respects models behave qualitatively similarly, there are large quantitative differences. In several sections the manuscript documents performances of and the differences among the models in these sensitivity tests, which is certainly a useful exercise. Unfortunately, the paper provides little insight into the causes of the differences. More analysis and expanded interpretation of the results are definitely needed and more simulations are highly recommended to make the manuscript publishable in ACP. At present, most of the sections describe the inter-model differences shown on the plots but provide little to none of substantive analysis. The paper summarizes a significant multi-institutional research effort and so readers expect to learn more than just that the models produce different results. Many of the statements in the paper that go beyond simple description of the results are either obvious (state previously well-established facts, e.g., that modeled cloud-aerosol interaction depends on droplet size distribution and autoconversion scheme) or speculative. I strongly encourage the authors to expand the analysis to tease out specific reasons for the differences and hopefully provide practical rather than general recommendations to other modelers and observationalists. Several more specific comments and recommendations are given below.*

*General comments:*

*1) Model and simulation choices: Although six models participate in the intercomparison, for each considered case no more than 2 LES models or 3 NWP models can be directly compared, so the representativeness of the results is somewhat questionable. The setup of LES and NWP models is so different by design, that comparing or even putting them on same plots is not really meaningful. Although in the text the paper acknowledges these differences between LES, run*

*with constant forcing and aimed primarily at steady state regimes, and NWP models, in which results at any location are affected by mesoscale variability evolving in time, the presentation of the results in figures implies that the two groups are stacked against each other, which will likely confuse some readers. Furthermore, out of 3 LES models (UCLALES-SALSA, COSMO-LES, and MIMICA) only MIMICA ran all the cases, while the other two do not even have a single overlapping case. Simulations from three NWP models (COSMO-NWP, WRF, and UM-CASIM) are more evenly distributed with 10 cases run by all three.*

**Despite the differences between LES and NWP models, we still believe that there is value in comparing results from both types of model. In fact, we believe that if the differences are properly discussed, as we have attempted to do, the differences add value to the study. NWP models represent mesoscale variability, but LES models are better at representing small-scale turbulence and typically have a higher resolution representation of the boundary-layer top. Results that are consistent between both types of model (such as the increase in LWP with increases in CDNC or CCN, or the low sensitivity of the liquid phase to changes in ICNC) are more robust than if this study had included only LES or only NWP models. We believe that there is a particular benefit in comparing the COSMO-LES and COSMO-NWP results, which share the same microphysics. We note that Browning *et al.*, (1993) recommended the comparison of multiple scales of model in order to develop parameterisations for weather and climate models. We also note that multiple previous studies have compared results from similarly different types of models as our study (e.g. Klein *et al.*, 2009, Morrison *et al.*, 2009, Moeng *et al.*, 1996, Petch *et al.*, 2007, Petch *et al.*, 2014, Varble *et al.*, 2014a, Varble *et al.*, 2014b).**

**Unfortunately, it is not currently possible for all of the models to perform all of the cases. In particular, the versions of COSMO-LES and WRF used in this study have not been configured to perform prognostic aerosol simulations. Conversely, UCLALES-SALSA cannot currently be configured to run with fixed aerosol concentrations or fixed cloud droplet number concentrations, and it also cannot yet simulate frozen cloud.**

*2) LES results: It is hard to make sense of LES results because the models disagree a lot even for a seemingly simple case of no ice and "high" droplet concentration (80 per cc). LES of boundary layer clouds have a long history of intercomparisons, including several for super-cooled clouds (e.g., M-PACE, SHEBA, and ISDAC cases), so the modeling community has a relatively good understanding of what the simulations should behave like under similar conditions. The models used in this study are relatively new and it is not clear how they measure up against an ensemble of previously tested models. Cited "numerical instabilities" that prevented COSMO-LES to be run for the required period of time are very disconcerting and cast shadow on all simulations from that model. Notable differences in LWP between USLALES-SALSA and MIMICA in the CCN80prog_noice case, which are apparent in figure 7, deserve an explanation, especially since UCLALES-SALSA employs an original and relatively untested microphysics scheme, which is different from what was previously used in the UCLALES model. Repeating these simulations with collision-coalescence (or autoconversion)*

*turned off may help to identify the sources for the difference. Explaining this difference may also shed light on much stronger sensitivity of SALSA microphysics to the droplet concentration reduction from 80 to 30 per cc.*

We note that model results in this work are more diverse than in previous model intercomparisons, due to the following factors: We use real atmospheric soundings to initialize our models, rather than idealized linear profiles. Each model uses its own methods to calculate radiative forcing rather than replacing these with simple parameterizations. We don't use nudging that would keep models close to the initial state, and many studies have more constraints on modelled microphysics, resolution, and domain. Also, most of our models in this study do not have common LES or microphysics components, which increases the differences between simulations. While this makes the attribution of differences between the models more challenging, this also yields results from each model that are more representative of what would have been found if this case was studied with each model independently. That is to say that the diversity in model results and the diversity in model sensitivities to perturbations in e.g. CDNC, CCN concentrations, and ICNC are more representative of a "natural" model diversity.

Each of the LES models in this study has either participated in or simulated a case from a previous model intercomparison study, and so their results can be compared against previously-tested models.

We have added the following to the manuscript:

"A comparison of UCLALES-SALSA results against those of a previous model intercomparison based on the second Dynamics and Chemistry of Marine Stratocumulus Field Study (DYCOMSII) can also be found in Tontilla *et al.*, (2017)."

"The MIMICA model has participated in the ISDAC model intercomparison study (Ovchinnikov *et al.*, 2014), and has also been used to simulate the DYCOMSII case (Savre J. *et al.*, 2014), and in both cases it compared well with other models."

"The COSMO model participated in the ISDAC LES model intercomparison study (Ovchinnikov *et al.*, 2014), and the predicted IWP and LWP were within the range of the other models."

The instabilities in COSMO-LES are visible in the full model results as waves in the upper atmosphere, which build after several hours and then propagate through the upper atmosphere. These waves do not reach the boundary layer during the simulations, and thus they don't influence the cloud in the boundary layer.

We have added the following to the manuscript:

"These instabilities are visible in the full model results as waves in the upper atmosphere. These waves do not reach the boundary layer during the simulations, and thus they don't influence the cloud in the boundary layer."

We have determined that the differences in the LWP between the UCLALES-SALSA and MIMICA CCN80prog_NOICE simulations were due primarily to differences in the prescribed subsidence between the two models. We have added the following text to the manuscript, and we include more discussion of and support for this conclusion below.

"The value of the divergence was chosen to be $1.5 \times 10^{-6}$ s$^{-1}$. Preliminary simulations with UCLALES-SALSA showed that a divergence of $1.5 \times 10^{-6}$ s$^{-1}$ was too low in this model to balance radiative cooling and the associated mixing, and the cloud layer would continuously rise at a rate similar to the clouds in the COSMO-LES CDNC30 simulations (e.g. Fig. 3). The increased length of the UCLALES-SALSA simulations, compared to the COSMO-LES simulations (discussed next paragraph), allows the cloud layer to rise to unrealistic altitudes. A larger value of $5.0 \times 10^{-6}$ s$^{-1}$ was therefore used instead for the subsidence in the UCLALES-SALSA simulations. While we do not investigate sensitivities to prescribed subsidence in this study, other studies have shown that differences in prescribed subsidence affect Arctic mixed-phase cloud LWP and IWP (Young *et al.*, 2018)."

"Differences in cloud thickness between MIMICA and UCLALES-SALSA (thickening in MIMICA and thinning with time in UCLALES-SALSA) for this case are primarily due to the different subsidence rates as described in Sect. 3. Simulations performed by UCLALES-SALSA using the same lower subsidence rate as the MIMICA simulations yielded a cloud layer with a similar LWP to the MIMICA simulation (~125 g m$^{-2}$ and 140 g m$^{-2}$, respectively), but the cloud layer rose at an unrealistic rate."

Additional simulations were performed using UCLALES-SALSA with the smaller subsidence used in the MIMICA and COSMO-LES simulations. We also note that the MIMICA simulations were initialised with a liquid cloud layer, while UCLALES-SALSA simulations were initialised with water vapour only. Therefore, an additional simulation with the lower subsidence value and the liquid water profile used in MIMICA was also performed. The results are shown in the following plot:

[Figure]

Further, additional UCLALES-SALSA simulations with different vertical resolutions, turbulence and surface parameters, and subsidence settings were performed, but these changes all had smaller effects on the LWP than the value of the subsidence and the initial liquid water profile.

As autoconversion to rain occurs in MIMICA, but does not occur in UCLALES-SALSA for the CCN80prog_NOICE case, we expect that turning off autoconversion would increase LWP in the MIMICA simulation, and would have no effect on the UCLALES-SALSA simulation. The differences in LWP would therefore be larger between the models than they are currently. We therefore do not expect that the additional simulations would be informative.

*3) Aerosol, CCN, and droplet concentrations: The study targets clouds with extremely low droplet concentrations (~1 to 10 per cc) and more justification is needed to convince readers that these concentrations are relevant for the considered case. This can be accomplished by answering, or at least discussing, the following questions. Are surface-based CCN measurements representative of cloud layer conditions, given that, according to the sounding in figure 2, the cloud layer between 600 and 1000 m appears to be decoupled from the surface?*

While the cloud is initially decoupled from the surface during the "cloudy" period, the cloudy layer descends to the surface during the transition to the "nearly-cloud-free" period, as can be seen in Fig. 1. Stronger support for the representativeness of the

surface-based measurements within the cloud layer comes from helicopter-based profiles of aerosol concentrations.

We have added the following to the manuscript:
"Additionally, helicopter profiles of aerosol number concentrations were performed from 19:53 UTC to 20:13 UTC on Aug. 31st and from 07:32 UTC to 07:55 UTC on Sep. 1st using a condensation particle counter (Kupiszewski et al., 2013). These indicate that the number concentrations of aerosol larger than 14 nm were generally below 10 cm$^{-3}$ up to 850 m altitude during the Aug. 31st profile and up to 500 m altitude during the Sep. 1st profile. With reference to Fig. 1, we note that these heights are similar to the locations of the observed cloud top heights at these time periods, and these altitudes were also similar to temperature inversion base heights observed via a scanning microwave radiometer (Kupiszewski et al., 2013)."

*Supersaturation of 0.2 % for which the CCN concentration were measured may represent conditions in boundary layer clouds in a typical aerosol environment. When the CCN concentration is very low, however, wouldn't higher supersaturation values be achievable even in clouds with moderate updrafts? Another way to look at it is this. Are there smaller aerosol particles, or less efficient CCN, that could be activated in the considered clouds?*

We show below the total observed aerosol number concentrations, in a figure analogous to Fig. 1 from the manuscript. The lower detection limit of the twin differential mobility particle sizer is about 3 nm. While the N3 concentrations are greater and more variable than the N50 concentrations, we note that the N3 concentrations are less than 10 cm$^{-3}$ for most of the "nearly-cloud-free" period, and that median concentrations are 2 cm$^{-3}$.

[Figure]

We agree that this could be better clarified. We have added the following to the manuscript:

"A second identical CCN counter was cycled between supersaturations of 0.11 and 0.73%."

"Total aerosol concentrations as measured by a twin differential mobility particle sizer with a lower detection limit of 3 nm fell generally below 10 $cm^{-3}$, with a median of 2 $cm^{-3}$ during the "nearly-cloud-free" period. Further details on the quality and data processing of ship-based aerosol measurements are available in Heintzenberg and Leck (2012). CCN concentrations measured at supersaturations as high as 0.73% during this period were also below 1 $cm^{-3}$."

We also refer the referee to supporting evidence from previous modelling studies: Birch et al., (2012) found that model results more closely matched observed surface radiation fluxes and surface temperatures if CCN concentrations were reduced to 1 $cm^{-3}$, and Hines and Bromwich (2017) found that model biases against surface radiative flux observations

**for the tenuous cloud regime period were reduced as the prescribed CDNC was reduced to 1 cm$^{-3}$.**

*Finally, are there any measurements, direct or via remote sensing retrievals, of actual droplet concentrations in these clouds that would serve as a target for model simulations?*

**We have added the following to the manuscript:**
**"In-cloud measurements were not performed due to aircraft icing concerns (Tjernstrom, 2014). Additionally, CloudSat+Cloud–Aerosol Lidar with Orthogonal Polarization (CALIOP) cloud retrievals are not available north of 82N, and are therefore unavailable for this case (Kay and Gettelman, 2009). Moderate Resolution Imaging Spectroradiometer (MODIS) retrievals have been shown to underestimate cloud cover in the Arctic, particularly over sea ice and for cloud top heights less than 2 km (Chan and Cosimo, 2013). We therefore consider MODIS-derived cloud information unreliable for this case. Therefore, no reliable observations of cloud droplet number concentrations are available for this case."**

*Technical comment:*
*P 5, lns 19-25: It is worth to provide a brief basic description of the ASCOS campaign, specifying location, overall synoptic situation, whether all measurements were collected from the surface or whether aircraft was involved, etc..*

**We have added the following to the manuscript:**
**"Observations during the ASCOS campaign were obtained on-board the icebreaker Oden, from two measurement sites set up on the ice floe, and by helicopter. However, helicopter observations were restricted to outside of clouds due to safety concerns regarding icing of the aircraft."**
**"These were the last two days of the ice drift period, which ended at about N87°09 W11°01. Observed winds were westerly at the site, with observed wind speeds varying between 2 and 6 m s$^{-1}$ during the two-day period. Conditions were dominated by a high-pressure system over the North Pole, yielding anti-cyclonic winds on the synoptic scale. Observed surface pressures rose from ~1025 to ~1030 hPa during the two-day period."**

*Table 1:*
*(i) typo in coarsest vertical resolution below 2 km for COSMO-LES; it should be smaller than 228.3 m;*

**This is not a typo. The coarsest resolution below 2 km in COSMO-LES is, in fact, 228.3 m. The coarsest resolution in the first 1.5 km is around 35.6 m. The resolution is finer below 1.5 km. We have added an additional row to the table indicating the coarsest resolution below 1.5 km to help avoid misleading the reader.**

*(ii) any reason why LES models don't use identical horizontal grid size? Seems like an extra and unnecessary source of uncertainty to deal with;*

**Each LES model is using different resolutions and domain size, because the LES models have different computational requirements. Each modelling group used their own methods to generate their vertical grids. Different horizontal size grids were chosen to balance computational efficiency with resolving the eddy scales generated at the cloud top.**

*(iii) please include domain size in the table, at least in horizontal directions. For NWP models it is given somewhere in the text, but I don't recall seeing numbers for LES.*

**We have added the requested row to Table 1.**

*P 9, ln 31: remove "five"*

**Done.**

*P 10, ln 8-9: delta_ICNC has units of concentration and, therefore, is not technically a "rate".*

**This has been rephrased to "The change in ICNC due to nucleation of cloud ice in each timestep was therefore:"**

*P 13, ln 16-17: Showing profiles from a single grid column in the middle of the domain is an unorthodox way to compare LES models. Wouldn't the comparison be more robust if domain mean profiles were used? NWP output can then also be averaged over LES domain-size area to be more comparable in terms of represented horizontal scales.*

**The choice of results at the center of the domain was originally made to better show the temporal variability of the model results, consistent with the variability that would be observed at a surface observation site. These temporal variations would be smoothed out by spatial averaging. We ultimately chose to focus our analysis more on attempting to explain the large inter-model differences, and less on comparison with observations. However, we note that such spatial averaging would not alter the conclusions of our study.**

**As an example, we show below the cloud properties from centre-of-domain and averaged over the central 500 km x 500 km (excluding the boundary conditions) for the UM-CASIM prognostic aerosol cases, including both in-cloud and out-of-cloud cells. The domain-mean values tend to be lower due to differences in cloud height among different model columns and the inclusion of cloud-free model cells, but the cloud structure and general sensitivities to initial CCN values and prescribed ICNC values discussed in the text are preserved.**

[Figure]

*P 13, ln 26: "Despite no inclusion of ice..." The sentence does not make sense, because ice won't help models in their current setup to produce clouds. Please re-phrase.*

**We intended to focus the reader's attention on the agreement with the observed LWC values, despite this missing process.**
**We have rephrased this to "All models produce clouds near 1 km altitude. Despite no inclusion of ice processes, the predicted LWC values are generally within a factor of two of those observed during the "cloudy" period."**

*P 14, figure 3: Here and in number of subsequent figures, the deepest mixed layer or highest cloud top seems to be predicted by a model with coarsest vertical resolution (COSMO-NWP). May be worth pointing this out.*

**Thank you, we have added the following to the text in Sect. 5.1: "The cloud-top height predicted by COSMO-NWP is greater than for any other model. This is consistent for all cases in this study simulated by COSMO-NWP. We note that COSMO-NWP has the coarsest vertical resolution of all the models participating in this study."**

*P 14, ln 10: "...adequately resolve..." This statement seems too optimistic. Although time varying advective tendencies at the studied location in NWP models are almost certainly more realistic than constant tendencies imposed on LES, it is not clear how "adequate" they are. Boundaries are 100's km away from that location and so even if the boundary conditions were perfect (they are not), shallow layers and sharp inversions can be significantly eroded during advection, e.g., due to excessive diffusion because of coarse vertical resolution.*

**We note that where inversions are controlled by cloud radiative cooling, these inversions can become better-defined in the high-resolution NWP model results than in the coarser resolution boundary conditions. However, we agree that our wording may have been too strong here.**

**We have edited this portion of the text to the following:**
**"However, other potential causes of the transition could be resolved by the models. In particular, the NWP models would be expected to yield more realistic changes in meteorological conditions due to advective transport, through changes with time in the boundary conditions applied to these models. However, the vertical atmospheric structure at the interiors of the domains will evolve to be different than at the boundaries. Nevertheless, the absence of this transition in these modelling results supports the interpretation that the LWC of these clouds is CCN-limited."**

*P 15, figure 4: High autoconversion rates in COSMO-LES between 200 and 700 m altitudes are puzzling since no cloud water is shown for these levels in figure 3. An explanation is needed here.*

**We note that autoconversion rates predicted by COSMO-LES are often multiple orders of magnitude larger than the other models, especially at low cloud droplet mass mixing ratios. We show below a scatter plot of rain autoconversion rates vs. cloud droplet mass mixing ratios for the CDNC30_NOICE and CDNC03_NOICE cases (now included in the manuscript as Fig. 5). Note that autoconversion rates predicted by COSMO-LES are > 2 x $10^{-6}$ g m$^{-3}$ s$^{-1}$ ( = 2 x $10^{-12}$ g cm$^{-3}$ s$^{-1}$) for droplet mass mixing ratios < 0.01 g kg-1, the lower limit chosen for Fig. 3.**

[Figure]

We have added the following to the text:
"Autoconversion rates greater than $2 \times 10^{-6}$ g m$^{-3}$ s$^{-1}$ exist even in regions where the cloud droplet mass concentration is less than 0.01 g cm$^{-3}$, the lower limit of the colour scale shown in Fig. 3. Autoconversion rates and cloud droplet mass mixing ratios both decrease by about two orders of magnitude from their maximums near cloud top to the layer between 200 and 700 m."

*P 16, ln 1-2: Since different papers often use different formulations and/or notation for gamma distributions, please provide the functional form of that distribution. This would also define your shape parameters, which are currently undefined. Also, in most common notations, nu=0 results in an exponential size distribution, which is often employed for precipitation species, but presents a questionable choice for cloud droplet size spectra. Please clarify what droplet size distributions are used in COSMO and UM-CASIM and, if they are indeed exponential, justify the choice.*

We agree that the equation used for the gamma distribution should be added to the manuscript, and we have done so.

We note the gamma distribution in mass space is defined as:
$$\frac{dN}{dX} = a \, x^{\nu} \exp(-b \, X^{\mu})$$

**where a is the intercept and b the slope of the dN/dx function.**
**The mass is related to the hydrometeor diameter by:**

$X = \frac{\pi}{6}\rho D^3$

**If we differentiate this with respect to D, we get:**

$\frac{dX}{dD} = \frac{\pi}{2}\rho D^2$

**By the chain rule, dN/dD=dN/dx*dx/dD. Therefore:**

$\frac{dN}{dD} = \frac{\pi}{2}\rho D^2 a\, X^\nu exp(-\,b\,X^\mu)$

**We substitute X(D) into the above equation:**

$\frac{dN}{dD} = \frac{\pi}{2}\rho D^2 a(\frac{\pi}{6}\rho D^3)^\nu exp(-\,b(\frac{\pi}{6}\rho D^3)^\mu)$

**Gathering terms, we get:**

$\frac{dN}{dD} = 3a(\frac{\pi}{6}\rho)^{(1+\nu)}D^{(2+3\nu)}exp(-\,b(\frac{\pi}{6}\rho)^\mu D^{3\mu}))$

**So we note that in diameter space, new intercept, slope, and shape parameters can be defined:**

$a_D = 3a(\frac{\pi}{6}\rho)^{(1+\nu)}$

$b_D = b(\frac{\pi}{6}\rho)^\mu$

$\nu_D = 2 + 3\nu$

$\mu_D = 3\mu$

**Which yields:**

$\frac{dN}{dD} = a_D D^{\nu_D} exp(-\,b_D D^{\mu_D})$

**So an exponential distribution in mass space (v=0) would have $v_D$=2 in diameter space, and thus is not an exponential distribution in diameter space.**

*P 16, ln 31: "mass concentration": I think "mixing ratio" was used earlier in the paper. Better to use the same terminology/unit throughout.*

**Thank you, the change has been made.**

*P 18: This is one of the sections, which lacks a clear message. What the reader is supposed to take out of this, except that the model results differ? The description in this section is mundane: turbulence may contribute, collision-coalescence could play a role, and an activation scheme obviously affects how many droplets are formed. Is there anything new that the intercomparison can teach us and that has some broader implications?*

**We have added the following to the text:**
**"This diversity in CDNC of 15-20 cm$^{-3}$ or 20-60 cm$^{-3}$ for the same constant CCN concentrations underscores the variability that exists in model results and model sensitivities to perturbations in aerosol concentrations. Unless the models are constrained through common forcings and common scientific choices, there will remain diversity in model results and model sensitivity, for both LES and NWP models."**

This implies that some caution must be exercised in interpreting the results of any one model. We revisit and emphasize this point in our conclusions section.

We have also added the following:
"As WRF and UM-CASIM have the same activation scheme, and the same minimum updraft velocity, we infer that remaining differences in CDNC are due to differences in sink terms. For the CCN30fixed case, CDNCs are similar in both models, but CDNCs simulated by UM-CASIM are greater in the CCN80fixed case. Therefore, CDNC sinks must be similar in the CCN30fixed case, but faster for WRF in the CCN80fixed case."

*P 19, figure 6: here and on other figures, lines that are called "red" and "purple" are hard to distinguish on this plot.*

We have increased the contrast between the red and purple lines on Figures 6 (now 7) and 11 (now 13).

*P 20, ln 7-9: "It is therefore possible..." As written, it is not clear if narrower spectra in UCLALES-SALSA is the author's speculation or an actual finding of the study. This can be shown clearly and explicitly by plotting the cloud droplet size distributions from different models.*

It is clear from the presence of cloud droplets, but zero autoconversion rates, that the cloud droplet size distribution within UCLALES-SALSA is sufficiently narrow as to contain zero cloud droplets larger than 50 µm. For a cloud droplet size distribution represented by any gamma distribution, the number of cloud droplets larger than 50 µm may be small, but will always be greater than zero.

We have rephrased this sentence to: "The UCLALES-SALSA model resolves narrower cloud droplet size distributions than those represented by the other models in this study, with no cloud droplets large enough to trigger partitioning into the rain category."

*P 20 ln 10-28: I find it odd to pull 1 out of 3 figures in this set of sensitivity experiments out of the main paper into the supplement leaving 1/2 page description in the text. Suggest to put the figure back into the paper.*

We have added the figure into the text as Fig. 9.

*Figure S2, caption: "CCN80prog_noice" should be "CCN30prog_noice"*

Thank you, this has been fixed.

*P 21, figure 7: Non-zero CDNC throughout the vertical column in MIMICA during the model spin up looks odd and should probably be masked outside of clouds.*

**We have masked the CDNC values from MIMICA during the spin-up period.**

*P 22, ln 2-3: Not clear what is meant by the statement "Evaporation of falling rain … transports moisture … to the lower cloud layer", since the cloud layer is presumably saturated. Please re-phrase.*

**This has been rephrased to: "Rain falling from the upper cloud layer evaporates before reaching the lower cloud layer. This transports moisture and aerosol vertically closer to the lower cloud layer, where they are subsequently mixed into the lower cloud layer by turbulence."**

*P 22, ln 21-25: This section summary is not very insightful. Do we need a model intercomparison to state that the decrease in CDNC leads to thinning or collapse of the cloud layer and that this effect is sensitive to cloud-rain partitioning?*

**The effects of CDNC reductions on liquid-phase low-latitude clouds have been previously studied in liquid-phase low-latitude clouds, but it was not clear in advance that Arctic mixed-phase clouds with little diurnal variation, nearly no surface fluxes, and extremely low CCN concentrations would behave in the same way. Additionally, we note that it is not always the case that cloud systems are most sensitive to aerosol concentrations, as opposed to other factors. For example, Miltenberger *et al.*, (2018) examine a case where changes in cloud properties due to changes in aerosol concentrations are difficult to detect compared with meteorological variability.**

**Additionally, the robustness of this response across models for this case and the diversity in the strength of the cloud response to changes in CDNC would not have been clear if only a single model had been used.**

*P 31, ln 1-2: The recommendation to include linkages between aerosol and clouds in models is too general to be useful. Please elaborate. Many, if not most, climate models include prognostic aerosol and CDNC. Whether prediction models are a different story though. Do you recommend that NWP models move to prognostic aerosol too? Should they consider assimilating aerosol information? Please be more specific.*

**We have added the following to the manuscript:**
**"In particular, we recommend that studies are carried out to determine if CCN-controlled cloudiness has a remote effect on important weather phenomena such as mid-latitude blocking. If it does, then we recommend that aerosol-cloud interactions be included in numerical weather prediction models to capture the impact on the more populated mid-latitude regions"**

*P 32, ln 16-26: Specific "threshold" CCN or CDNC values for a given model and a single case are of little value to the broader modeling community. On the other hand, the authors can do more to disentangle some of the effects at play here. E.g., CDNC is certainly a factor in autoconversion rate, but the rate also depends on LWC. The models predict hugely different LWC and it is not clear whether different autoconversion rates are the reason or the consequence of differences in LWC. The most direct way to determine this is of course to swap the autoconversion parameterizations between the models. Another approach would be to examine autoconversion rate normalized by LWC, which may provide some hints into the interplay between the two but could still be affected by other factors.*

**We have rephrased the text to stress the differences in model sensitivity to changes in CDNC or CCN concentrations, as opposed to critical values. We believe that the range of sensitivities to changes in CDNC or CCN concentrations is of interest to the modelling community, being one indicator of uncertainties in cloud response to changes in aerosol concentrations. Our results suggest that some caution is necessary in interpreting the results of any single model, including the sensitivities of model results to perturbations in aerosol concentrations.**

**The same autoconversion scheme (Seifert and Beheng, 2006) is used in COSMO-LES, COSMO-NWP and WRF, and a similar scheme is used in MIMICA (Seifert and Beheng, 2001). COSMO-LES, COSMO-NWP, and WRF all prescribed the same maximum cloud droplet radius to be used for autoconversion (40 μm), and MIMICA used a smaller value (25 μm), yet yields smaller autoconversion rates than COSMO-LES, and similar autoconversion rates to COSMO-NWP and WRF. It is therefore unlikely that the differences in the model results are due primarily to different autoconversion schemes, and we would not expect an additional sensitivity study based on swapping the autoconversion schemes in the models to be fruitful.**

**We plot above the autoconversion rates vs. the cloud droplet mass mixing ratios for the CDNC30_NOICE and CDNC03_NOICE cases, and this has been added to the paper as Fig. 5. We note that the autoconversion rate for a given value of the cloud droplet mass mixing ratio can vary by orders of magnitude between different models. Within a given model, the autoconversion rate for a given model at a given cloud droplet mass mixing ratio is well-constrained, with the exception of the COSMO-LES results. The variability in the COSMO-LES results is likely a feedback from the larger mass mixing ratios of rain predicted by this model, which increase autoconversion rates in the Seifert and Beheng (2006) scheme. Given that the autoconversion schemes are the same or similar between many of the models, we infer that these differences are due primarily to differences in the cloud droplet size distribution.**

**We have added the following to the text, in the discussion of the CDNC30_NOICE case: "As cloud droplet activation is prescribed in this case, activation is similarly treated in all models except for UM-CASIM, and no frozen processes are permitted in this case, we**

believe that the differences in autoconversion rates per unit cloud droplet mass are due primarily to the differences in the representation of the cloud droplet size distribution."

We have also added the following to our conclusions section:
"Large differences in autoconversion rates per unit cloud droplet mass were simulated despite a similar treatment of autoconversion in four of the models, even in cases with prescribed cloud droplet activation and no frozen cloud processes permitted."

We also note that while we don't examine the autoconversion rate normalised by the LWC, we do briefly discuss the cloud droplet mass mixing ratio normalised by the autoconversion rate, approximately the inverse of the former value. This discussion begins on page 14, line 16 in the version of the manuscript currently online.

*P 33, ln 9-13: Surface fluxes no doubt are important in the Arctic, but how is this relevant to the current study. Aren't the fluxes set to zero for LES models and very small in NWP models? And isn't the cloud layer, in fact, decoupled from the surface to begin with?*

While the cloud is initially decoupled from the surface during the "cloudy" period, the cloudy layer descends to the surface during the transition to the "nearly-cloud-free" period, as can be seen in Figure 1.

Surface fluxes are indeed prescribed to be zero for the LES models. In the NWP models, surface fluxes are small as long as a sufficiently thick cloud layer is predicted. As shown in Fig. 11 (now 14) and Fig. 12 (now 15), longwave cooling of the surface is significant under thin-cloud or cloud-free conditions. In the NWP models where surface temperatures are not prescribed, this can yield a significant sensible heat flux into the surface, cooling the near-surface layer and increasing the near-surface stability. The effect on the potential temperature profile in UM-CASIM can be seen in Fig. 8 (now 10). This also clearly has an effect in the WRF CDNC03 simulations, as the cloud base intersects the surface in these simulations, as shown in Figures 5 (now 6) and 6 (now 7).

We have edited the text to make this more clear.

*Referee #2:*

*General comments:*

*The authors test the sensitivities of simulated Arctic clouds to aerosol perturbations represented differently by multiple variables, such as prescribed droplet number concentrations, prescribed or variable cloud condensation nuclei (CCN) concentrations, and also prescribed ice crystal number concentrations, using three large-eddy simulation (LES) models and three numerical weather prediction (NWP) models. Microphysical processes in the simulated clouds are investigated in detail. The sets of simulations listed in Table 2 are well designed to test the sensitivity of clouds to different perturbations. Observational data from a field campaign is also presented, which helps the evaluation of the simulation results. They conclude that the clouds (or their water content) are CCN-limited, meaning that the properties of the clouds are heavily dependent on the concentrations/existence of CCN. They also conclude that changes in these Arctic clouds may have impacts on the surface radiative balance since these clouds tend to have a warming effect. Figure 11 and 12 are especially interesting and highlight the findings from this paper. I would like to suggest some minor revisions/questions/comments below;*

*Specific comments:*

*Figure 1: Is there any observational data of surface precipitation available?*

**Surface precipitation during this period was close to the lower detection limit of the available instrumentation.**

*Page 7 line 1-2: Does this mean that microphysical processes in these clouds are purely liquid-based, not involving ice, due to possibly the lack of ice-nucleating particles (INPs)? Or ice often exists in these clouds (e.g. Figure 1), but just precipitation processes are dominantly through warm-rain?*

**Implicitly included in the hypothesis is that precipitation processes are dominantly through warm-rain. We have added the following to the text:**
**"It is implicit in this hypothesis that in-cloud precipitation occurs predominantly through liquid-phase processes, although frozen-phase processes could contribute to precipitation formation, and glaciation would be an alternate cause of cloud dissipation."**

*Page 11 line 6-8: Does the fact that MIMICA is initialized at much earlier than the other two LES models have any impacts on the results?*

**We have added the following to the text:**
**"As we have not prescribed any time-varying surface fluxes or large-scale forcings for**

**the LES models, and the diurnal cycles in this case are weak, the LES model results are largely independent of the start time for this case."**

**We also note that with few exceptions, LES model results evolve towards a semi-stable state, and the statistics shown in Figures 6 (now 7), 10 (now 13), 11 (now 14), and 12 (now 15) do not depend strongly on the length of time used for analysis, nor the amount of time from the beginning of the simulation. These exceptions include the MIMICA ICNC1p00 cases, where the cloud glaciated. In the CDNC30_ICNC1p00 case, it is possible that the COSMO-LES simulation would glaciate if the simulation was extended for a longer time period, but we believe that this is unlikely, as the cloud does not thin with time during the resolved simulation.**

*Figure 2: It may be nice to also show actual temperature so that the temperatures for cloud-base and cloud-top can be roughly estimated.*

**We have added the absolute temperature to Fig. 2.**

*Page 13 line 16-17: Does this mean that the results are taken from a single column at the center of the simulation domain? According to Figure S1 there seems to be a wide spatial variation in simulated results, but are analyzed results quite similar/different if domain-averaging or other methods are used, instead of extracting data from the center of the domain?*

**Yes, the results are taken at the centre of the domain.**
**Please see our response to Referee #1 on a related question.**

*Figure 3 and 5: Since the observed quantity is LWC, how does simulated LWC look like (maybe add it in the third row)? Also, it would be helpful if you add dashed lines to indicate the defined "cloudy" and "nearly-cloud-free" periods in the "Observed LWC" figure.*

**We have added LWC to these two figures. We have also added dashed lines to indicate the cloudy and nearly-cloud-free periods in the observed LWC subplot, as requested.**

*Figure 4 and 5 captions: Although it says "Rain sedimentation tendencies for COSMO-NWP are not available.", the quantities are still plotted (bottom row, middle column), if I'm understanding it correctly?*

**You are correct, the rain sedimentation rates are plotted. The statements were in error, and have been removed.**

*Figure 6: Can you maybe add a column for LWC so that the comparison of observation and simulations is possible?*

**We have added a column for LWC, as requested.**

*Figure S2 caption: "CCN80prog_noice" should be modified to "CCN30prog_NOICE"*

**Thank you, this has been fixed.**

*Page 22 line 30-32: While simulated results may be relatively insensitive to the inclusion of ice, IWC seems not to be negligible in Figure 1 as compared to LWC, though scales are different for LWC and IWC there. Does this mean that simulations are underrepresenting ice mass?*

**The models are under-representing ice mass in this case. We now state this explicitly in the following paragraph which discusses IWC.**

*Figure 9: Although I'm aware of the consistent color scale for rain, ice, snow, and graupel, can you change the color scale for snow so that more information can be seen in colors? Or if snow and graupel masses do not play a major role in the whole microphysical processes in those clouds, the bottom two rows could be omitted. Also, I suggest adding observed and simulated IWC to the bottom row if possible.*

**We have lowered the colour scale for snow by one order of magnitude. We have added IWC with LWC for the CDNC30_ICNC0p20 case as a separate figure in the text (Fig. 11).**

*Page 25 line 30-31: I wonder if some of the differences are due to other reasons, such as ambient conditions (simulated meteorology, especially when LES and NWP simulations are compared) and/or model resolutions, for example?*

**While we expect that process parameterization and size distribution representation are the primary contributors to the differences in process rates, it is likely that there are also contributions from simulated meteorology and model resolution.**

**We have added the following to the text:**
**"Additional contributions to these differences would come from differences in model meteorology and model resolution."**

*Figure 10: Maybe some of the columns that are not discussed much in the text can be omitted, so that each plot becomes a little larger?*

**As each column is discussed in the text, and each process rate is significant for at least one of the models, it is not clear to us which one could be omitted. We have therefore retained all of the columns.**

*Figure 11 and 12: If I understand the figure and the timeline in Figure 1 correctly, the boxes in these figures represent the simulated results for the transition period from cloudy to nearly-cloud-free period (1200UTC August 31st – 1200UTC September 1st). However, it does*

*not include the cloudy period itself. That means, if one were to evaluate model performance or compare simulations with observations in this figure, should the boxes lie somewhere between hatched and shaded regions? Maybe the shading and hatching are for reference, but can you also provide the observed average over the same period (1200UTC August 31st – 1200UTC September 1st)? Also, hatching seems to be a little weak and hard to see, so I suggest thickening it. Additionally, either shading for cloud-free period or hatching for cloudy period can be modified to a different color (currently both of them look grey), so that their differences become clearer (e.g., the row for net LW).*

**If the tenuous cloud hypothesis is correct, then none of the prescribed CDNC or fixed CCN concentration experiments will capture the "cloudy" to "nearly-cloud-free" transition. However, the differing prescribed CDNC and CCN concentrations may be more or less representative of the conditions during each period independently. Therefore, in Fig. 11 (now 14), we think that it is appropriate to compare the (usually) stable cloud states reached by each simulation with the "cloudy" and "nearly-cloud-free" observed cloud states. We therefore choose the time period for model analysis to appropriately capture the model results while removing the time necessary to reach the (usually) stable cloud states, instead of trying to match simulated time periods to observed time periods.**

**We have added the following to the text:**
**"We note that we do not expect the prescribed CDNC or prescribed CCN cases to capture the "cloudy" to "nearly-cloud-free" transition, so we do not attempt to sample the models during these observed time periods. However, if the tenuous cloud hypothesis is correct, the cloud states resulting in each model for the cases with greater prescribed CDNC and CCN concentrations would be expected to be more representative of the "cloudy" period, and the cloud states for the cases with lesser prescribed CDNC and CCN concentrations would be expected to be more representative of the "nearly-cloud-free" period."**

**The prognostic aerosol simulations would include the model processes necessary to potentially capture the "cloudy" to "nearly-cloud-free" transition. However, in most cases, the LWP and IWP reach stable values that do not vary strongly with time. This can be seen for the LWP in Figs. 7 (now 8), S2 (now 9), and 8 (now 10). The UCLALES-SALSA CCN30_NOICE case is a notable exception. We therefore felt that it would be clearest to the reader to use a consistent presentation style to Fig. 11 (now 14).**

**We have darkened the hatching and shading to make them easier to see, and we have changed the colour of the shading to green, to make it easier to distinguish from the hatching on the net LW subplots.**

*Figure 11 and 12 caption: Modify "indicates" to "indicate" in "...the shaded regions indicates..."*

**Thank you, this has been fixed.**

*Page 26 line 4: Remove "be" in "This would be result in..."*

**Thank you, this has been fixed.**

*P.26 line 8-9: Does COSMO-LES have 7 hours for spin-up + 9 hours for analysis? At page 11 line 3, it was stated to be 2 hours of spin-up.*

**We have been imprecise with our use of "spin-up". In the first instance, it was the time before all model processes were in effect. In the latter instance, it is the time allowed for the model to reach a stable state before model results are used for analysis. We now use "spin-up" exclusively for the former purpose in the text.**

*Page 28 line 33: Add "is" in "...This primarily due to..."*

**Thank you, this has been fixed.**

*It is better to consistently capitalize the word "_NOICE" throughout the paper, since there are currently places with "_noice" instead.*

**We agree. The change has been made.**

*Figures: Minor suggestion, but it may be a good idea for all figures to have consistent labels. For example, it may be in the form of "1e-01", "0.1", or "10ˆ1" (e.g., Figure 6).*

**We believe that it is clearest to the reader to express some variables, such as concentrations and potential temperatures, with ordinary formatting. Due to the small magnitudes, the tendencies must be expressed using scientific notation. However, we now attempt to be more consistent in using the same notation for the same variables across different figures.**

**References:**

Browning, K. A., Betts, A., Jonas, P. R., Kershaw, R., Manton, M., Mason, P. J., Miller, M., Moncrieff, M. W., Sundqvist, H., Tao, W. K., Tiedtke, M., Hobbs, P. V., Mitchell, J., Raschke, E., Stewart, R. E. and Simpson, J.: The GEWEX Cloud System Study (GCSS), Bull. Am. Meteorol. Soc., 74(3), 387–399, doi:10.1175/1520-0477(1993)074<0387:TGCSS>2.0.CO;2, 1993.

Birch, C. E., Brooks, I. M., Tjernström, M., Shupe, M. D., Mauritsen, T., Sedlar, J., Lock, a. P., Earnshaw, P., Persson, P. O. G., Milton, S. F. and Leck, C.: Modelling atmospheric structure, cloud and their response to CCN in the central Arctic: ASCOS case studies, Atmos. Chem. Phys., 12(7), 3419–3435, doi:10.5194/acp-12-3419-2012, 2012.

Chan, M. A. and Comiso, J. C.: Arctic Cloud Characteristics as Derived from MODIS, CALIPSO, and CloudSat, J. Clim., 26(10), 3285–3306, doi:10.1175/JCLI-D-12-00204.1, 2013.

Kay, J. E. and Gettelman, A.: Cloud influence on and response to seasonal Arctic sea ice loss, J. Geophys. Res. Atmos., 114(18), 1–18, doi:10.1029/2009JD011773, 2009.

Klein, S. A., McCoy, R. B., Morrison, H., Ackerman, A. S., Avramov, A., Boer, G. de, Chen, M., Cole, J. N. S., Del Genio, A. D., Falk, M., Foster, M. J., Fridlind, A., Golaz, J.-C., Hashino, T., Harrington, J. Y., Hoose, C., Khairoutdinov, M. F., Larson, V. E., Liu, X., Luo, Y., McFarquhar, G. M., Menon, S., Neggers, R. A. J., Park, S., Poellot, M. R., Schmidt, J. M., Sednev, I., Shipway, B. J., Shupe, M. D., Spangenberg, D. A., Sud, Y. C., Turner, D. D., Veron, D. E., Salzen, K. von, Walker, G. K., Wang, Z., Wolf, A. B., Xie, S., Xu, K.-M., Yang, F. and Zhang, G.: Intercomparison of model simulations of mixed-phase clouds observed during the ARM Mixed-Phase Arctic Cloud Experiment. I: single-layer cloud, Q. J. R. Meteorol. Soc., 135(641), 979–1002, doi:10.1002/qj.416, 2009.

Kupiszewski, P., Leck, C., Tjernström, M., Sjogren, S., Sedlar, J., Graus, M., Müller, M., Brooks, B., Swietlicki, E., Norris, S. and Hansel, A.: Vertical profiling of aerosol particles and trace gases over the central Arctic Ocean during summer, Atmos. Chem. Phys., 13(24), 12405–12431, doi:10.5194/acp-13-12405-2013, 2013.

Hines, K. M. and Bromwich, D. H.: Simulation of Late Summer Arctic Clouds during ASCOS with Polar WRF, Mon. Weather Rev., 145(2), 521–541, doi:10.1175/MWR-D-16-0079.1, 2017.

Moeng, C.-H., Cotton, W. R., Stevens, B., Bretherton, C., Rand, H. A., Chlond, A., Khairoutdinov, M., Krueger, S., Lewellen, W. S., MacVean, M. K., Pasquier, J. R. M., Siebesma, A. P. and Sykes, R. I.: Simulation of a Stratocumulus-Topped Planetary Boundary Layer: Intercomparison among Different Numerical Codes, Bull. Am. Meteorol. Soc., 77(2), 261–278, doi:10.1175/1520-0477(1996)077<0261:SOASTP>2.0.CO;2, 1996.

Miltenberger, A. K., Field, P. R., Hill, A. A., Shipway, B. J. and Wilkinson, J. M.: Aerosol-cloud interactions in mixed-phase convective clouds. Part 2: Meteorological ensemble, Atmos. Chem. Phys. Discuss., (February), 1–37, doi:10.5194/acp-2018-167, 2018.

Ovchinnikov, M., Ackerman, A. S., Avramov, A., Cheng, A., Fan, J., Fridlind, A. M., Ghan, S., Harrington, J., Hoose, C., Korolev, A., McFarquhar, G. M., Morrison, H., Paukert, M., Savre, J., Shipway, B. J., Shupe, M. D., Solomon, A. and Sulia, K.: Intercomparison of large-eddy simulations of Arctic mixed-phase clouds: Importance of ice size distribution assumptions, J. Adv. Model. Earth Syst., 6(1), 223–248, doi:10.1002/2013MS000282, 2014.

Petch, J. C., Willett, M., Wong, R. Y. and Woolnough, S. J.: Modelling suppressed and active convection. Comparing a numerical weather prediction, cloud-resolving and single-column model, Q. J. R. Meteorol. Soc., doi:10.1002/qj.109, 2007.

Petch, J., Hill, A., Davies, L., Fridlind, A., Jakob, C., Lin, Y., Xie, S. and Zhu, P.: Evaluation of intercomparisons of four different types of model simulating TWP-ICE, Q. J. R. Meteorol. Soc., 140(680), 826–837, doi:10.1002/qj.2192, 2014.

Savre, J., Ekman, A. M. L. and Svensson, G.: Technical note: Introduction to MIMICA, a large-eddy simulation solver for cloudy planetary boundary layers, J. Adv. Model. Earth Syst., 6, 1–20, doi:10.1002/2013MS000292, 2014.

Seifert, A. and Beheng, K. D.: A double-moment parameterization for simulating autoconversion, accretion and selfcollection, Atmos. Res., 59–60, 265–281, doi:10.1016/S0169-8095(01)00126-0, 2001.

Seifert, A. and Beheng, K. D.: A two-moment cloud microphysics parameterization for mixed-phase clouds. Part 1: Model description, Meteorol. Atmos. Phys., 92(1), 45–66, doi:10.1007/s00703-005-0112-4, 2006.

Tjernström, M., Leck, C., Birch, C. E., Bottenheim, J. W., Brooks, B. J., Brooks, I. M., Bäcklin, L., Chang, R. Y.-W., de Leeuw, G., Di Liberto, L., de la Rosa, S., Granath, E., Graus, M., Hansel, a., Heintzenberg, J., Held, a., Hind, a., Johnston, P., Knulst, J., Martin, M., Matrai, P. a., Mauritsen, T., Müller, M., Norris, S. J., Orellana, M. V., Orsini, D. a., Paatero, J., Persson, P. O. G., Gao, Q., Rauschenberg, C., Ristovski, Z., Sedlar, J., Shupe, M. D., Sierau, B., Sirevaag, a., Sjogren, S., Stetzer, O., Swietlicki, E., Szczodrak, M., Vaattovaara, P., Wahlberg, N., Westberg, M. and Wheeler, C. R.: The Arctic Summer Cloud Ocean Study (ASCOS): overview and experimental design, Atmos. Chem. Phys., 14(6), 2823–2869, doi:10.5194/acp-14-2823-2014, 2014.

Tonttila, J., Maalick, Z., Raatikainen, T., Kokkola, H., Kühn, T. and Romakkaniemi, S.: UCLALES–SALSA v1.0: a large-eddy model with interactive sectional microphysics for aerosol, clouds and precipitation, Geosci. Model Dev., 10(1), 169–188, doi:10.5194/gmd-10-169-2017, 2017.

Varble, A., Zipser, E. J., Fridlind, A. M., Zhu, P., Ackerman, A. S., Chaboureau, J., Collis, S., Fan, J., Hill, A. and Shipway, B.: Evaluation of cloud-resolving and limited area model intercomparison simulations using TWP-ICE observations: 1. Deep convective updraft properties, J. Geophys. Res. Atmos., 119(24), 13,891-13,918, doi:10.1002/2013JD021371, 2014a.

Varble, A., Zipser, E. J., Fridlind, A. M., Zhu, P., Ackerman, A. S., Chaboureau, J. P., Fan, J., Hill, A., Shipway, B. and Williams, C.: Evaluation of cloud-resolving and limited area model intercomparison simulations using twp-ice observations: 2. precipitation microphysics, J. Geophys. Res., doi:10.1002/2013JD021372, 2014b.

Young, G., Connolly, P. J., Dearden, C. and Choularton, T. W.: Relating large-scale subsidence to convection development in Arctic mixed-phase marine stratocumulus, Atmos. Chem. Phys., 18(3), 1475–1494, doi:10.5194/acp-18-1475-2018, 2018.

---

## Author Response (AR2)

We thank the referee for their comments. We copy their comments in italics below, and respond to them point-by-point.

*The authors have answered my questions and revised the manuscript accordingly. The revised manuscript seems to include more necessary information and statements. I have just a few more comments on the revised manuscript.*

*1.	Although the authors have responded to my previous question about extracting the data from the column in the middle, it is better to show domain-average values or other spatially averaged quantities. If the data from one column is used, then it is necessary to show that the column is representative (in terms of cloud properties and cloud response to sensitivity tests) of simulated clouds in the entire domain, and also clarify the spatial variability of the results.*

Unfortunately, it would not be possible to process the domain mean values for all variables and all models and to re-do the analysis before the deadline for revisions. However, we can do more to show that the centre-of-domain values are representative.

We show below figures similar to Fig. 14 and Fig. 15 in the revised manuscript, showing spatial variability across a 100 km$^2$ area centred on the centre-of-domain, instead of temporal variability. These figures will be included with the manuscript as supplementary figures. For each case and for each of the NWP models, we show temporal medians (over the period after Aug. 31$^{st}$, 1200 UTC) of the spatial medians and interquartile ranges over a 100 km$^2$ area for LWP, IWP, and net surface LW radiative flux. For comparison, we include as black dots the temporal medians of the centre-of-domain values (denoted by the horizontal black bars in Fig. 14 and Fig. 15).

We note first that the temporal medians of the centre-of-domain values generally fall within the temporal medians of the spatial interquartile range. We also note that the spatial median values show qualitatively the same sensitivities as the centre-of-domain values to changes in prescribed CDNC, prescribed ICNC, and fixed or prognostic CCN: LWP increases with increases in CDNC or CCN, and is less sensitive to changes in ICNC. COSMO-NWP shows less sensitivity to changes in CDNC or CCN than WRF or UM-CASIM. IWP increases with ICNC. We also note with reference to Fig. 14 and Fig. 15 that the temporal medians of the spatial interquartile range is generally smaller than the temporal interquartile range of the centre-of-domain values. That is, the spatial variability across the 100 km$^2$ area is smaller than the temporal variability in the centre-of-domain values. We therefore do not expect that use of spatial averages rather than centre-of-domain values would change our conclusions.

We do not have the spatial statistics or spatially-averaged values readily available for the LES models, but we expect that the LES models would show less spatial variability than the NWP models. The LES models did not have time-varying boundary conditions and

the surface properties were homogenous, so there are fewer sources of spatial variability in the LES models than the NWP models.

We have added the following text to the manuscript:
"In order to assess this, we show statistics from the NWP models over a 100 km$^2$ area in the centre of the domain in the supplement material (Fig. S1-S3). Fig. S1 shows characteristics of the distribution of LWP and IWP within the specified 100 km$^2$ area as simulated by the three NWP models for the CDNC30_ICNC0p20 case. Figure S2 and Fig. S3 show statistics of LWP, IWP, and net surface longwave radiation for the NWP models for all of the sensitivity studies. We note that the centre-of-domain values are nearly always within the interquartile range of the 100 km$^2$ area values. Furthermore, centre-of-domain values are sufficiently close to the domain medians, and have similar enough responses to changes in CDNC, CCN, and ICNC, as not to change the conclusions of our study. We expect less spatial variability in the LES models than the NWP models, which were run with periodic boundary conditions and fixed surface fluxes. Thus, the centre-of-domain points are representative for the domain in both NWP and LES models."

[Figure]

Figure S2:
Water paths and net longwave radiation for all simulations without aerosol processing. Top row: liquid water path, middle row: ice water path, bottom row: surface net longwave radiation. Each subplot shows results from a single model. From left to right: COSMO-NWP, WRF, and UM-CASIM. Simulations with prescribed CDNCs of 3 cm$^{-3}$

(CDNC03) and 30 cm$^{-3}$ (CDNC30) are shown as blue and red boxes, respectively, and simulations with prescribed CCN concentrations of 30 cm$^{-3}$ (CCN30fixed) and 80 cm$^{-3}$ (CCN80fixed) are shown as purple and turquoise boxes, respectively. Within each subplot, the ICNC is increased from left to right as 0 L$^{-1}$, 0.02 L$^{-1}$, 0.2 L$^{-1}$, and 1 L$^{-1}$. Boxes show the interquartile range over a 100 km$^2$ area in the centre of the domain, and the black horizontal lines denote the medians. Black dots show the centre-of-domain values (same as the black horizontal lines in Fig. 15). For all values, the median over model results after Aug. 31$^{st}$ 1200 UTC is shown. Hatched regions indicate observed interquartile range for the "cloudy" period, and the green shaded regions indicate the range for the "nearly-cloud-free" period.

[Figure]

**Figure S3:**

Water paths and net longwave radiation for all simulations with aerosol processing. Top row: liquid water path, middle row: ice water path, bottom row: surface net longwave radiation. Each subplot shows results from a single model. Left column: COSMO-NWP. Right column: UM-CASIM. Simulations with an initial CCN concentration of 3 $cm^{-3}$ (CCN03prog), 30 $cm^{-3}$ (CCN30prog) and 80 $cm^{-3}$ (CCN80prog) are shown as purple, orange and green boxes, respectively. Within each subplot, the ICNC is increased from left to right as 0 $L^{-1}$, 0.02 $L^{-1}$, 0.2 $L^{-1}$, and 1 $L^{-1}$. Boxes show the interquartile range over a 100 $km^2$ area in the centre of the domain, and the black horizontal lines denote the medians. Black dots show the centre-of-domain values (same as the black horizontal lines in Fig. 15). For all values, the median over model results after Aug. 31$^{st}$ 1200 UTC is shown. Hatched regions indicate observed interquartile range for the "cloudy" period, and the green shaded regions indicate the range for the "nearly-cloud-free" period.

*2.       The authors mention the general underrepresentation of IWC in the simulations in the revised manuscript (6.1 first paragraph), and I wonder if that prevents some of the simulated clouds from glaciating/dissipating. The cloud in the MIMICA run with ICNC of 1 /L gets glaciated/dissipated, but even for that run IWP seems not as high as observed (according to Figure 14). This may be worth mentioning in the section of Conclusions.*

We note that the IWP in the MIMICA CDNC30_ICNC1p00 is greater than 11 g m$^{-2}$ (compared to an observed median value of 8.5 g m$^{-2}$ for the "cloudy" period) for the first 12 hours of simulation, but this is not captured in Fig. 14. The IWP decreases after this point, due to the dissipation of the cloud. We also note the large uncertainty (a factor of two) in the observed IWP.

We have added the following sentence to the conclusions section: "As the IWP was generally less than the observed IWP, it is possible that the contribution of glaciation to dissipation was also underestimated."

*3.       It would be helpful to add labels of (a), (b), (c),… to the plots in Figures 7 and 13, and use them when discussing relevant processes in the text.*

We have added labels to the subplots. However, as both of these plots have more than 26 subplots, we have adopted the convention '(aa)', where the first letter refers to the column and the second letter refers to the row.

[revised manuscript text omitted]